# Feature Learning for Interpretable, Performant Decision Trees

**Jack H. Good, Torin Kovach, Kyle Miller, Artur Dubrawski**
Carnegie Mellon University
{jhgood,tkovach,mille856,awd}@andrew.cmu.edu

## Abstract

Decision trees are regarded for high interpretability arising from their hierarchical partitioning structure built on simple decision rules. However, in practice, this is not realized because axis-aligned partitioning of realistic data results in deep trees, and because ensemble methods are used to mitigate overfitting. Even then, model complexity and performance remain sensitive to transformation of the input, and extensive expert crafting of features from the raw data is common. We propose the first system to alternate sparse feature learning with differentiable decision tree construction to produce small, interpretable trees with good performance. It benchmarks favorably against conventional tree-based models and demonstrates several notions of interpretability of a model and its predictions.

## 1   Introduction

In recent years, AI has made its way into increasingly critical applications, leading to the popularization of the topic of *Trustworthy AI*; AI should not only perform well in its primary task, but exhibit properties that earn trust, such as robustness, provable adherence to safety specifications, and interpretability to humans. For example, predictors that assist medical diagnosis are more useful and more likely to be trusted if their predictions are easily understood by medical personnel.

Interpretability is highly subjective and application-dependent by nature, and there is no universal standard of what makes a system highly interpretable. However, within machine learning (ML), there are some generally agreed-upon rules of thumb; for example, smaller models are considered more interpretable than structurally similar but larger ones, and rule-based representations such as decision trees are considered more interpretable than "black box" models such as neural networks.

Decision trees in particular are often used as the quintessential example of interpretable ML; they can be interpreted holistically (also called *global interpretation*) as a hierarchical partitioning of the input based on simple decision rules, and individual predictions can be interpreted (also called called *local interpretation*) as a series of such rules on a path from the tree's root to a leaf. Conventionally, these decision rules have the form $x_i < t$ for some feature $i$ of input vector $\mathbf{x}$ and decision threshold $t$, thus representing axis-aligned hyperplanes in the input space, but many other decision rules are possible. For example, rules of the form $\sum_i a_i x_i < b$ for some $\mathbf{a}$, $b$ are called oblique splits, and the trees using them oblique decision trees (ODTs).

In practice, however, there are barriers to the practical application of decision trees as interpretable models. First, due to various factors such as greedy growth, axis-aligned decision rules, and the lack of any objective to promote robust, large-margin predictions, they tend to generalize poorly to yet-unseen data; while this can be somewhat alleviated by pruning, the most effective and popular solution is to use large ensembles with strategies such as bagging as in Random Forests [9], random thresholds as in ExtraTrees [22], or boosting as in XGBoost [11]. In addition, decision trees often grow quite deep and use many decision rules for any given prediction, even for extremely simple data. Together, these make both global and local interpretation much more difficult.

37th Conference on Neural Information Processing Systems (NeurIPS 2023).

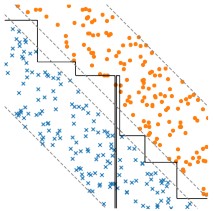 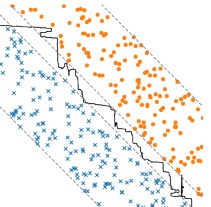 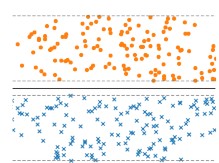 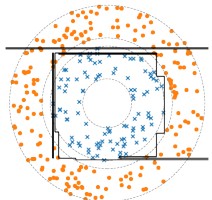

| (a) A decision tree is relatively complex and generalizes poorly. | (b) A random forest is very complex and generalizes better, but not perfectly. | (c) After rotation, a decision tree is simple and generalizes perfectly. | (d) Other data needs a different transformation to learn a simple, general tree. |

Figure 1: An example of trees fitted to toy data to motivate the potential benefit of learning simple feature transforms. Points were sampled uniformly in the bands denoted by dashed lines.

We posit that these barriers are due, at least in part, to the sensitivity of decision trees to transformations of the input resulting from greedy construction and simple decision rules. Of these, key limitation is the latter; even if we replace greedy construction with a perfect tree learner, simple distributions can nonetheless require an arbitrarily large axis-aligned tree to fit. However, after a transformation of that distribution, even a greedy algorithm can produce a small tree, as shown in Section 3.5. Figure 1 gives a motivating example of a case where trees do poorly as interpretable learners on simple data, but succeed after a transformation. The sensitivity to the feature representation is also related to other limitations of tree-based models, such as their dependence on features hand-crafted using expert knowledge to achieve good performance on some problems, and their overall weakness on other problems, such as image classification.

This motivates the idea of learning transformations of the input features, or equivalently, more expressive decision rules such as oblique splits, to strike a balance between complexity of the rules and complexity of the overall model. This learning problem is not easy; axis-aligned decision rule learning, as in the popular CART algorithm [8], is only practical because the search is restricted to a finite set of candidate rules that is efficiently exhaustively searchable. However, finding even optimal oblique splits, which are among the simplest decision rules, is NP-Hard [26], and even oblique splits may not be considered interpretable unless they are sparse, that is, having few nonzero coefficients.

Nonetheless, methods have been developed to learn such trees, either by heuristically choosing splits in a tree construction process, or by constructing an axis-aligned tree and then optimizing the parameters of the splits (see Section 2 for examples). However, these suffer from either greedy, suboptimal choice of split, or from keeping the original structure of the tree. They are also usually restricted to a certain kind of split, such as oblique splits.

Here we propose a novel, flexible framework that is the first to alternate optimization of feature transformation parameters with complete regrowth of the tree. This is made possible by recent advances in fuzzy decision tree methods. Refitting the tree completely throughout training reshapes the tree and moves thresholds as needed; it also maintains the property that each splitting rule is maximally informative given the available features and data. Together these result in smaller, more interpretable trees for a given performance. Moreover, the flexibility of this framework to support various feature transformations provides a foundation for future work focusing on interpretable prediction with other data modalities, such as images and time series.

## 2 Related Work

Decision trees are one of the oldest techniques in machine learning still widely in use today. It has long been known that constructing optimal binary decision trees is NP-Complete [29], so most algorithms use a greedy approach to "grow" a tree from root to leaf by successively partitioning the data by choosing the split that maximally improves the purity of labels in each partition. The partitions are usually axis-aligned, that is, involving only one feature of the input, to make the search practical. One of the most popular decision tree algorithms today is Classification And Regression Trees (CART) [8]. Unconstrained decision trees can always fit separable data perfectly, but are prone to overfitting; to improve generalization, growth can be constrained, but more often, trees are used in

randomized ensembles such as random forests [9] and ExtraTrees [22], or in boosted ensembles such as AdaBoost [19] and XGboost [11].

There is extensive work using decision trees extracted from more complex models either to entirely replace the model, or as an interpretable proxy. Examples include [12, 37, 43, 3, 35, 36, 25, 7].

Other work attempts to improve the interpretability of decision trees themselves by introducing new decision rules. These rules are more complex than the axis-aligned threshold rules, but compensate by improving performance of single-tree models while reducing their size. For example, [6] uses a branch-and-bound algorithm to search for bivariate splits. [16, 17, 23] use parallel coordinates, which incorporate information from multiple data features while being easy to visualize. [13] uses bilevel optimization to find nonlinear splits for growing trees.

The most common augmented decision rule, however, is linear splits, that is, splitting using arbitrary hyperplanes rather than axis-aligned hyperplanes. These are often called oblique splits, and the trees that use them oblique decision trees (ODTs). Algorithms for growing ODTs are first proposed along with CART [8]. [34] later discusses the difficulty of finding optimal linear splits due to local minima and proposes a heuristic using randomization and deterministic hill climbing.

Another approach is to grow an axis-aligned tree, fix the structure, then treat it as a parametric model in order to learn coefficients for linear splits. This can be made possible using fuzzy splits, wherein a decision is a weighted combination of both subtrees rather than wholly one or the other, to make the tree differentiable. These have gone by names such as fuzzy, soft, neural, or differentiable decision trees; we use fuzzy decision trees (FDTs) as an umbrella term. In the case of ODTs, a splitting function such as sigmoid is used to map a linear combination of feature values to the weight of each subtree. [42] is the first to train globally optimized trees in this way. Later, [32] proposes ensembles of such trees, both as standalone models and as the final layer of a neural architecture, [33] presents various improvements such as balance and sparsity regularizations and dynamic learning rate adjustments, and [40] applies them to reinforcement learning.

Tree Alternating Optimization (TAO) is a recent and successful ODT learning algorithm that initializes similarly, but unlike FDTs, uses a linear discriminator at each node, such as logistic regression or linear SVM, then alternates optimization over each depth level of the tree. It can be used with a sparsity penalty to produce sparse ODTs, that is, where many or most of the coefficients are 0, making linear splits more interpretable. While it is shown that this alternating optimization never causes the loss to increase, there are no convergence or approximation guarantees. It was initially proposed for classification [10], then extended in various ways, including regression ensembles [45]; boosted ensembles [21]; clustering [20]; semi-supervised learning [46]; interpretable image classification [41]; interpretable natural language processing [28]; and as a tool for understanding parts of neural networks [27].

Our approach is different from any existing work in that we consider a broader scope of feature types, but more importantly, in that we continually and completely refit the decision tree throughout the learning process; this requires an efficient differentiable decision tree. For this, we rely on Kernel Density Decision Trees (KDDTs) [24], a recently proposed FDT that represents uncertainty through kernels rather than splitting functions and learns axis-aligned fuzzy splits using a generalization of CART. The kernel representation also has the benefit of avoiding redundant splitting that may occur in other FDT formalisms because repeating similar splits can make the division less "fuzzy" and therefore reduce loss by inflating the tree size. See the paper for background on axis-aligned FDTs. Our approach is also unique in that, for each output of the feature transformation, we weight the regularization by how much the feature is used for decisions in the training data, resulting in a more balanced enforcement of sparsity throughout the tree.

## 3 Methods

### 3.1 Alternating optimization trees and features

Given training data with features $\mathbf{X} \in \mathbb{R}^{n \times p}$ and regression targets or one-hot class label vectors $\mathbf{Y} \in \mathbb{R}^{n \times q}$, we aim to learn parameters $\theta \in \mathbb{R}^d$ of feature transformation $f_\theta : \mathbb{R}^p \to \mathbb{R}^{p'}$ and a decision tree $T : \mathbb{R}^{p'} \to \mathbb{R}^q$ to minimize a loss function $\mathcal{L}(\theta, T, \mathbf{X}, \mathbf{Y})$ for the composed predictive

model $T \circ f_\theta$. The tree is typically subject to constraints such as a max depth, minimum sample weight per node, or cost-complexity pruning.

Viewed within this framework, the prevailing approach in prior work is equivalent to the following process: (1) initialize $\theta$ such that $f_\theta$ is identity; (2) with $\theta$ fixed, optimize $T$ using a greedy tree growth strategy such as CART; (3) with $T$ fixed, optimize $\theta$ using gradient-based optimization. Some methods use a growing and/or pruning heuristic during (3) so that the structure of $T$ may change.

Instead, we propose an approach inspired by alternating optimization, a common strategy to pragmatically solve difficult optimization problems, whereby we alternate fitting a tree as in (2) with one or more gradient-based updates to $\theta$ as in (3). In this case, alternating allows us to use different optimization strategies for the feature transform parameters $\theta$ and the non-parametric decision tree $T$. At the end, we can use the fuzzy tree $T$, or replace its fuzzy splits with conventional crisp splits; the latter improves interpretability when predictions would otherwise take multiple paths, but can result in a small drop in performance.

## 3.2 Efficient differentiable trees

Conventional decision trees are piecewise-constant, so there is no meaningful loss gradient with respect to $\theta$. Instead, we use kernel density decision trees (KDDTs) [24], a type of fuzzy decision tree that interprets inputs with local uncertainty defined by a kernel function. In essence, they fit using a kernel density estimate computed from $\mathbf{X}$ rather than the raw values, and predictions are smoothed over the same kernel. This smoothing makes both the predictions and loss differentiable. KDDTs are fitted using an efficient generalization of CART, making them suitable for applications such as this where frequently repeated fitting makes efficiency crucial.

Here we present a simplified formulation of KDDTs to focus on this use case. A KDDT $T : \mathbb{R}^p \to \mathbb{R}^q$ is defined by underlying decision tree $T_0 : \mathbb{R}^p \to \mathbb{R}^q$ and product kernel $k : \mathbb{R}^p \to \mathbb{R}$ with $k(\mathbf{x}) = \prod_{i=1}^{p} F_i'(x_i)$, and $F_i$ the marginal CDF of the kernel along each dimension $i$. For the KDDT fitting algorithm, each $F_i'$ must be piecewise-constant, but this restriction does not apply for prediction; for example, one might use a histogram approximation of a Gaussian distribution for fitting and the Gaussian itself for prediction.

Then the KDDT is the output element-wise convolution $T^{(j)} \equiv T_0^{(j)} * k$. Suppose $T_0$ partitions $\mathbb{R}^p$ into hyper-interval leaves $L \subseteq \mathbb{R}^p$ with bounds $\ell_{L,i}$, $u_{L,i}$ (which may be $-\infty$ or $\infty$) on each dimension $i$. Each leaf has a value $\mathbf{v}_L \in \mathbb{R}^q$. Then the prediction is

$$T^{(j)}(\mathbf{x}) = (T_0^{(j)} * k)(\mathbf{x}) = \sum_L \mu_L(\mathbf{x}) v_L^{(j)} \tag{1}$$

where $\mu_L$ is the leaf membership function $\mu_L(\mathbf{x}) = \prod_{i \in [p]} F_i(x_i - \ell_{L,i}) - F_i(x_i - u_{L,i})$ and $\mathbf{v}_L \in \mathbb{R}^q$ is the leaf value $v_L^{(j)} = \frac{\sum_{i \in [n]} \mu_L(\mathbf{X}_i) Y_{i,j}}{\sum_{i \in [n]} \mu_L(\mathbf{X}_i)}$. The Jacobian is thus

$$\nabla_{i,j} T(\mathbf{x}) = \sum_L \nabla_i \mu_L(\mathbf{x}) v_L^{(j)}$$

$$\nabla_i \mu_L(\mathbf{x}) = (F_i'(x_i - \ell_{L,i}) - F_i'(x_i - u_{L,i})) \prod_{i' \neq i} F_i(x_i - \ell_{L,i}) - F_i(x_i - u_{L,i})$$

using the same traversal algorithm to compute (1) from [24]. Trees are fitted using an information gain based on an impurity measure, so the same impurity should be used as the loss function for feature learning. For classification, we use the Gini index; the resulting loss function, including the feature transformation $f$, is $\mathcal{L}(\theta, T, \mathbf{X}, \mathbf{Y}) = \frac{1}{n} \sum_L w_L (1 - \mathbf{v}_L^T \mathbf{v}_L)$ with $w_L = \sum_{i \in [n]} \mu_L(f_\theta(\mathbf{X}_i))$ the sample weight at leaf $L$. Then we compute the loss gradient $\nabla_\theta \mathcal{L}$ in terms of $\mu$, $\nabla \mu$, and $\nabla_\theta f$, using the sparsity of $\mu$ for efficient computation during tree traversal. For regression, we use variance of regression target values as the impurity, which can be expressed as a shifted version of the Gini loss, so the gradient computation is the same.

## 3.3 Kernel choice

In this context, the kernel is mainly used as a smoother to make the tree differentiable, so the choice of shape is not crucial. Good choices are a uniform (box) or Gaussian kernel. However, the *size* of

the kernel relative to the scale of the input is important; we adopt the term *bandwidth* from kernel density estimation to describe kernel scaling. If the bandwidth is too small, a KDDT is very close to a conventional decision tree, and the loss gradient is either zero or highly unstable because it results from very few data near split boundaries; if the bandwidth is too large, however, splits are very soft, the tree grows large, the discriminative power is weak, and computation is slow due to dense membership of data in leaves. To control the bandwidth, we use one of two strategies: (1) design the feature transform so that the output range is limited, for example, to $[0, 1]$, and choose an appropriate kernel bandwidth; or (2) use a regularization on the feature transform parameters to automatically scale the feature outputs appropriately relative to the kernel. We generally find the latter approach to be more effective and describe the use of regularizations in detail in Section 3.5.

Another related consideration is the *support* of the kernel, that is, the size of the range on which it has nonzero density. In general, the larger the support, the more decision paths that data belong to, increasing the cost of fitting, gradient computation, and prediction (if using a kernel during prediction). For example, a Gaussian kernel truncated at 3 standard deviations and another with the same variance but truncated at 6 standard deviations are nearly identical, but the latter is more computationally expensive to use with a KDDT.

## 3.4  Parameterized feature transforms

In principle, any differentiable parameterized class of feature transforms can be used, ranging from simple linear transforms as in ODTs to expressive classes of functions such as MLPs that have high discriminative power on their own. Generally, more expressive feature transforms result in smaller trees, but are probably less interpretable, depending on the application. The goal then is to choose expressive feature classes that remain interpretable in the desired context when conjoined into a rule list, as in the decision path of a tree.

In this work, we focus primarily on tabular data, to which tree-based models are most successfully applied, and suggest some general feature transformation primitives based on feature engineering techniques typically used with tree-based models. These can be composed, or their outputs concatenated so different kinds of transforms can be used together in the tree. We generally recommend starting with linear and prototype features, then proceeding with whatever improves performance while meeting interpretability requirements for the application.

Before describing the primitives, we note a general consideration for features used with decision trees: decision trees consider only the order of inputs and not the distance between values. While this is not strictly true for fuzzy decision trees such as KDDTs, it is best to avoid adding unnecessary feature complexity by using element-wise monotonic transformations of features, such as bias terms, monotonic activation functions, etc., on the final output of the feature transformation.

**Identity.** It may be beneficial to include the original, unmodified features along with transformed features so that the simplest possible rules are considered during construction of the tree. This also, in some sense, safeguards against situations where transformed features may actually be less informative than the original features.

**Element-wise transformation.** As noted above, element-wise monotonic transformations are largely meaningless on their own, but can be useful if composed with other transforms. For example, the composition of element-wise square with linear transformation allows for conic decision rules. Some element-wise mappings worth considering have no parameters, such as $x \mapsto \exp(x)$ and $x \mapsto \log(x)$; others have parameters to be learned, such as $x \mapsto x^a$. A notable option is the element-wise shifted power transformation

$$x \mapsto \begin{cases} \frac{\mathrm{sgn}(x+\alpha)|x+\alpha|^\lambda - 1}{\lambda(x+\alpha)^{\lambda-1}} & \text{if } \lambda \neq 0 \\ (x+\alpha)\mathrm{sgn}(x+\alpha)\log(x+\alpha) & \text{if } \lambda = 0 \end{cases}$$

which is used to alter distribution shape, typically to make values more normally distributed.

**Linear transformation.** The linear transformation is $\mathbf{x} \mapsto \mathbf{Ax} + \mathbf{b}$ with weights $\mathbf{A}$ and bias $\mathbf{b}$. The bias is used only if composing with another primitive. Using only a linear transformation results in an ODT. These can be initialized randomly, as identity, or using a linear dimension reduction strategy such as PCA.

| data | LR | MLP | DT | RF | ET | XGB | ours: linear | | ours: proto | |
| n, p, q | | | | | | | fuzzy | crisp | fuzzy | crisp |
|---|---|---|---|---|---|---|---|---|---|---|
| iris [18] | **0.960** | 0.953 | 0.947 | 0.947 | 0.953 | 0.947 | **0.960** | **0.960** | 0.947 | 0.927 |
| 150, 4 (4), 3 | - | - | 6.4 | 7.2e2 | 2.1e3 | 4.3e2 | 6.1 | 7.6 | 6.5 | 2.9 |
| heart-disease [30] | **0.822** | 0.792 | 0.707 | 0.802 | 0.795 | 0.792 | 0.812 | 0.812 | 0.793 | 0.779 |
| 303, 13 (20), 2 | - | - | 13.9 | 4.8e3 | 1.1e4 | 7.9e2 | 21.6 | 19.4 | 5.0 | 3.9 |
| dry-bean [31] | 0.925 | **0.934** | 0.912 | 0.923 | 0.921 | 0.928 | 0.920 | 0.913 | 0.915 | 0.901 |
| 13611, 16 (16), 7 | - | - | 99.8 | 6.7e4 | 2.0e5 | 1.3e4 | 1.1e2 | 45.8 | 1.6e2 | 94.2 |
| wine [1] | 0.983 | **0.989** | 0.904 | 0.977 | **0.989** | 0.955 | 0.983 | 0.983 | 0.961 | 0.961 |
| 178, 13 (13), 3 | - | - | 8.5 | 9.4e2 | 3.3e3 | 2.4e2 | 2.0 | 2.0 | 2.3 | 2.3 |
| car [5] | 0.926 | 0.992 | 0.977 | 0.964 | 0.971 | **0.994** | 0.991 | 0.992 | 0.980 | 0.979 |
| 1728, 6 (21), 4 | - | - | 95.3 | 2.3e4 | 3.1e4 | 4.5e3 | 29.0 | 29.0 | 59.5 | 55.9 |
| wdbc [44] | 0.974 | **0.975** | 0.935 | 0.965 | 0.970 | 0.968 | 0.972 | 0.972 | 0.961 | 0.961 |
| 569, 30 (30), 2 | - | - | 13.0 | 1.9e3 | 6.0e3 | 2.7e2 | 1.3 | 1.3 | 6.4 | 5.0 |
| sonar [38] | 0.755 | 0.879 | 0.735 | 0.826 | **0.880** | 0.855 | 0.818 | 0.799 | 0.798 | 0.817 |
| 208, 60 (60), 2 | - | - | 14.1 | 2.0e3 | 5.6e3 | 3.0e2 | 5.7 | 3.9 | 10.3 | 10.3 |
| pendigits [2] | 0.952 | **0.994** | 0.964 | 0.993 | **0.994** | 0.991 | 0.981 | 0.976 | 0.950 | 0.931 |
| 10992, 16 (16), 10 | - | - | 3.2e2 | 3.8e4 | 9.8e4 | 8.5e3 | 2.6e2 | 2.4e2 | 3.1e2 | 3.1e2 |
| ionosphere [39] | 0.875 | 0.917 | 0.892 | 0.934 | **0.943** | **0.943** | 0.932 | 0.920 | 0.920 | 0.909 |
| 351, 34 (34), 2 | - | - | 15.5 | 2.2e3 | 5.9e3 | 3.4e2 | 3.9 | 5.5 | 10.0 | 6.3 |

Table 1: Results of tabular data benchmarks. Number of attributes $p$ is listed before and after one-hot encoding categorical attributes. Each cell shows 10-fold cross-validation accuracy and average number of splits. The best accuracy for each data set is in bold.

**Distance to prototype.** Another feature type maps a point to its distance to each of $k$ prototypes $\mathbf{p}_i$; in this sense, a decision rule $d(\mathbf{x}, \mathbf{p}_i; \mathbf{S}_i) < t$ can be interpreted simply as "is $\mathbf{x}$ sufficiently similar to $\mathbf{p}_i$". We use Mahalanobis distance $d(\mathbf{x}, \mathbf{p}_i; \mathbf{S}_i) = (\mathbf{x} - \mathbf{p}_i)^T \mathbf{S}_i^{-1}(\mathbf{x} - \mathbf{p}_i)$, where the learnable parameters are prototypes $\mathbf{p}_i$ and inverse covariance matrices $\mathbf{S}_i^{-1}$. The covariance matrix can be reparameterized in various ways to restrict the measurement of distance[1], for example, by making it diagonal, or by using the same for all prototypes. It should also be parameterized such that it is positive semidefinite; otherwise hyperbolic decision rules may be learned and the distance interpretation no longer applies. $\mathbf{S}_i^{-1}$ can be regularized for sparsity, so that the distance is based on few features. The parameters can be initialized randomly or by using a mixture modeling algorithm.

**Fuzzy cluster membership.** Clustering partitions data into $k$ clusters with high internal similarity; usually each cluster $i$ is defined by a center $\mathbf{c}_i$, and each point belongs to the cluster with the closest center. Conventional clustering is not differentiable, so we instead use fuzzy clustering, a variant that assigns a degree of membership to each cluster, with the membership values summing to 1. Given distance functions $d_i$, $i \in [k]$, the membership in cluster $i$ is

$$ w_i(\mathbf{x}) = \frac{1}{\sum_{j=1}^{k} \left( \frac{d(\mathbf{x}, \mathbf{c}_i; \mathbf{S}_i)}{d(\mathbf{x}, \mathbf{c}_j; \mathbf{S}_j)} \right)^{\frac{2}{m-1}}} $$

with $m \in (1, \infty)$ a hyperparameter determining the "softness" of the cluster assignment, usually just set to 2, and $d$ the Mahalanobis distance as defined previously. In this way, soft clustering can be viewed as a transformation of the "distance to prototype" features so that the resulting transformed features are interdependent; a decision rule $w_i(\mathbf{x}) > t$ is interpreted as "$\mathbf{x}$ is sufficiently closer to $\mathbf{c}_i$ than the other centers". The parameters can be initialized randomly or by using a fuzzy clustering algorithm such as fuzzy $c$-means [15, 4] with all $\mathbf{S}_i$ initialized as identity.

### 3.5 Regularization

For each transformed feature $j \in [p']$, the parameters $\theta_j \subseteq \theta$ used to compute the $j$th feature $f^{(j)}$ may be subject to a regularization weighted by the total usage of feature $j$, that is, the sum of total training sample weights over nodes that split using feature $j$. This weighting serves to balance the effects of the loss and regularization; a feature used in the decisions for relatively few data will have

---

[1]For examples of different covariance types, see https://scikit-learn.org/stable/modules/mixture.html.

a relatively smaller loss gradient, and so should have a proportionally smaller regularization, and vice versa.

One reason to apply regularization is to improve the interpretability of features. The L1 regularization increases the sparsity of parameters, which can reduce the complexity of interpretation. Other regularizations might improve interpretability in an application-specific way. Weighting by feature usage also ensures that often-used features are the most interpretable.

Another reason to apply regularization is to automatically regulate kernel bandwidth. By using regularization that shrinks the output range of the features, such as a L1 or L2 regularization on a linear transformation, we effectively increase the bandwidth; meanwhile, minimizing the loss function shrinks the effective bandwidth because a smaller bandwidth allows fewer data to have split decision paths, resulting in purer leaves. This makes the actual choice of kernel bandwidth relative to regularization strength so that only one or the other must be tuned, automatically adapts effective bandwidth individually for each feature, and overall improves performance compared to using a fixed effective bandwidth.

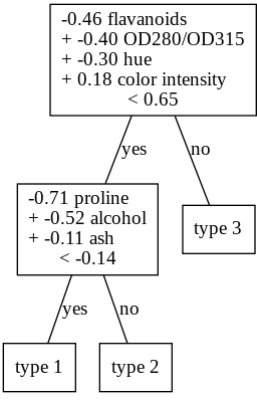

Figure 2: Tree with sparse linear features on the wine data set.

## 4 Evaluation

This section contains evaluation and demonstration of interpretable models. Comprehensive results, as well as additional experiment details, are in the supplementary material. Unless otherwise noted, all results are from crisp trees.

### 4.1 Benchmarks

We compare various configurations of our algorithm against popular tree-based baselines including decision trees, random forests, and ExtraTrees. We report 10-fold cross validation accuracy and average number of splits in the model. Additional metrics, including average decision path length, feature sparsity, and inference time are shown in the supplementary material. The data sets are selected from among the the most viewed tabular classification data sets on the UCI machine learning repository [14] at the time of writing. Categorical attributes are one-hot encoded, and the data is normalized to mean 0 and standard deviation 1. For our models, we show in the main paper results for linear features and distance-to-prototype features with diagonal inverse covariance. Each is regularized with L1 coefficient $\lambda_1 = .01$ to promote sparsity. Our models and the conventional decision trees have cost-complexity pruning $\alpha$ selected by cross-validation. Other hyperparameters are fixed and described in the supplementary material.

On every data set, at least one of our models matches or comes close to the best baseline accuracy while being much smaller. Also note that, since we choose the pruning parameter $\alpha$ by cross-validation, there are many cases where a smaller tree than

| feature | p1 | p2 | p3 |
|---|---|---|---|
| alcohol | 2.03 | -1.66 | -0.15 |
| malic acid | 0.85 | -0.35 | 1.15 |
| ash | 0.06 | -1.00 | -0.62 |
| alcalinity of ash | -1.16 | 0.11 | 0.22 |
| magnesium | 0.95 | -2.06 | -0.08 |
| total phenols | 1.27 | 0.70 | -1.46 |
| flavanoids | 0.04 | 2.69 | -0.54 |
| nonflav. phenols | -0.86 | -0.33 | 0.29 |
| proanthocynanins | 0.74 | -0.24 | 0.57 |
| color intensity | 0.70 | -1.95 | 2.51 |
| hue | 0.43 | 1.10 | -1.05 |
| OD280/OD315 | 0.51 | 0.50 | -1.31 |
| proline | 1.91 | -1.55 | -1.06 |

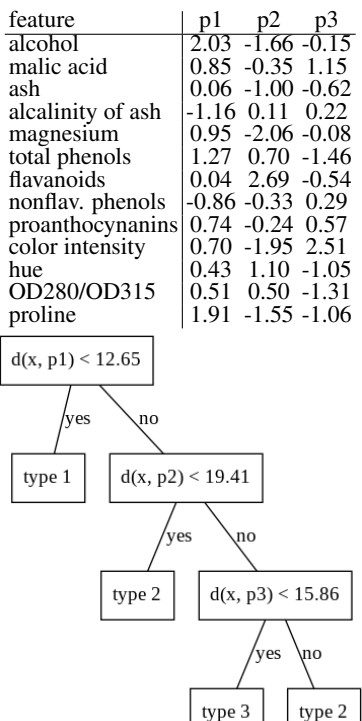

Figure 3: Tree with prototype features on the wine data set.

the one reported performs similarly. Especially for data sets where the ensembles greatly outperform the basic decision tree, the reduction in size for a performant model by using our method is huge. This carries the additional benefit that inference is much faster for our models, as shown in the supplementary material. Models with linear features are often the best for a given dataset, while prototype features lag slightly behind, suggesting that prototype features are best used to supplement linear features rather than on their own. We also note that these benchmarks all use the same hyperparameters; this shows that good performance does not require great tuning effort, but such effort will probably result in a better model for most applications.

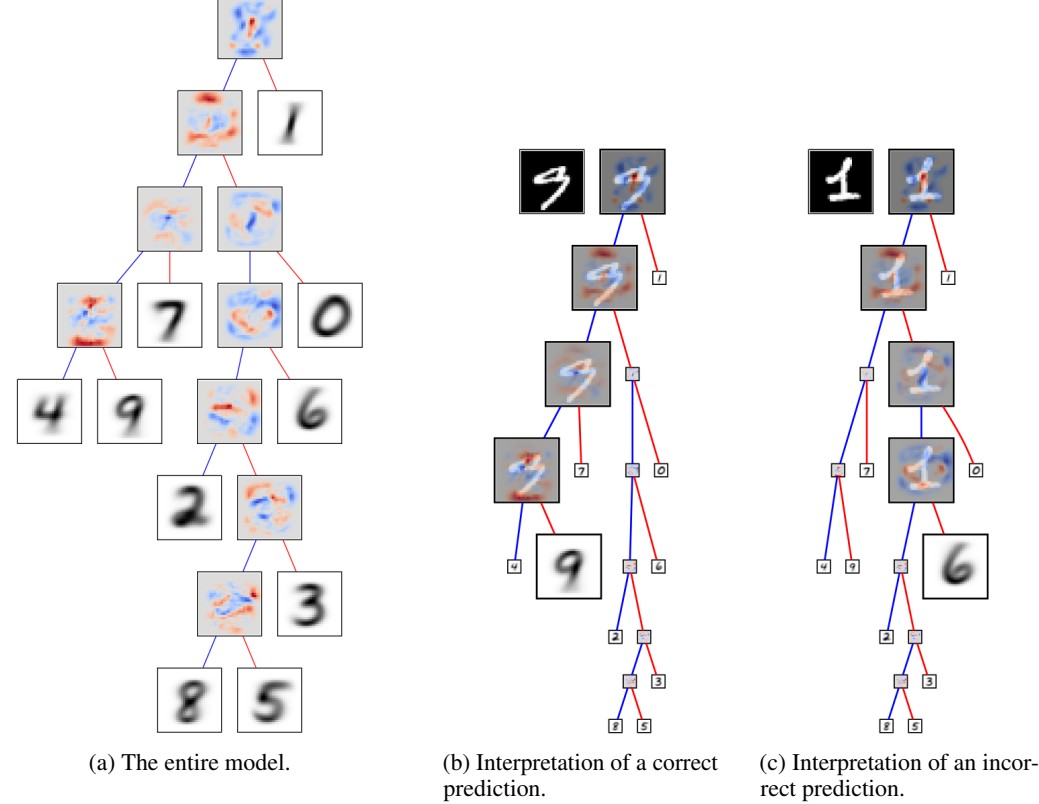

(a) The entire model.

(b) Interpretation of a correct prediction.

(c) Interpretation of an incorrect prediction.

Figure 4: An interpretable MNIST classifier with 92.19% test accuracy.

We also separately show results on MNIST and fashion-MNIST; these experiments are described in Section 4.3 and the results are shown in Table 2.

## 4.2 Example interpretation

We show two examples of interpretation for trees trained on the wine data set. Both are trained with $\alpha = 0.01$ as in the benchmarks and achieve 97.2% accuracy on a 20% test split.

A tree with linear features is shown in Figure 2. It has a stronger sparsity regularizer $\lambda_1 = 0.1$ compared to the benchmarks. Wines with low flavanoids, protein concentration, and hue are classified as type 3. There is also a smaller relationship between type 3 and high color intensity. Of the remaining wines, those with low proline and alcohol are type 2, and the rest are type 1. Similarly, there is also a smaller relationship between ash content and type 1 that may be important if the decision is not clear based on proline and alcohol.

A tree with prototype features is shown in Figure 3. Here we use simple Euclidean distance for decision rules. Each prototype defines a certain wine profile; for instance, prototype 1 is high in alcohol and proline and close to average in other attributes. Wines similar to prototype 1 are type 1; wines similar to prototype 2 are type 2; wines similar to prototype 3 are type 3; the rest are type 2.

## 4.3 MNIST features

Next we fit models to MNIST and fashion-MNIST to demonstrate performance on simple image classification. Here we use linear features with some extra

| MNIST | | | | |
|---|---|---|---|---|
| $\alpha$ | fuzzy acc. | crisp acc. | splits | path len. |
| $10^{-2}$ | .9226 | .9219 | 9 | 4.33 |
| $10^{-3}$ | .9500 | .9419 | 36 | 5.91 |
| $10^{-4}$ | .9664 | .9610 | 162 | 7.68 |
| $10^{-5}$ | .9708 | .9602 | 1659 | 10.05 |
| fashion-MNIST | | | | |
| $\alpha$ | fuzzy acc. | crisp acc. | splits | path len. |
| $10^{-2}$ | .8059 | .8024 | 8 | 3.97 |
| $10^{-3}$ | .8439 | .8364 | 20 | 4.69 |
| $10^{-4}$ | .8669 | .8586 | 98 | 6.24 |
| $10^{-5}$ | .8675 | .8472 | 2209 | 11.55 |

Table 2: Results for MNIST trees.

constraints to improve interpretability. First, in addition to a L1 regularization to promote sparsity, we use a smoothness regularization which penalizes the average squared difference of each weight with its neighbors, not including diagonal. The idea here is that a weight image with smooth shapes is easier to understand than one with a handful of sparse pixels, while also being more expressive. We also constrain the tree's threshold values to be zero to ease local interpretation. In this way, we can interpret a decision rule by asking "do the pen strokes better match the blue, or the red"? For example, in the root node in Figure 4a, we see that digits with ink primarily in a small near-vertical stroke in the center are classified as 1, whereas those with more ink in the closely surrounding space are other digits. Complete predictions are shown Figures 4b and 4c.

The performance and size of models for various cost-complexity pruning $\alpha$ are shown in Table 2, and the smallest MNIST models are shown in Figure 4. Larger models are shown in the supplementary material. Each internal node shows the linear feature's weight image, with negative values in blue and positive in red. Each leaf shows the average of the training data belonging to it.

Testing on MNIST allows us to make some comparison of our models to TAO [10], which seems to be the best existing method for learning trees with sparse linear features. Ultimately, while their smallest model achieves 89.81% test accuracy with 16 splits, our smallest achieves 92.19% with just 9 splits. Likewise, their overall best-performing model achieves 94.31%, whereas ours achieves 96.10% without fuzzy splitting or 97.08% with it. The latter is on par with performance of random forests for MNIST, accomplished with a single (albeit large) tree with interpretable features. This shows that our proposed approach can achieve state-of-the-art status for learning sparse oblique decision trees, while also having greater flexibility to choose different features and adapt them to application-specific interpretability needs.

### 4.4 Information hierarchy

An additional property of our models is that, differently from other ODTs such as TAO, we continually refit the tree using CART, which greedily chooses the most informative splits first. Each subtree's class labels are thus as pure as possible, and classes that are easy to separate are separated first. By contrast, in a tree which is trained globally without this greedy splitting, any particular split may not actually result in improvement in label purity, as long as it is informative to splits further down the tree.

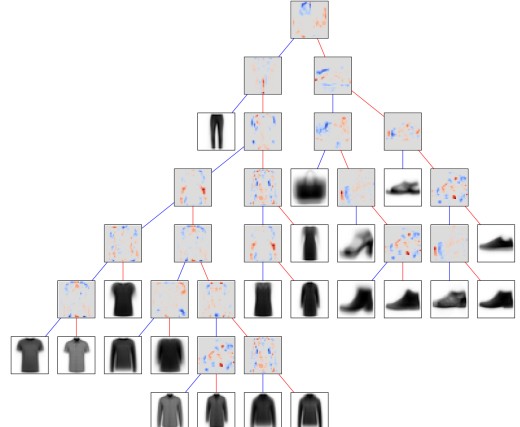

Figure 5: An interpretable fashion-MNIST classifier. Because CART selects the most informative splits first, the tree is a visually intuitive hierarchy. For example, tops, bottoms, shoes, and bags each have a distinct subtree; within tops, there are subtrees for sleeve length, collar shape, etc.

A compelling example is shown in Figure 5, which shows the tree trained on fashion-MNIST with $\alpha = 10^{-3}$. Without exception, the tree forms a hierarchy of visual similarity of fashion items, as described in the caption. This allows us to interpret predictions to some extent *without even knowing the features*. For example, in Figure 5, if the decision path for an image of a tee shirt ends at the upper leftmost leaf (dress), instead of the one below it (tee shirt), we can tell by the leaf data alone that the model correctly thinks it is a top without long sleeves, but mistakenly thinks that it is sleeveless instead of short-sleeved. A similar structure of visually cohesive subtrees is observed in our MNIST trees, as in Figure 4, but the trend is less obvious because digits are more visually distinct in general compared to the items that comprise fashion-MNIST. We do not observe this property in other ODTs; for example, in [41], a TAO tree fitted to fashion-MNIST does not exhibit this visual hierarchy property.

## 5 Discussion

We have proposed a new system for learning small, interpretable decision trees and showed examples under several notions of interpretability, including sparse linear combinations, similarity to prototypes, visual matching, and class-based hierarchies. Compared to ensembles and deep models, our single-

tree models are interpretable because they are small, use sparse features, have a sparsely activated hierarchical representation based on logical rules, and ultimately work by simply partitioning the data. Moreover, compared to other methods for single-tree models, ours are highly flexible to customize feature types to meet interpretability needs and produce smaller models with more meaningful hierarchical structure and better performance. The tradeoff is that the models can take relatively long to train due to repeated fitting of fuzzy trees, and while our results show that good models can be obtained without extensive hyperparameter tuning, the absolute best model for a given scenario certainly may require careful tuning of several hyperparameters.

Future work will incorporate new, more specialized feature types to extend the framework to differently structured data such as images (more challenging than MNIST), time series, and text. We will also work to improve the framework for regression tasks, which are not yet analyzed in this work because the piecewise-constant nature of decision trees means that small trees with simple feature transformations are less than ideally suited for challenging regression tasks.

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
