# Feature Learning for Interpretable, Performant Decision Trees
## Supplementary Material

## 1 Experiment Specification

Here we cover the full specification of the experiments. Some details were omitted from the main text.

### 1.1 Benchmarking

Our experiments compare decision trees, random forests, ExtraTrees, and our proposed models with linear and distance-to-prototype features. All reported statistics are the average of 10-fold cross-validation, and in this document, we also report standard deviation.

All data sets were retrieved from the UCI Machine Learning Repository [1]. If there were separate training and test sets, they were combined before creating the random 10-fold split. Categorical attributes are one-hot encoded. All attributes are normalized to mean 0 and standard deviation 1. This makes feature learning more consistent, and it makes interpretation unitless.

For baseline models, we used implementations from scikit-learn [5], and our methods are implemented using Python along with PyTorch [4] for automatic differentiation. The splitting criterion for all models is Gini Index. Additional details for each model type follow.

- Decision tree. Using 10-fold cross validation (on the union of the 9 training folds for the experimental layer of cross-validation), we select the cost-complexity pruning $\alpha$ that results in the best accuracy. Our candidate $\alpha$ values include 0 and 15 evenly spaced values on the log scale from $10^{-8}$ to $10^{-1}$, that is, $\{0, 1 \times 10^{-8}, 3.16 \times 10^{-8}, 1 \times 10^{-7}, 3.16 \times 10^{-7}, \ldots, 1 \times 10^{-1}\}$.

- Random forest. We use default settings from scikit-learn. The ensemble contains 100 trees, each trained on a bootstrap sample with the same size as the original data, and the features consider for each split are limited to a uniformly sampled subset with size equal to the square root of the total number of features. There is no pruning.

- ExtraTrees. We use default settings from scikit-learn. The ensemble contains 100 trees, the features consider for each split are limited to a uniformly sampled subset with size equal to the square root of the total number of features, and for each feature, one candidate threshold is sampled uniformly in the range of data belonging to the current subtree. There is no pruning.

- Ours. For the KDDT, we use a box kernel with radius 0.1, that is, $k(\mathbf{x}) \propto \prod_{i=1}^{p} \mathbf{1}\{|x_i| \leq 0.1\}$. Do not perform a split if it would result in a leaf with total sample weight less than 1. We select cost-complexity pruning $\alpha$ from $[.0001, .0003, .001, .003, .01, .03, .1]$ by 10-fold cross-validation of accuracy. We implement feature learning using PyTorch [4] for automatic differentiation and stochastic optimization. The optimizer is Adam [3] with learning rate 0.01, and the optimization procedure uses minibatch gradient descent with batch size 1024. Training runs for 10 epochs fitting the tree before each minibatch, then 1000 epochs fitting the tree once every 10 epochs. For the dry-bean and pendigits data sets, we instead train for 10 then 100 epochs because of their much larger size. This process allows the tree structure changes to be responsive early in training when the features are changing rapidly, then saves time by fitting less frequently as the features converge. We apply either L1 or L2 regularization to the feature parameters, each with coefficient 0.01. The details for each feature type follow.

- Linear features. We use a linear transformation without bias, with the same number of outputs as inputs. We initialize either as identity or uniformly at random in the range $\pm\sqrt{6/p}$, where $p$ is the number of inputs to the linear transformation. Results in the main paper use random initialization.

- Distance-to-prototype features. We use a number of prototypes equal to the number of attributes in the data. Initialization is either random, with prototypes being samples from the unit Gaussian and the inverse covariance being identity, or by using the centers and inverse covariance matrices from a fitted Gaussian Mixture model from scikit-learn, with matching constraints on the covariance. Results in the main paper use random initialization. We constrain that inverse covariance matrices be positive definite so that the features represent distance. For these experiments, we also constrain inverse covariance matrices to be diagonal for the sake of easier interpretation, that is, so that it can be interpreted as Euclidean distance with each input being scaled differently. Regularization is applied only to the covariance parameters, although L1 regularization could also be applied to the prototypes themselves to make them sparse in the sense that they only sparsely differ from the global average. This may be useful for data with many attributes, so that prototypes can be described by just a few features.

## 1.2 MNIST Training

The KDDTs for the MNIST and Fashion-MNIST models also use a box kernel with radius 1. We used the Adam optimizer [3] with learning rate 0.001, and minibatch gradient descent with batch size 1024. For 10 epochs, the tree is fitted once per batch to the batch itself; the large batch size is chosen to ensure that the tree fitted to each batch is representative enough of the tree for the entire training data, while being faster than fitting to the entire MNIST training set at each batch. We then train for 100 epochs with the tree fitted to the entire training set once per epoch. The feature transformation is a linear mapping from $28^2 = 784$ to 784 features. For smaller models, a smaller number of features would be fine and would speed up training. We use L1 regularization with coefficient 0.3 and a weight image smoothing regularizer with coefficient 0.03 for MNIST, and respectively 3 and 0.01 for Fashion-MNIST. The smoothing penalty itself is calculated as the average squared difference of each pixel with its neighbors (not including diagonal).

# 2 Complete Results

In the complete experiment results, we show the mean and standard deviation from 10-fold cross-validation with five evaluation metrics:

1. validation accuracy

2. total number of nodes

3. average length of a validation decision path

4. average gini coefficient (a good measure of sparsity [2], lower is sparser) of the parameters for each feature, weighted by each feature's usage on the validation fold

5. total time for inference on the validation fold, in milliseconds

We make some observations based on the additional results:

- Random initialization more often results in better-performing models.

- Even when one of our models has many nodes, which may make global interpretation difficult, the average path length grows much less, so local interpretation is still simple.

- L1 regularization does result in sparser features compared to L2 regularization. With L2 regularization, features are usually very dense, with Gini index near 1.

- Our models have faster inference time than similarly performing ensembles.

| data | metric | Baselines | | | | | |
| | | LR | MLP | DT | RF | ET | XGB |
| $n, p, q$ | | | | | | | |
| iris | acc | $.960 \pm .044$ | $.953 \pm .052$ | $.947 \pm .058$ | $.947 \pm .065$ | $.953 \pm .052$ | $.947 \pm .058$ |
| 150, 4 (4), 3 | nodes | $0.0 \pm 0.0$ | $0.0 \pm 0.0$ | $6.4 \pm 2.2$ | $720.4 \pm 50.7$ | $2057.3 \pm 109.7$ | $432.6 \pm 45.0$ |
| | path len | $0.00 \pm 0.00$ | $0.00 \pm 0.00$ | $2.48 \pm 0.38$ | $261.74 \pm 17.31$ | $449.09 \pm 37.55$ | $0.00 \pm 0.00$ |
| | gini | $.000 \pm .000$ | $.000 \pm .000$ | $.000 \pm .000$ | $.000 \pm .000$ | $.000 \pm .000$ | $.000 \pm .000$ |
| | time | $0.7 \pm 0.2$ | $9.7 \pm 26.6$ | $0.6 \pm 0.0$ | $128.4 \pm 45.5$ | $34.2 \pm 18.8$ | $78.7 \pm 63.7$ |
| heart-disease | acc | $.822 \pm .021$ | $.792 \pm .067$ | $.707 \pm .060$ | $.802 \pm .065$ | $.795 \pm .052$ | $.792 \pm .038$ |
| 303, 13 (20), 2 | nodes | $0.0 \pm 0.0$ | $0.0 \pm 0.0$ | $13.9 \pm 16.1$ | $4827.2 \pm 108.1$ | $10648.6 \pm 206.0$ | $788.6 \pm 20.1$ |
| | path len | $0.00 \pm 0.00$ | $0.00 \pm 0.00$ | $3.03 \pm 1.71$ | $584.43 \pm 15.94$ | $771.86 \pm 30.70$ | $0.00 \pm 0.00$ |
| | gini | $.000 \pm .000$ | $.000 \pm .000$ | $.000 \pm .000$ | $.000 \pm .000$ | $.000 \pm .000$ | $.000 \pm .000$ |
| | time | $0.4 \pm 0.2$ | $0.4 \pm 0.0$ | $0.3 \pm 0.0$ | $6.4 \pm 0.1$ | $6.6 \pm 0.1$ | $3.3 \pm 0.7$ |
| dry-bean | acc | $.925 \pm .007$ | $.934 \pm .005$ | $.912 \pm .008$ | $.923 \pm .006$ | $.921 \pm .007$ | $.928 \pm .006$ |
| 13611, 16 (16), 7 | nodes | $0.0 \pm 0.0$ | $0.0 \pm 0.0$ | $99.8 \pm 3.8$ | $66504.9 \pm 530.6$ | $197338.7 \pm 1264.0$ | $12907.8 \pm 166.2$ |
| | path len | $0.00 \pm 0.00$ | $0.00 \pm 0.00$ | $7.05 \pm 0.14$ | $1142.00 \pm 7.96$ | $1287.98 \pm 12.75$ | $0.00 \pm 0.00$ |
| | gini | $.000 \pm .000$ | $.000 \pm .000$ | $.000 \pm .000$ | $.000 \pm .000$ | $.000 \pm .000$ | $.000 \pm .000$ |
| | time | $0.9 \pm 0.2$ | $1.4 \pm 0.0$ | $0.7 \pm 0.0$ | $38.3 \pm 0.3$ | $49.2 \pm 0.3$ | $4.3 \pm 0.2$ |
| wine | acc | $.983 \pm .025$ | $.989 \pm .022$ | $.904 \pm .077$ | $.977 \pm .028$ | $.989 \pm .022$ | $.955 \pm .043$ |
| 178, 13 (13), 3 | nodes | $0.0 \pm 0.0$ | $0.0 \pm 0.0$ | $8.5 \pm 2.1$ | $936.4 \pm 19.5$ | $3315.1 \pm 43.1$ | $242.1 \pm 15.0$ |
| | path len | $0.00 \pm 0.00$ | $0.00 \pm 0.00$ | $3.30 \pm 0.40$ | $332.74 \pm 6.59$ | $579.64 \pm 16.11$ | $0.00 \pm 0.00$ |
| | gini | $.000 \pm .000$ | $.000 \pm .000$ | $.000 \pm .000$ | $.000 \pm .000$ | $.000 \pm .000$ | $.000 \pm .000$ |
| | time | $0.6 \pm 0.2$ | $0.4 \pm 0.0$ | $0.3 \pm 0.0$ | $6.2 \pm 0.2$ | $6.2 \pm 0.0$ | $1.7 \pm 0.2$ |
| car | acc | $.926 \pm .021$ | $.992 \pm .007$ | $.977 \pm .012$ | $.964 \pm .013$ | $.971 \pm .011$ | $.994 \pm .006$ |
| 1728, 6 (21), 4 | nodes | $0.0 \pm 0.0$ | $0.0 \pm 0.0$ | $95.3 \pm 6.6$ | $23031.0 \pm 243.0$ | $31240.4 \pm 330.8$ | $4478.2 \pm 48.9$ |
| | path len | $0.00 \pm 0.00$ | $0.00 \pm 0.00$ | $4.48 \pm 0.27$ | $610.58 \pm 21.80$ | $617.75 \pm 23.81$ | $0.00 \pm 0.00$ |
| | gini | $.000 \pm .000$ | $.000 \pm .000$ | $.000 \pm .000$ | $.000 \pm .000$ | $.000 \pm .000$ | $.000 \pm .000$ |
| | time | $0.8 \pm 0.1$ | $1.1 \pm 0.0$ | $0.7 \pm 0.1$ | $16.4 \pm 0.3$ | $17.0 \pm 0.5$ | $3.7 \pm 0.3$ |
| wdbc | acc | $.974 \pm .021$ | $.975 \pm .024$ | $.935 \pm .032$ | $.965 \pm .019$ | $.970 \pm .028$ | $.968 \pm .021$ |
| 569, 30 (30), 2 | nodes | $0.0 \pm 0.0$ | $0.0 \pm 0.0$ | $13.0 \pm 6.5$ | $1881.4 \pm 59.9$ | $6045.5 \pm 193.9$ | $274.4 \pm 8.2$ |
| | path len | $0.00 \pm 0.00$ | $0.00 \pm 0.00$ | $3.96 \pm 1.25$ | $462.02 \pm 12.92$ | $667.99 \pm 21.11$ | $0.00 \pm 0.00$ |
| | gini | $.000 \pm .000$ | $.000 \pm .000$ | $.000 \pm .000$ | $.000 \pm .000$ | $.000 \pm .000$ | $.000 \pm .000$ |
| | time | $0.8 \pm 0.1$ | $0.8 \pm 0.0$ | $0.6 \pm 0.0$ | $7.7 \pm 2.1$ | $6.6 \pm 0.1$ | $2.0 \pm 0.4$ |
| sonar | acc | $.755 \pm .094$ | $.879 \pm .064$ | $.735 \pm .096$ | $.826 \pm .104$ | $.880 \pm .058$ | $.855 \pm .063$ |
| 208, 60 (60), 2 | nodes | $0.0 \pm 0.0$ | $0.0 \pm 0.0$ | $14.1 \pm 6.7$ | $2022.6 \pm 16.4$ | $5586.1 \pm 77.9$ | $301.1 \pm 6.0$ |
| | path len | $0.00 \pm 0.00$ | $0.00 \pm 0.00$ | $3.86 \pm 1.46$ | $490.86 \pm 10.09$ | $708.90 \pm 25.75$ | $0.00 \pm 0.00$ |
| | gini | $.000 \pm .000$ | $.000 \pm .000$ | $.000 \pm .000$ | $.000 \pm .000$ | $.000 \pm .000$ | $.000 \pm .000$ |
| | time | $0.4 \pm 0.2$ | $0.4 \pm 0.0$ | $0.3 \pm 0.0$ | $6.2 \pm 0.2$ | $6.4 \pm 0.1$ | $1.8 \pm 0.2$ |
| pendigits | acc | $.952 \pm .005$ | $.994 \pm .003$ | $.964 \pm .004$ | $.993 \pm .002$ | $.994 \pm .002$ | $.991 \pm .002$ |
| 10992, 16 (16), 10 | nodes | $0.0 \pm 0.0$ | $0.0 \pm 0.0$ | $322.0 \pm 13.4$ | $38475.5 \pm 232.0$ | $98345.3 \pm 655.6$ | $8464.5 \pm 60.7$ |
| | path len | $0.00 \pm 0.00$ | $0.00 \pm 0.00$ | $10.13 \pm 0.22$ | $974.59 \pm 5.43$ | $1142.83 \pm 7.26$ | $0.00 \pm 0.00$ |
| | gini | $.000 \pm .000$ | $.000 \pm .000$ | $.000 \pm .000$ | $.000 \pm .000$ | $.000 \pm .000$ | $.000 \pm .000$ |
| | time | $29.8 \pm 37.4$ | $36.1 \pm 63.6$ | $9.3 \pm 13.0$ | $271.2 \pm 80.6$ | $43.2 \pm 0.8$ | $4.6 \pm 0.7$ |
| ionosphere | acc | $.875 \pm .069$ | $.917 \pm .062$ | $.892 \pm .048$ | $.934 \pm .056$ | $.943 \pm .049$ | $.943 \pm .053$ |
| 351, 34 (34), 2 | nodes | $0.0 \pm 0.0$ | $0.0 \pm 0.0$ | $15.5 \pm 9.0$ | $2205.7 \pm 88.3$ | $5919.3 \pm 267.2$ | $335.4 \pm 20.4$ |
| | path len | $0.00 \pm 0.00$ | $0.00 \pm 0.00$ | $5.17 \pm 2.43$ | $645.54 \pm 39.76$ | $889.93 \pm 48.63$ | $0.00 \pm 0.00$ |
| | gini | $.000 \pm .000$ | $.000 \pm .000$ | $.000 \pm .000$ | $.000 \pm .000$ | $.000 \pm .000$ | $.000 \pm .000$ |
| | time | $0.4 \pm 0.2$ | $0.4 \pm 0.0$ | $0.3 \pm 0.0$ | $6.4 \pm 0.3$ | $6.5 \pm 0.1$ | $1.8 \pm 0.2$ |

Ours: linear features, L2 regularization, non-random initialization, fuzzy

| data | metric | 1e-1 | 3e-2 | 1e-2 | 3e-3 | 1e-3 | 3e-4 | 1e-4 |
|---|---|---|---|---|---|---|---|---|
| iris | acc | $.960 \pm .044$ | $.960 \pm .044$ | $.953 \pm .043$ | $.953 \pm .052$ | $.967 \pm .033$ | $.960 \pm .044$ | $.967 \pm .033$ |
| | nodes | $2.0 \pm 0.0$ | $2.0 \pm 0.0$ | $3.8 \pm 0.7$ | $5.0 \pm 0.9$ | $8.7 \pm 1.7$ | $12.6 \pm 1.5$ | $16.1 \pm 1.6$ |
| | path len | $1.67 \pm 0.09$ | $1.67 \pm 0.09$ | $2.28 \pm 0.31$ | $2.41 \pm 0.22$ | $2.82 \pm 0.28$ | $3.16 \pm 0.40$ | $3.66 \pm 0.41$ |
| | gini | $.618 \pm .010$ | $.618 \pm .014$ | $.458 \pm .081$ | $.500 \pm .059$ | $.548 \pm .071$ | $.565 \pm .052$ | $.561 \pm .061$ |
| | time | $1.2 \pm 0.0$ | $4.3 \pm 9.5$ | $3.7 \pm 7.6$ | $1.5 \pm 0.2$ | $2.4 \pm 0.4$ | $3.3 \pm 0.3$ | $4.1 \pm 0.4$ |
| heart | acc | $.799 \pm .053$ | $.795 \pm .028$ | $.776 \pm .043$ | $.766 \pm .051$ | $.782 \pm .058$ | $.743 \pm .041$ | $.762 \pm .058$ |
| | nodes | $1.0 \pm 0.0$ | $1.7 \pm 0.5$ | $5.7 \pm 1.7$ | $19.5 \pm 3.3$ | $21.7 \pm 1.7$ | $26.8 \pm 2.6$ | $27.1 \pm 3.6$ |
| | path len | $1.00 \pm 0.00$ | $1.40 \pm 0.28$ | $2.67 \pm 0.62$ | $4.15 \pm 0.51$ | $4.84 \pm 0.58$ | $5.45 \pm 0.63$ | $6.00 \pm 0.98$ |
| | gini | $.888 \pm .010$ | $.881 \pm .025$ | $.862 \pm .014$ | $.824 \pm .045$ | $.830 \pm .034$ | $.814 \pm .041$ | $.836 \pm .032$ |
| | time | $0.7 \pm 0.0$ | $0.9 \pm 0.2$ | $2.1 \pm 0.5$ | $3.9 \pm 0.6$ | $4.3 \pm 0.3$ | $5.4 \pm 0.5$ | $5.5 \pm 0.7$ |
| dry-bean | acc | $.792 \pm .005$ | $.913 \pm .009$ | $.919 \pm .006$ | $.918 \pm .004$ | $.920 \pm .006$ | $.925 \pm .006$ | $.925 \pm .005$ |
| | nodes | $4.0 \pm 0.0$ | $6.0 \pm 0.0$ | $8.0 \pm 0.0$ | $9.9 \pm 0.7$ | $18.2 \pm 1.0$ | $41.2 \pm 3.2$ | $94.3 \pm 3.5$ |
| | path len | $2.83 \pm 0.03$ | $3.22 \pm 0.05$ | $3.40 \pm 0.08$ | $3.61 \pm 0.16$ | $4.72 \pm 0.13$ | $6.13 \pm 0.32$ | $7.48 \pm 0.33$ |
| | gini | $.902 \pm .002$ | $.902 \pm .003$ | $.910 \pm .002$ | $.909 \pm .003$ | $.909 \pm .002$ | $.906 \pm .003$ | $.908 \pm .002$ |
| | time | $3.3 \pm 0.1$ | $4.2 \pm 0.1$ | $5.4 \pm 0.1$ | $6.2 \pm 0.3$ | $10.2 \pm 0.4$ | $15.4 \pm 1.2$ | $31.1 \pm 0.9$ |
| wine | acc | $.961 \pm .026$ | $.972 \pm .028$ | $.972 \pm .028$ | $.972 \pm .028$ | $.966 \pm .027$ | $.977 \pm .028$ | $.972 \pm .028$ |
| | nodes | $2.0 \pm 0.0$ | $2.0 \pm 0.0$ | $2.8 \pm 0.4$ | $2.7 \pm 0.5$ | $2.9 \pm 0.3$ | $2.8 \pm 0.4$ | $2.7 \pm 0.5$ |
| | path len | $1.67 \pm 0.09$ | $1.67 \pm 0.09$ | $1.92 \pm 0.15$ | $1.90 \pm 0.16$ | $1.95 \pm 0.15$ | $1.96 \pm 0.18$ | $1.89 \pm 0.18$ |
| | gini | $.871 \pm .011$ | $.875 \pm .009$ | $.884 \pm .011$ | $.885 \pm .007$ | $.891 \pm .005$ | $.886 \pm .009$ | $.885 \pm .009$ |
| | time | $0.6 \pm 0.0$ | $0.6 \pm 0.0$ | $0.7 \pm 0.1$ | $0.7 \pm 0.1$ | $0.7 \pm 0.0$ | $0.7 \pm 0.1$ | $0.7 \pm 0.1$ |
| car | acc | $.700 \pm .044$ | $.914 \pm .034$ | $.965 \pm .013$ | $.986 \pm .009$ | $.992 \pm .012$ | $.992 \pm .009$ | $.992 \pm .009$ |
| | nodes | $0.0 \pm 0.0$ | $2.8 \pm 0.4$ | $5.0 \pm 0.8$ | $12.3 \pm 1.6$ | $18.8 \pm 2.7$ | $23.7 \pm 2.1$ | $24.5 \pm 4.1$ |
| | path len | $0.00 \pm 0.00$ | $1.60 \pm 0.16$ | $1.90 \pm 0.18$ | $2.86 \pm 0.34$ | $3.23 \pm 0.52$ | $3.61 \pm 0.29$ | $3.70 \pm 0.63$ |
| | gini | $.000 \pm .000$ | $.905 \pm .010$ | $.897 \pm .015$ | $.902 \pm .015$ | $.902 \pm .014$ | $.903 \pm .007$ | $.901 \pm .010$ |
| | time | $0.2 \pm 0.0$ | $1.2 \pm 0.1$ | $1.7 \pm 0.2$ | $3.6 \pm 0.4$ | $5.3 \pm 0.7$ | $6.6 \pm 0.7$ | $6.8 \pm 1.2$ |
| wdbc | acc | $.965 \pm .025$ | $.967 \pm .023$ | $.960 \pm .027$ | $.963 \pm .021$ | $.968 \pm .025$ | $.960 \pm .024$ | $.963 \pm .024$ |
| | nodes | $1.0 \pm 0.0$ | $1.0 \pm 0.0$ | $1.0 \pm 0.0$ | $2.1 \pm 0.9$ | $5.9 \pm 2.1$ | $13.9 \pm 4.2$ | $19.6 \pm 3.6$ |
| | path len | $1.00 \pm 0.00$ | $1.00 \pm 0.00$ | $1.00 \pm 0.00$ | $1.59 \pm 0.48$ | $2.51 \pm 0.31$ | $3.76 \pm 0.68$ | $4.54 \pm 0.39$ |
| | gini | $.954 \pm .003$ | $.955 \pm .003$ | $.954 \pm .003$ | $.951 \pm .003$ | $.951 \pm .003$ | $.950 \pm .003$ | $.951 \pm .004$ |
| | time | $0.5 \pm 0.0$ | $0.5 \pm 0.0$ | $0.5 \pm 0.0$ | $0.9 \pm 0.3$ | $1.8 \pm 0.5$ | $3.8 \pm 1.0$ | $5.3 \pm 1.0$ |
| sonar | acc | $.765 \pm .093$ | $.802 \pm .043$ | $.798 \pm .080$ | $.817 \pm .088$ | $.779 \pm .074$ | $.856 \pm .038$ | $.807 \pm .063$ |
| | nodes | $1.0 \pm 0.0$ | $2.2 \pm 0.6$ | $4.5 \pm 0.8$ | $5.6 \pm 0.9$ | $6.8 \pm 3.0$ | $7.6 \pm 2.1$ | $8.0 \pm 2.4$ |
| | path len | $1.00 \pm 0.00$ | $1.60 \pm 0.24$ | $2.53 \pm 0.31$ | $2.84 \pm 0.50$ | $3.34 \pm 0.93$ | $3.53 \pm 0.70$ | $3.54 \pm 0.77$ |
| | gini | $.971 \pm .002$ | $.972 \pm .001$ | $.973 \pm .001$ | $.950 \pm .073$ | $.973 \pm .002$ | $.973 \pm .001$ | $.973 \pm .002$ |
| | time | $0.5 \pm 0.0$ | $0.9 \pm 0.2$ | $1.5 \pm 0.2$ | $1.7 \pm 0.2$ | $2.0 \pm 0.7$ | $2.2 \pm 0.5$ | $2.3 \pm 0.6$ |
| pendigits | acc | $.094 \pm .003$ | $.911 \pm .022$ | $.955 \pm .011$ | $.974 \pm .006$ | $.986 \pm .003$ | $.989 \pm .003$ | $.991 \pm .002$ |
| | nodes | $0.0 \pm 0.0$ | $9.3 \pm 0.5$ | $12.1 \pm 1.0$ | $17.9 \pm 1.7$ | $27.4 \pm 3.0$ | $57.5 \pm 6.6$ | $125.9 \pm 9.5$ |
| | path len | $0.00 \pm 0.00$ | $4.18 \pm 0.32$ | $4.46 \pm 0.31$ | $5.18 \pm 0.23$ | $5.64 \pm 0.45$ | $6.58 \pm 0.42$ | $8.19 \pm 0.60$ |
| | gini | $.000 \pm .000$ | $.891 \pm .008$ | $.892 \pm .005$ | $.891 \pm .006$ | $.896 \pm .006$ | $.899 \pm .002$ | $.897 \pm .005$ |
| | time | $0.3 \pm 0.0$ | $4.5 \pm 0.1$ | $5.5 \pm 0.5$ | $7.7 \pm 0.5$ | $10.7 \pm 1.0$ | $19.6 \pm 1.9$ | $37.9 \pm 2.6$ |
| ionosphere | acc | $.857 \pm .057$ | $.932 \pm .060$ | $.932 \pm .048$ | $.935 \pm .034$ | $.934 \pm .050$ | $.929 \pm .048$ | $.946 \pm .038$ |
| | nodes | $1.0 \pm 0.0$ | $2.1 \pm 0.3$ | $3.8 \pm 0.4$ | $4.8 \pm 0.7$ | $8.5 \pm 1.9$ | $9.5 \pm 1.8$ | $11.8 \pm 2.0$ |
| | path len | $1.00 \pm 0.00$ | $1.77 \pm 0.13$ | $2.91 \pm 0.24$ | $3.52 \pm 0.57$ | $4.49 \pm 0.51$ | $4.70 \pm 0.85$ | $5.39 \pm 0.79$ |
| | gini | $.948 \pm .003$ | $.929 \pm .012$ | $.925 \pm .005$ | $.926 \pm .005$ | $.939 \pm .005$ | $.934 \pm .007$ | $.931 \pm .004$ |
| | time | $0.7 \pm 0.1$ | $1.1 \pm 0.1$ | $1.7 \pm 0.1$ | $2.0 \pm 0.3$ | $3.4 \pm 0.7$ | $2.4 \pm 0.4$ | $3.0 \pm 0.5$ |

Ours: linear features, L2 regularization, non-random initialization, crisp

| data | metric | 1e-1 | 3e-2 | 1e-2 | 3e-3 | 1e-3 | 3e-4 | 1e-4 |
|------|--------|------|------|------|------|------|------|------|
| iris | acc | $.960 \pm .044$ | $.960 \pm .044$ | $.953 \pm .043$ | $.967 \pm .033$ | $.960 \pm .044$ | $.960 \pm .053$ | $.933 \pm .067$ |
| | nodes | $2.0 \pm 0.0$ | $2.0 \pm 0.0$ | $3.8 \pm 0.7$ | $5.0 \pm 0.9$ | $8.7 \pm 1.7$ | $12.6 \pm 1.5$ | $16.1 \pm 1.6$ |
| | path len | $1.67 \pm 0.09$ | $1.67 \pm 0.09$ | $2.29 \pm 0.33$ | $2.38 \pm 0.22$ | $2.81 \pm 0.25$ | $3.15 \pm 0.44$ | $3.66 \pm 0.42$ |
| | gini | $.584 \pm .103$ | $.580 \pm .111$ | $.459 \pm .078$ | $.483 \pm .082$ | $.554 \pm .039$ | $.565 \pm .054$ | $.564 \pm .047$ |
| | time | $0.7 \pm 0.0$ | $0.7 \pm 0.0$ | $1.1 \pm 0.2$ | $1.4 \pm 0.2$ | $2.1 \pm 0.3$ | $2.9 \pm 0.3$ | $3.6 \pm 0.3$ |
| heart | acc | $.799 \pm .055$ | $.795 \pm .028$ | $.776 \pm .043$ | $.763 \pm .052$ | $.782 \pm .066$ | $.749 \pm .046$ | $.762 \pm .058$ |
| | nodes | $1.0 \pm 0.0$ | $1.7 \pm 0.5$ | $5.7 \pm 1.7$ | $19.5 \pm 3.3$ | $21.7 \pm 1.7$ | $26.8 \pm 2.6$ | $27.1 \pm 3.6$ |
| | path len | $1.00 \pm 0.00$ | $1.41 \pm 0.29$ | $2.66 \pm 0.63$ | $4.12 \pm 0.49$ | $4.82 \pm 0.58$ | $5.45 \pm 0.62$ | $6.01 \pm 1.01$ |
| | gini | $.888 \pm .010$ | $.881 \pm .024$ | $.862 \pm .014$ | $.824 \pm .045$ | $.830 \pm .034$ | $.814 \pm .041$ | $.836 \pm .032$ |
| | time | $0.6 \pm 0.0$ | $0.8 \pm 0.1$ | $1.9 \pm 0.5$ | $3.5 \pm 0.5$ | $3.6 \pm 0.3$ | $4.6 \pm 0.5$ | $4.8 \pm 0.6$ |
| dry-bean | acc | $.788 \pm .007$ | $.910 \pm .007$ | $.917 \pm .007$ | $.919 \pm .006$ | $.917 \pm .008$ | $.923 \pm .005$ | $.922 \pm .006$ |
| | nodes | $4.0 \pm 0.0$ | $6.0 \pm 0.0$ | $8.0 \pm 0.0$ | $9.9 \pm 0.7$ | $18.2 \pm 1.0$ | $41.2 \pm 3.2$ | $94.3 \pm 3.5$ |
| | path len | $2.84 \pm 0.03$ | $3.24 \pm 0.05$ | $3.40 \pm 0.09$ | $3.60 \pm 0.17$ | $4.74 \pm 0.15$ | $6.19 \pm 0.34$ | $7.62 \pm 0.33$ |
| | gini | $.902 \pm .002$ | $.902 \pm .003$ | $.910 \pm .002$ | $.909 \pm .003$ | $.909 \pm .002$ | $.906 \pm .004$ | $.908 \pm .002$ |
| | time | $3.1 \pm 0.1$ | $3.9 \pm 0.1$ | $4.9 \pm 0.1$ | $5.6 \pm 0.3$ | $8.8 \pm 0.3$ | $12.7 \pm 0.9$ | $25.2 \pm 0.7$ |
| wine | acc | $.961 \pm .026$ | $.972 \pm .028$ | $.972 \pm .028$ | $.972 \pm .028$ | $.966 \pm .027$ | $.977 \pm .028$ | $.972 \pm .028$ |
| | nodes | $2.0 \pm 0.0$ | $2.0 \pm 0.0$ | $2.8 \pm 0.4$ | $2.7 \pm 0.5$ | $2.9 \pm 0.3$ | $2.8 \pm 0.4$ | $2.7 \pm 0.5$ |
| | path len | $1.67 \pm 0.09$ | $1.68 \pm 0.09$ | $1.92 \pm 0.16$ | $1.90 \pm 0.16$ | $1.95 \pm 0.15$ | $1.96 \pm 0.18$ | $1.89 \pm 0.17$ |
| | gini | $.871 \pm .011$ | $.875 \pm .009$ | $.884 \pm .011$ | $.885 \pm .007$ | $.891 \pm .005$ | $.886 \pm .008$ | $.885 \pm .009$ |
| | time | $0.5 \pm 0.0$ | $0.5 \pm 0.0$ | $0.6 \pm 0.1$ | $0.6 \pm 0.1$ | $0.7 \pm 0.0$ | $0.6 \pm 0.1$ | $0.6 \pm 0.1$ |
| car | acc | $.700 \pm .044$ | $.908 \pm .031$ | $.965 \pm .013$ | $.986 \pm .009$ | $.992 \pm .012$ | $.992 \pm .009$ | $.992 \pm .010$ |
| | nodes | $0.0 \pm 0.0$ | $2.8 \pm 0.4$ | $5.0 \pm 0.8$ | $12.3 \pm 1.6$ | $18.8 \pm 2.7$ | $23.7 \pm 2.1$ | $24.5 \pm 4.1$ |
| | path len | $0.00 \pm 0.00$ | $1.60 \pm 0.16$ | $1.91 \pm 0.18$ | $2.86 \pm 0.34$ | $3.23 \pm 0.52$ | $3.61 \pm 0.28$ | $3.70 \pm 0.64$ |
| | gini | $.000 \pm .000$ | $.905 \pm .010$ | $.897 \pm .014$ | $.902 \pm .015$ | $.902 \pm .014$ | $.903 \pm .007$ | $.901 \pm .009$ |
| | time | $0.2 \pm 0.0$ | $1.1 \pm 0.1$ | $1.6 \pm 0.2$ | $3.2 \pm 0.4$ | $4.7 \pm 0.6$ | $5.9 \pm 0.6$ | $6.0 \pm 1.0$ |
| wdbc | acc | $.965 \pm .025$ | $.967 \pm .023$ | $.960 \pm .027$ | $.965 \pm .022$ | $.967 \pm .027$ | $.951 \pm .025$ | $.961 \pm .037$ |
| | nodes | $1.0 \pm 0.0$ | $1.0 \pm 0.0$ | $1.0 \pm 0.0$ | $2.1 \pm 0.9$ | $5.9 \pm 2.1$ | $13.9 \pm 4.2$ | $19.6 \pm 3.6$ |
| | path len | $1.00 \pm 0.00$ | $1.00 \pm 0.00$ | $1.00 \pm 0.00$ | $1.59 \pm 0.48$ | $2.52 \pm 0.32$ | $3.75 \pm 0.69$ | $4.51 \pm 0.41$ |
| | gini | $.954 \pm .003$ | $.955 \pm .003$ | $.954 \pm .003$ | $.951 \pm .003$ | $.951 \pm .003$ | $.950 \pm .003$ | $.951 \pm .004$ |
| | time | $0.5 \pm 0.0$ | $0.5 \pm 0.0$ | $0.5 \pm 0.0$ | $0.8 \pm 0.2$ | $1.6 \pm 0.4$ | $3.4 \pm 0.9$ | $4.6 \pm 0.7$ |
| sonar | acc | $.760 \pm .099$ | $.817 \pm .045$ | $.788 \pm .081$ | $.813 \pm .092$ | $.775 \pm .076$ | $.856 \pm .038$ | $.807 \pm .063$ |
| | nodes | $1.0 \pm 0.0$ | $2.2 \pm 0.6$ | $4.5 \pm 0.8$ | $5.6 \pm 0.9$ | $6.8 \pm 3.0$ | $7.6 \pm 2.1$ | $8.0 \pm 2.4$ |
| | path len | $1.00 \pm 0.00$ | $1.59 \pm 0.24$ | $2.52 \pm 0.32$ | $2.87 \pm 0.53$ | $3.34 \pm 0.94$ | $3.55 \pm 0.71$ | $3.56 \pm 0.75$ |
| | gini | $.971 \pm .002$ | $.972 \pm .001$ | $.973 \pm .001$ | $.949 \pm .074$ | $.973 \pm .002$ | $.974 \pm .001$ | $.973 \pm .002$ |
| | time | $0.5 \pm 0.0$ | $0.8 \pm 0.2$ | $1.3 \pm 0.2$ | $1.6 \pm 0.2$ | $1.8 \pm 0.6$ | $2.0 \pm 0.4$ | $2.1 \pm 0.5$ |
| pendigits | acc | $.094 \pm .003$ | $.910 \pm .021$ | $.953 \pm .011$ | $.971 \pm .008$ | $.983 \pm .004$ | $.987 \pm .004$ | $.988 \pm .003$ |
| | nodes | $0.0 \pm 0.0$ | $9.3 \pm 0.5$ | $12.1 \pm 1.0$ | $17.9 \pm 1.7$ | $27.4 \pm 3.0$ | $57.5 \pm 6.6$ | $125.9 \pm 9.5$ |
| | path len | $0.00 \pm 0.00$ | $4.18 \pm 0.32$ | $4.46 \pm 0.31$ | $5.18 \pm 0.23$ | $5.64 \pm 0.45$ | $6.59 \pm 0.43$ | $8.20 \pm 0.61$ |
| | gini | $.000 \pm .000$ | $.891 \pm .008$ | $.892 \pm .005$ | $.891 \pm .005$ | $.896 \pm .006$ | $.899 \pm .002$ | $.897 \pm .005$ |
| | time | $0.3 \pm 0.0$ | $4.2 \pm 0.1$ | $5.0 \pm 0.4$ | $6.9 \pm 0.5$ | $9.3 \pm 0.8$ | $16.7 \pm 1.5$ | $32.2 \pm 2.2$ |
| ionosphere | acc | $.855 \pm .054$ | $.932 \pm .060$ | $.926 \pm .048$ | $.937 \pm .036$ | $.923 \pm .048$ | $.923 \pm .051$ | $.946 \pm .039$ |
| | nodes | $1.0 \pm 0.0$ | $2.1 \pm 0.3$ | $3.8 \pm 0.4$ | $4.8 \pm 0.7$ | $8.5 \pm 1.9$ | $9.5 \pm 1.8$ | $11.8 \pm 2.0$ |
| | path len | $1.00 \pm 0.00$ | $1.78 \pm 0.14$ | $2.95 \pm 0.24$ | $3.58 \pm 0.62$ | $4.58 \pm 0.55$ | $4.77 \pm 0.89$ | $5.50 \pm 0.79$ |
| | gini | $.948 \pm .003$ | $.929 \pm .012$ | $.925 \pm .005$ | $.927 \pm .005$ | $.939 \pm .005$ | $.934 \pm .008$ | $.931 \pm .004$ |
| | time | $0.7 \pm 0.0$ | $1.0 \pm 0.1$ | $1.6 \pm 0.1$ | $1.9 \pm 0.2$ | $3.0 \pm 0.6$ | $2.2 \pm 0.4$ | $2.6 \pm 0.4$ |

Ours: prototype features, L2 regularization, non-random initialization, fuzzy

| data | metric | 1e-1 | 3e-2 | 1e-2 | 3e-3 | 1e-3 | 3e-4 | 1e-4 |
|---|---|---|---|---|---|---|---|---|
| iris | acc | $.927 \pm .092$ | $.933 \pm .094$ | $.953 \pm .067$ | $.967 \pm .054$ | $.947 \pm .065$ | $.947 \pm .065$ | $.947 \pm .065$ |
| | nodes | $2.0 \pm 0.0$ | $2.0 \pm 0.0$ | $3.7 \pm 1.0$ | $4.9 \pm 0.8$ | $5.7 \pm 0.9$ | $5.6 \pm 0.9$ | $5.6 \pm 0.9$ |
| | path len | $1.67 \pm 0.09$ | $1.67 \pm 0.09$ | $2.09 \pm 0.25$ | $2.23 \pm 0.23$ | $2.31 \pm 0.26$ | $2.28 \pm 0.24$ | $2.28 \pm 0.24$ |
| | gini | $.580 \pm .024$ | $.578 \pm .023$ | $.580 \pm .030$ | $.584 \pm .038$ | $.578 \pm .034$ | $.576 \pm .032$ | $.577 \pm .032$ |
| | time | $1.1 \pm 0.0$ | $1.1 \pm 0.0$ | $1.6 \pm 0.3$ | $1.9 \pm 0.2$ | $2.2 \pm 0.3$ | $2.1 \pm 0.3$ | $2.1 \pm 0.3$ |
| heart | acc | $.756 \pm .061$ | $.746 \pm .055$ | $.776 \pm .054$ | $.765 \pm .065$ | $.766 \pm .058$ | $.729 \pm .069$ | $.749 \pm .046$ |
| | nodes | $1.0 \pm 0.0$ | $1.1 \pm 0.3$ | $6.4 \pm 0.9$ | $25.0 \pm 3.6$ | $48.0 \pm 4.3$ | $51.4 \pm 5.9$ | $52.5 \pm 5.8$ |
| | path len | $1.00 \pm 0.00$ | $1.04 \pm 0.13$ | $2.80 \pm 0.25$ | $4.41 \pm 0.46$ | $5.75 \pm 0.52$ | $5.94 \pm 0.32$ | $5.84 \pm 0.41$ |
| | gini | $.910 \pm .007$ | $.911 \pm .007$ | $.881 \pm .011$ | $.868 \pm .014$ | $.863 \pm .011$ | $.862 \pm .016$ | $.863 \pm .016$ |
| | time | $0.9 \pm 0.0$ | $0.9 \pm 0.1$ | $2.5 \pm 0.3$ | $8.0 \pm 1.1$ | $14.4 \pm 1.2$ | $15.5 \pm 1.6$ | $15.8 \pm 1.6$ |
| dry-bean | acc | $.681 \pm .119$ | $.880 \pm .016$ | $.885 \pm .016$ | $.902 \pm .007$ | $.909 \pm .003$ | $.897 \pm .010$ | $.904 \pm .013$ |
| | nodes | $3.2 \pm 1.0$ | $6.1 \pm 0.3$ | $8.4 \pm 0.7$ | $16.1 \pm 2.1$ | $31.0 \pm 3.7$ | $60.2 \pm 5.9$ | $172.5 \pm 12.8$ |
| | path len | $2.37 \pm 0.56$ | $3.08 \pm 0.15$ | $3.37 \pm 0.16$ | $4.26 \pm 0.30$ | $5.38 \pm 0.24$ | $6.03 \pm 0.29$ | $7.53 \pm 0.38$ |
| | gini | $.858 \pm .021$ | $.832 \pm .019$ | $.737 \pm .063$ | $.658 \pm .056$ | $.619 \pm .078$ | $.729 \pm .031$ | $.714 \pm .027$ |
| | time | $3.1 \pm 0.7$ | $4.5 \pm 0.1$ | $5.8 \pm 0.4$ | $9.5 \pm 1.0$ | $16.4 \pm 2.1$ | $20.2 \pm 1.8$ | $50.1 \pm 3.2$ |
| wine | acc | $.939 \pm .052$ | $.898 \pm .056$ | $.893 \pm .047$ | $.939 \pm .068$ | $.871 \pm .066$ | $.894 \pm .084$ | $.933 \pm .054$ |
| | nodes | $2.0 \pm 0.0$ | $2.6 \pm 0.7$ | $6.1 \pm 1.6$ | $7.3 \pm 2.5$ | $8.9 \pm 3.3$ | $10.0 \pm 2.6$ | $8.8 \pm 1.5$ |
| | path len | $1.69 \pm 0.09$ | $1.92 \pm 0.28$ | $2.99 \pm 0.49$ | $3.08 \pm 0.60$ | $3.31 \pm 0.57$ | $3.66 \pm 0.69$ | $3.47 \pm 0.34$ |
| | gini | $.861 \pm .019$ | $.862 \pm .020$ | $.789 \pm .059$ | $.792 \pm .051$ | $.766 \pm .060$ | $.763 \pm .025$ | $.765 \pm .044$ |
| | time | $0.9 \pm 0.1$ | $1.1 \pm 0.2$ | $1.9 \pm 0.4$ | $2.2 \pm 0.6$ | $2.5 \pm 0.7$ | $2.8 \pm 0.6$ | $2.5 \pm 0.3$ |
| car | acc | $.700 \pm .044$ | $.700 \pm .044$ | $.701 \pm .045$ | $.780 \pm .087$ | $.794 \pm .116$ | $.825 \pm .098$ | $.821 \pm .109$ |
| | nodes | $0.0 \pm 0.0$ | $0.2 \pm 0.4$ | $2.7 \pm 1.9$ | $13.9 \pm 7.0$ | $43.7 \pm 27.1$ | $96.5 \pm 61.5$ | $116.4 \pm 67.0$ |
| | path len | $0.00 \pm 0.00$ | $0.20 \pm 0.40$ | $1.68 \pm 1.06$ | $3.63 \pm 0.62$ | $4.85 \pm 0.70$ | $6.30 \pm 1.18$ | $6.61 \pm 1.10$ |
| | gini | $.000 \pm .000$ | $.182 \pm .363$ | $.716 \pm .358$ | $.890 \pm .016$ | $.869 \pm .015$ | $.873 \pm .015$ | $.869 \pm .017$ |
| | time | $0.3 \pm 0.0$ | $0.4 \pm 0.2$ | $1.3 \pm 0.6$ | $4.2 \pm 1.8$ | $11.2 \pm 6.1$ | $23.0 \pm 13.6$ | $27.5 \pm 14.4$ |
| wdbc | acc | $.930 \pm .047$ | $.946 \pm .045$ | $.951 \pm .030$ | $.933 \pm .041$ | $.931 \pm .028$ | $.930 \pm .034$ | $.910 \pm .033$ |
| | nodes | $1.0 \pm 0.0$ | $1.0 \pm 0.0$ | $2.9 \pm 0.7$ | $9.6 \pm 2.2$ | $14.9 \pm 3.8$ | $19.5 \pm 2.8$ | $21.8 \pm 3.0$ |
| | path len | $1.00 \pm 0.00$ | $1.00 \pm 0.00$ | $1.83 \pm 0.31$ | $3.65 \pm 0.77$ | $4.55 \pm 0.92$ | $5.35 \pm 0.81$ | $5.81 \pm 0.47$ |
| | gini | $.936 \pm .011$ | $.942 \pm .011$ | $.938 \pm .008$ | $.888 \pm .018$ | $.875 \pm .023$ | $.855 \pm .034$ | $.866 \pm .016$ |
| | time | $0.7 \pm 0.0$ | $0.7 \pm 0.0$ | $1.2 \pm 0.2$ | $2.9 \pm 0.6$ | $4.2 \pm 0.9$ | $5.3 \pm 0.7$ | $5.9 \pm 0.8$ |
| sonar | acc | $.716 \pm .111$ | $.730 \pm .050$ | $.721 \pm .052$ | $.730 \pm .088$ | $.711 \pm .114$ | $.721 \pm .089$ | $.770 \pm .075$ |
| | nodes | $1.0 \pm 0.0$ | $4.4 \pm 1.0$ | $11.3 \pm 1.3$ | $16.2 \pm 3.1$ | $18.0 \pm 1.8$ | $18.6 \pm 2.7$ | $18.0 \pm 2.9$ |
| | path len | $1.00 \pm 0.00$ | $2.32 \pm 0.35$ | $4.11 \pm 0.51$ | $4.92 \pm 0.82$ | $5.06 \pm 0.61$ | $5.05 \pm 0.75$ | $5.14 \pm 0.97$ |
| | gini | $.888 \pm .069$ | $.929 \pm .026$ | $.921 \pm .029$ | $.908 \pm .036$ | $.911 \pm .038$ | $.902 \pm .025$ | $.905 \pm .031$ |
| | time | $0.7 \pm 0.0$ | $1.6 \pm 0.2$ | $3.1 \pm 0.3$ | $4.3 \pm 0.7$ | $4.7 \pm 0.4$ | $4.8 \pm 0.6$ | $4.6 \pm 0.7$ |
| pendigits | acc | $.094 \pm .003$ | $.709 \pm .031$ | $.811 \pm .017$ | $.865 \pm .015$ | $.909 \pm .011$ | $.926 \pm .011$ | $.931 \pm .007$ |
| | nodes | $0.0 \pm 0.0$ | $8.1 \pm 0.7$ | $15.6 \pm 1.7$ | $31.8 \pm 2.3$ | $75.5 \pm 3.5$ | $174.3 \pm 12.1$ | $402.2 \pm 24.6$ |
| | path len | $0.00 \pm 0.00$ | $3.74 \pm 0.20$ | $4.60 \pm 0.23$ | $5.66 \pm 0.25$ | $6.79 \pm 0.28$ | $7.72 \pm 0.25$ | $9.76 \pm 0.39$ |
| | gini | $.000 \pm .000$ | $.558 \pm .083$ | $.428 \pm .108$ | $.348 \pm .094$ | $.299 \pm .073$ | $.283 \pm .049$ | $.241 \pm .042$ |
| | time | $0.4 \pm 0.0$ | $4.4 \pm 0.4$ | $7.0 \pm 0.7$ | $12.0 \pm 0.8$ | $23.9 \pm 1.1$ | $48.5 \pm 3.0$ | $101.7 \pm 6.1$ |
| ionosphere | acc | $.878 \pm .057$ | $.889 \pm .050$ | $.923 \pm .036$ | $.903 \pm .043$ | $.903 \pm .045$ | $.903 \pm .041$ | $.900 \pm .039$ |
| | nodes | $1.0 \pm 0.0$ | $1.9 \pm 0.5$ | $4.0 \pm 1.3$ | $9.9 \pm 3.2$ | $19.3 \pm 3.8$ | $20.6 \pm 3.1$ | $21.6 \pm 4.1$ |
| | path len | $1.00 \pm 0.00$ | $1.65 \pm 0.40$ | $2.60 \pm 0.55$ | $3.92 \pm 0.65$ | $6.19 \pm 1.03$ | $6.44 \pm 1.23$ | $6.81 \pm 1.56$ |
| | gini | $.748 \pm .009$ | $.771 \pm .047$ | $.765 \pm .023$ | $.770 \pm .037$ | $.780 \pm .026$ | $.794 \pm .023$ | $.764 \pm .047$ |
| | time | $0.9 \pm 0.0$ | $1.3 \pm 0.2$ | $2.0 \pm 0.4$ | $4.0 \pm 1.1$ | $5.1 \pm 1.0$ | $7.2 \pm 1.0$ | $7.3 \pm 1.2$ |

Ours: prototype features, L2 regularization, non-random initialization, crisp

| data | metric | 1e-1 | 3e-2 | 1e-2 | 3e-3 | 1e-3 | 3e-4 | 1e-4 |
|------|--------|------|------|------|------|------|------|------|
| iris | acc | $.927 \pm .092$ | $.933 \pm .094$ | $.953 \pm .067$ | $.973 \pm .044$ | $.953 \pm .060$ | $.953 \pm .060$ | $.953 \pm .060$ |
| | nodes | $2.0 \pm 0.0$ | $2.0 \pm 0.0$ | $3.7 \pm 1.0$ | $4.9 \pm 0.8$ | $5.7 \pm 0.9$ | $5.6 \pm 0.9$ | $5.6 \pm 0.9$ |
| | path len | $1.67 \pm 0.09$ | $1.67 \pm 0.09$ | $2.09 \pm 0.25$ | $2.23 \pm 0.23$ | $2.31 \pm 0.26$ | $2.29 \pm 0.24$ | $2.29 \pm 0.24$ |
| | gini | $.580 \pm .024$ | $.578 \pm .023$ | $.580 \pm .030$ | $.584 \pm .038$ | $.578 \pm .034$ | $.576 \pm .032$ | $.577 \pm .032$ |
| | time | $1.0 \pm 0.0$ | $1.0 \pm 0.0$ | $1.5 \pm 0.3$ | $1.8 \pm 0.2$ | $2.0 \pm 0.2$ | $2.0 \pm 0.2$ | $2.0 \pm 0.2$ |
| heart | acc | $.756 \pm .061$ | $.746 \pm .055$ | $.779 \pm .055$ | $.769 \pm .060$ | $.766 \pm .065$ | $.736 \pm .069$ | $.743 \pm .051$ |
| | nodes | $1.0 \pm 0.0$ | $1.1 \pm 0.3$ | $6.4 \pm 0.9$ | $24.9 \pm 3.5$ | $48.0 \pm 4.3$ | $51.4 \pm 5.9$ | $52.5 \pm 5.8$ |
| | path len | $1.00 \pm 0.00$ | $1.04 \pm 0.13$ | $2.80 \pm 0.25$ | $4.41 \pm 0.46$ | $5.75 \pm 0.53$ | $5.94 \pm 0.32$ | $5.84 \pm 0.41$ |
| | gini | $.910 \pm .007$ | $.911 \pm .007$ | $.881 \pm .011$ | $.868 \pm .014$ | $.863 \pm .011$ | $.862 \pm .016$ | $.863 \pm .016$ |
| | time | $0.8 \pm 0.1$ | $0.8 \pm 0.1$ | $2.3 \pm 0.3$ | $7.1 \pm 0.9$ | $12.9 \pm 1.1$ | $20.5 \pm 2.2$ | $14.2 \pm 1.4$ |
| dry-bean | acc | $.680 \pm .118$ | $.877 \pm .016$ | $.883 \pm .015$ | $.894 \pm .006$ | $.902 \pm .004$ | $.894 \pm .010$ | $.902 \pm .013$ |
| | nodes | $3.2 \pm 1.0$ | $6.1 \pm 0.3$ | $8.4 \pm 0.7$ | $16.1 \pm 2.1$ | $31.0 \pm 3.7$ | $60.2 \pm 5.9$ | $172.5 \pm 12.8$ |
| | path len | $2.37 \pm 0.57$ | $3.08 \pm 0.15$ | $3.38 \pm 0.16$ | $4.26 \pm 0.31$ | $5.39 \pm 0.24$ | $6.03 \pm 0.30$ | $7.54 \pm 0.39$ |
| | gini | $.858 \pm .021$ | $.832 \pm .019$ | $.738 \pm .063$ | $.659 \pm .057$ | $.618 \pm .080$ | $.730 \pm .032$ | $.714 \pm .027$ |
| | time | $2.9 \pm 0.6$ | $4.3 \pm 0.1$ | $5.2 \pm 0.3$ | $8.3 \pm 0.9$ | $13.5 \pm 1.5$ | $18.2 \pm 1.4$ | $43.6 \pm 2.9$ |
| wine | acc | $.933 \pm .060$ | $.898 \pm .056$ | $.893 \pm .047$ | $.939 \pm .068$ | $.882 \pm .076$ | $.894 \pm .084$ | $.933 \pm .054$ |
| | nodes | $2.0 \pm 0.0$ | $2.6 \pm 0.7$ | $6.1 \pm 1.6$ | $7.3 \pm 2.5$ | $8.9 \pm 3.3$ | $10.0 \pm 2.6$ | $8.8 \pm 1.5$ |
| | path len | $1.69 \pm 0.09$ | $1.92 \pm 0.28$ | $3.01 \pm 0.49$ | $3.07 \pm 0.59$ | $3.30 \pm 0.59$ | $3.68 \pm 0.69$ | $3.46 \pm 0.35$ |
| | gini | $.861 \pm .019$ | $.862 \pm .020$ | $.790 \pm .057$ | $.793 \pm .051$ | $.764 \pm .062$ | $.763 \pm .026$ | $.766 \pm .044$ |
| | time | $0.9 \pm 0.1$ | $1.0 \pm 0.1$ | $1.7 \pm 0.3$ | $2.0 \pm 0.5$ | $2.3 \pm 0.7$ | $2.5 \pm 0.5$ | $2.3 \pm 0.3$ |
| car | acc | $.700 \pm .044$ | $.700 \pm .044$ | $.701 \pm .045$ | $.780 \pm .087$ | $.795 \pm .118$ | $.824 \pm .096$ | $.821 \pm .114$ |
| | nodes | $0.0 \pm 0.0$ | $0.2 \pm 0.4$ | $2.7 \pm 1.9$ | $13.9 \pm 7.0$ | $43.7 \pm 27.1$ | $96.5 \pm 61.5$ | $116.6 \pm 67.2$ |
| | path len | $0.00 \pm 0.00$ | $0.20 \pm 0.40$ | $1.68 \pm 1.06$ | $3.64 \pm 0.62$ | $4.85 \pm 0.70$ | $6.30 \pm 1.18$ | $6.60 \pm 1.10$ |
| | gini | $.000 \pm .000$ | $.182 \pm .363$ | $.716 \pm .358$ | $.890 \pm .016$ | $.869 \pm .015$ | $.873 \pm .015$ | $.869 \pm .017$ |
| | time | $0.3 \pm 0.0$ | $0.4 \pm 0.2$ | $1.2 \pm 0.6$ | $3.8 \pm 1.6$ | $10.1 \pm 5.5$ | $20.2 \pm 11.9$ | $24.3 \pm 12.9$ |
| wdbc | acc | $.930 \pm .047$ | $.946 \pm .045$ | $.947 \pm .029$ | $.933 \pm .041$ | $.930 \pm .027$ | $.930 \pm .034$ | $.910 \pm .033$ |
| | nodes | $1.0 \pm 0.0$ | $1.0 \pm 0.0$ | $2.9 \pm 0.7$ | $9.6 \pm 2.2$ | $14.9 \pm 3.8$ | $19.5 \pm 2.8$ | $21.8 \pm 3.0$ |
| | path len | $1.00 \pm 0.00$ | $1.00 \pm 0.00$ | $1.83 \pm 0.31$ | $3.65 \pm 0.78$ | $4.55 \pm 0.91$ | $5.35 \pm 0.81$ | $5.82 \pm 0.48$ |
| | gini | $.936 \pm .011$ | $.942 \pm .011$ | $.938 \pm .008$ | $.888 \pm .019$ | $.876 \pm .020$ | $.855 \pm .034$ | $.865 \pm .016$ |
| | time | $0.7 \pm 0.0$ | $0.7 \pm 0.0$ | $1.1 \pm 0.2$ | $2.6 \pm 0.5$ | $3.8 \pm 0.9$ | $4.8 \pm 0.6$ | $5.3 \pm 0.6$ |
| sonar | acc | $.716 \pm .111$ | $.730 \pm .050$ | $.716 \pm .050$ | $.730 \pm .088$ | $.716 \pm .113$ | $.730 \pm .092$ | $.770 \pm .078$ |
| | nodes | $1.0 \pm 0.0$ | $4.4 \pm 1.0$ | $11.3 \pm 1.3$ | $16.2 \pm 3.1$ | $18.0 \pm 1.8$ | $18.6 \pm 2.7$ | $18.0 \pm 2.9$ |
| | path len | $1.00 \pm 0.00$ | $2.32 \pm 0.35$ | $4.10 \pm 0.51$ | $4.92 \pm 0.82$ | $5.07 \pm 0.62$ | $5.07 \pm 0.73$ | $5.16 \pm 0.96$ |
| | gini | $.888 \pm .069$ | $.929 \pm .026$ | $.921 \pm .029$ | $.908 \pm .036$ | $.911 \pm .038$ | $.903 \pm .024$ | $.905 \pm .031$ |
| | time | $0.7 \pm 0.0$ | $1.4 \pm 0.2$ | $2.8 \pm 0.3$ | $3.9 \pm 0.6$ | $4.2 \pm 0.3$ | $4.3 \pm 0.5$ | $4.2 \pm 0.6$ |
| pendigits | acc | $.094 \pm .003$ | $.708 \pm .031$ | $.809 \pm .018$ | $.864 \pm .015$ | $.908 \pm .011$ | $.925 \pm .011$ | $.930 \pm .007$ |
| | nodes | $0.0 \pm 0.0$ | $8.1 \pm 0.7$ | $15.6 \pm 1.7$ | $31.8 \pm 2.3$ | $75.5 \pm 3.5$ | $174.3 \pm 12.1$ | $402.2 \pm 24.6$ |
| | path len | $0.00 \pm 0.00$ | $3.74 \pm 0.21$ | $4.60 \pm 0.23$ | $5.66 \pm 0.25$ | $6.79 \pm 0.28$ | $7.72 \pm 0.25$ | $9.76 \pm 0.38$ |
| | gini | $.000 \pm .000$ | $.558 \pm .083$ | $.428 \pm .108$ | $.348 \pm .094$ | $.300 \pm .073$ | $.283 \pm .049$ | $.241 \pm .042$ |
| | time | $0.3 \pm 0.0$ | $4.0 \pm 0.3$ | $6.4 \pm 0.6$ | $10.9 \pm 0.7$ | $21.5 \pm 0.9$ | $43.2 \pm 2.7$ | $89.6 \pm 5.1$ |
| ionosphere | acc | $.878 \pm .057$ | $.892 \pm .049$ | $.923 \pm .036$ | $.903 \pm .043$ | $.897 \pm .048$ | $.903 \pm .041$ | $.900 \pm .039$ |
| | nodes | $1.0 \pm 0.0$ | $1.9 \pm 0.5$ | $4.0 \pm 1.3$ | $9.9 \pm 3.2$ | $19.2 \pm 3.8$ | $20.6 \pm 3.1$ | $21.6 \pm 4.1$ |
| | path len | $1.00 \pm 0.00$ | $1.65 \pm 0.40$ | $2.60 \pm 0.55$ | $3.92 \pm 0.65$ | $6.20 \pm 1.03$ | $6.44 \pm 1.23$ | $6.80 \pm 1.57$ |
| | gini | $.748 \pm .009$ | $.771 \pm .047$ | $.765 \pm .023$ | $.770 \pm .037$ | $.782 \pm .027$ | $.794 \pm .023$ | $.764 \pm .047$ |
| | time | $0.7 \pm 0.0$ | $0.9 \pm 0.2$ | $1.3 \pm 0.2$ | $2.6 \pm 0.7$ | $4.6 \pm 0.8$ | $6.6 \pm 0.9$ | $6.7 \pm 1.2$ |

Ours: linear features, L1 regularization, non-random initialization, fuzzy

| data | metric | 1e-1 | 3e-2 | 1e-2 | 3e-3 | 1e-3 | 3e-4 | 1e-4 |
|---|---|---|---|---|---|---|---|---|
| iris | acc | $.953 \pm .043$ | $.953 \pm .043$ | $.967 \pm .033$ | $.967 \pm .033$ | $.960 \pm .033$ | $.953 \pm .043$ | $.953 \pm .043$ |
| | nodes | $2.0 \pm 0.0$ | $2.0 \pm 0.0$ | $3.7 \pm 0.5$ | $5.0 \pm 1.2$ | $9.5 \pm 1.3$ | $13.9 \pm 2.2$ | $18.8 \pm 4.1$ |
| | path len | $1.67 \pm 0.09$ | $1.67 \pm 0.09$ | $2.25 \pm 0.21$ | $2.37 \pm 0.34$ | $3.09 \pm 0.42$ | $3.51 \pm 0.45$ | $3.86 \pm 0.57$ |
| | gini | $.067 \pm .168$ | $.072 \pm .168$ | $.106 \pm .049$ | $.077 \pm .051$ | $.095 \pm .077$ | $.082 \pm .086$ | $.077 \pm .072$ |
| | time | $0.6 \pm 0.0$ | $0.7 \pm 0.0$ | $1.0 \pm 0.1$ | $1.6 \pm 0.3$ | $2.6 \pm 0.3$ | $3.6 \pm 0.5$ | $4.7 \pm 0.9$ |
| heart | acc | $.772 \pm .056$ | $.799 \pm .052$ | $.796 \pm .065$ | $.772 \pm .036$ | $.763 \pm .074$ | $.766 \pm .082$ | $.766 \pm .096$ |
| | nodes | $1.0 \pm 0.0$ | $1.3 \pm 0.5$ | $6.4 \pm 1.5$ | $25.5 \pm 3.8$ | $36.2 \pm 3.8$ | $41.3 \pm 3.2$ | $42.9 \pm 5.8$ |
| | path len | $1.00 \pm 0.00$ | $1.19 \pm 0.30$ | $2.62 \pm 0.28$ | $4.57 \pm 0.35$ | $5.20 \pm 0.34$ | $5.47 \pm 0.44$ | $5.65 \pm 0.56$ |
| | gini | $.432 \pm .072$ | $.505 \pm .139$ | $.347 \pm .101$ | $.220 \pm .081$ | $.164 \pm .039$ | $.182 \pm .069$ | $.165 \pm .051$ |
| | time | $0.6 \pm 0.0$ | $0.7 \pm 0.2$ | $1.8 \pm 0.5$ | $6.4 \pm 2.0$ | $7.1 \pm 0.7$ | $8.1 \pm 0.6$ | $8.4 \pm 1.1$ |
| dry-bean | acc | $.636 \pm .062$ | $.902 \pm .009$ | $.914 \pm .007$ | $.912 \pm .004$ | $.915 \pm .006$ | $.920 \pm .007$ | $.920 \pm .007$ |
| | nodes | $2.8 \pm 0.4$ | $6.0 \pm 0.0$ | $7.7 \pm 0.5$ | $12.1 \pm 1.4$ | $27.7 \pm 2.0$ | $59.6 \pm 2.0$ | $88.1 \pm 10.7$ |
| | path len | $2.14 \pm 0.25$ | $3.13 \pm 0.22$ | $3.31 \pm 0.12$ | $3.83 \pm 0.19$ | $5.28 \pm 0.23$ | $6.37 \pm 0.14$ | $6.46 \pm 0.31$ |
| | gini | $.619 \pm .031$ | $.658 \pm .019$ | $.674 \pm .026$ | $.535 \pm .076$ | $.510 \pm .043$ | $.359 \pm .042$ | $.406 \pm .033$ |
| | time | $2.6 \pm 0.3$ | $4.5 \pm 0.1$ | $5.3 \pm 0.3$ | $7.3 \pm 0.7$ | $14.5 \pm 0.8$ | $20.9 \pm 0.7$ | $51.1 \pm 24.9$ |
| wine | acc | $.972 \pm .028$ | $.989 \pm .022$ | $.972 \pm .028$ | $.960 \pm .026$ | $.966 \pm .027$ | $.983 \pm .026$ | $.978 \pm .027$ |
| | nodes | $2.0 \pm 0.0$ | $2.1 \pm 0.3$ | $3.0 \pm 0.4$ | $3.0 \pm 0.4$ | $5.0 \pm 1.5$ | $9.1 \pm 2.7$ | $13.1 \pm 3.0$ |
| | path len | $1.68 \pm 0.08$ | $1.72 \pm 0.21$ | $2.10 \pm 0.18$ | $2.03 \pm 0.17$ | $2.55 \pm 0.36$ | $3.39 \pm 0.48$ | $4.23 \pm 0.57$ |
| | gini | $.795 \pm .015$ | $.792 \pm .034$ | $.754 \pm .043$ | $.749 \pm .062$ | $.752 \pm .060$ | $.737 \pm .033$ | $.733 \pm .025$ |
| | time | $0.6 \pm 0.0$ | $0.6 \pm 0.0$ | $0.8 \pm 0.1$ | $0.7 \pm 0.1$ | $1.1 \pm 0.3$ | $1.8 \pm 0.5$ | $2.5 \pm 0.5$ |
| car | acc | $.700 \pm .044$ | $.891 \pm .032$ | $.946 \pm .038$ | $.976 \pm .013$ | $.992 \pm .006$ | $.996 \pm .006$ | $.992 \pm .006$ |
| | nodes | $0.0 \pm 0.0$ | $2.5 \pm 0.8$ | $5.9 \pm 1.5$ | $14.0 \pm 2.4$ | $29.8 \pm 6.0$ | $32.4 \pm 3.7$ | $39.3 \pm 7.5$ |
| | path len | $0.00 \pm 0.00$ | $1.53 \pm 0.29$ | $2.08 \pm 0.26$ | $2.77 \pm 0.14$ | $3.39 \pm 0.25$ | $3.64 \pm 0.42$ | $3.64 \pm 0.34$ |
| | gini | $.000 \pm .000$ | $.580 \pm .133$ | $.566 \pm .035$ | $.542 \pm .071$ | $.580 \pm .082$ | $.572 \pm .062$ | $.507 \pm .043$ |
| | time | $0.2 \pm 0.0$ | $1.1 \pm 0.3$ | $1.9 \pm 0.4$ | $4.0 \pm 0.7$ | $8.0 \pm 1.7$ | $8.6 \pm 1.2$ | $10.2 \pm 1.5$ |
| wdbc | acc | $.975 \pm .020$ | $.974 \pm .021$ | $.974 \pm .016$ | $.968 \pm .015$ | $.970 \pm .019$ | $.967 \pm .029$ | $.963 \pm .030$ |
| | nodes | $1.0 \pm 0.0$ | $1.0 \pm 0.0$ | $1.0 \pm 0.0$ | $3.0 \pm 0.0$ | $7.2 \pm 2.5$ | $17.1 \pm 4.4$ | $31.5 \pm 4.7$ |
| | path len | $1.00 \pm 0.00$ | $1.00 \pm 0.00$ | $1.00 \pm 0.00$ | $2.00 \pm 0.00$ | $2.79 \pm 0.62$ | $3.55 \pm 0.63$ | $4.67 \pm 0.62$ |
| | gini | $.816 \pm .018$ | $.812 \pm .023$ | $.810 \pm .021$ | $.824 \pm .034$ | $.933 \pm .007$ | $.936 \pm .013$ | $.937 \pm .008$ |
| | time | $0.5 \pm 0.0$ | $0.5 \pm 0.0$ | $0.5 \pm 0.0$ | $1.1 \pm 0.0$ | $2.2 \pm 0.6$ | $3.2 \pm 0.8$ | $8.2 \pm 1.5$ |
| sonar | acc | $.711 \pm .088$ | $.769 \pm .095$ | $.822 \pm .058$ | $.783 \pm .075$ | $.803 \pm .083$ | $.851 \pm .069$ | $.836 \pm .081$ |
| | nodes | $1.0 \pm 0.0$ | $2.5 \pm 0.7$ | $7.3 \pm 1.2$ | $7.2 \pm 1.0$ | $13.3 \pm 3.8$ | $18.7 \pm 4.8$ | $27.5 \pm 6.2$ |
| | path len | $1.00 \pm 0.00$ | $1.75 \pm 0.19$ | $3.04 \pm 0.25$ | $3.15 \pm 0.44$ | $3.93 \pm 0.51$ | $4.90 \pm 0.68$ | $5.63 \pm 0.63$ |
| | gini | $.946 \pm .008$ | $.910 \pm .028$ | $.854 \pm .043$ | $.871 \pm .037$ | $.893 \pm .041$ | $.920 \pm .022$ | $.928 \pm .018$ |
| | time | $0.5 \pm 0.0$ | $1.0 \pm 0.2$ | $2.1 \pm 0.3$ | $2.1 \pm 0.3$ | $3.6 \pm 1.0$ | $5.0 \pm 1.3$ | $5.9 \pm 1.5$ |
| pendigits | acc | $.094 \pm .003$ | $.888 \pm .014$ | $.942 \pm .010$ | $.964 \pm .007$ | $.974 \pm .007$ | $.977 \pm .005$ | $.984 \pm .004$ |
| | nodes | $0.0 \pm 0.0$ | $9.1 \pm 0.3$ | $11.7 \pm 0.9$ | $18.6 \pm 1.5$ | $42.1 \pm 5.0$ | $120.7 \pm 7.1$ | $290.0 \pm 12.9$ |
| | path len | $0.00 \pm 0.00$ | $3.83 \pm 0.27$ | $4.40 \pm 0.39$ | $5.08 \pm 0.20$ | $5.92 \pm 0.29$ | $7.49 \pm 0.37$ | $8.95 \pm 0.31$ |
| | gini | $.000 \pm .000$ | $.803 \pm .013$ | $.795 \pm .027$ | $.751 \pm .014$ | $.689 \pm .042$ | $.503 \pm .029$ | $.329 \pm .032$ |
| | time | $0.3 \pm 0.0$ | $4.5 \pm 0.1$ | $5.5 \pm 0.3$ | $7.9 \pm 0.5$ | $15.2 \pm 1.5$ | $55.7 \pm 12.3$ | $142.1 \pm 25.0$ |
| ionosphere | acc | $.829 \pm .044$ | $.895 \pm .057$ | $.920 \pm .038$ | $.923 \pm .034$ | $.906 \pm .056$ | $.897 \pm .069$ | $.937 \pm .031$ |
| | nodes | $1.0 \pm 0.0$ | $2.2 \pm 0.4$ | $4.4 \pm 0.5$ | $6.2 \pm 1.2$ | $14.9 \pm 3.1$ | $26.4 \pm 4.8$ | $30.4 \pm 5.4$ |
| | path len | $1.00 \pm 0.00$ | $1.91 \pm 0.32$ | $3.43 \pm 0.41$ | $3.89 \pm 0.44$ | $5.51 \pm 0.84$ | $7.59 \pm 0.87$ | $8.98 \pm 1.28$ |
| | gini | $.447 \pm .056$ | $.640 \pm .147$ | $.532 \pm .130$ | $.657 \pm .071$ | $.691 \pm .060$ | $.772 \pm .080$ | $.781 \pm .076$ |
| | time | $0.5 \pm 0.0$ | $0.9 \pm 0.1$ | $1.4 \pm 0.1$ | $1.8 \pm 0.3$ | $3.9 \pm 0.8$ | $6.3 \pm 1.1$ | $7.9 \pm 1.5$ |

Ours: linear features, L1 regularization, non-random initialization, crisp

| data | metric | 1e-1 | 3e-2 | 1e-2 | 3e-3 | 1e-3 | 3e-4 | 1e-4 |
|------|--------|------|------|------|------|------|------|------|
| iris | acc | $.953 \pm .043$ | $.953 \pm .043$ | $.980 \pm .031$ | $.960 \pm .033$ | $.973 \pm .033$ | $.920 \pm .065$ | $.967 \pm .033$ |
| | nodes | $2.0 \pm 0.0$ | $2.0 \pm 0.0$ | $3.7 \pm 0.5$ | $5.0 \pm 1.2$ | $9.5 \pm 1.3$ | $13.9 \pm 2.2$ | $18.8 \pm 4.1$ |
| | path len | $1.67 \pm 0.09$ | $1.67 \pm 0.09$ | $2.25 \pm 0.21$ | $2.38 \pm 0.33$ | $3.05 \pm 0.44$ | $3.52 \pm 0.45$ | $3.87 \pm 0.55$ |
| | gini | $.063 \pm .170$ | $.071 \pm .168$ | $.101 \pm .046$ | $.079 \pm .057$ | $.091 \pm .075$ | $.079 \pm .086$ | $.063 \pm .046$ |
| | time | $0.7 \pm 0.0$ | $0.7 \pm 0.0$ | $1.1 \pm 0.1$ | $1.4 \pm 0.2$ | $2.3 \pm 0.3$ | $3.1 \pm 0.4$ | $4.1 \pm 0.8$ |
| heart | acc | $.772 \pm .056$ | $.789 \pm .056$ | $.796 \pm .061$ | $.779 \pm .034$ | $.766 \pm .078$ | $.766 \pm .078$ | $.762 \pm .095$ |
| | nodes | $1.0 \pm 0.0$ | $1.3 \pm 0.5$ | $6.4 \pm 1.5$ | $25.5 \pm 3.8$ | $36.1 \pm 3.8$ | $41.3 \pm 3.2$ | $42.9 \pm 5.8$ |
| | path len | $1.00 \pm 0.00$ | $1.19 \pm 0.30$ | $2.61 \pm 0.30$ | $4.55 \pm 0.37$ | $5.21 \pm 0.31$ | $5.47 \pm 0.42$ | $5.61 \pm 0.53$ |
| | gini | $.432 \pm .072$ | $.505 \pm .140$ | $.346 \pm .102$ | $.220 \pm .081$ | $.165 \pm .039$ | $.181 \pm .070$ | $.165 \pm .051$ |
| | time | $0.6 \pm 0.0$ | $0.7 \pm 0.1$ | $1.6 \pm 0.5$ | $6.0 \pm 1.6$ | $6.2 \pm 0.6$ | $7.1 \pm 0.5$ | $7.4 \pm 1.0$ |
| dry-bean | acc | $.632 \pm .061$ | $.891 \pm .010$ | $.910 \pm .010$ | $.905 \pm .006$ | $.906 \pm .006$ | $.913 \pm .009$ | $.908 \pm .008$ |
| | nodes | $2.8 \pm 0.4$ | $6.0 \pm 0.0$ | $7.7 \pm 0.5$ | $12.1 \pm 1.4$ | $27.7 \pm 2.0$ | $44.1 \pm 3.2$ | $88.1 \pm 10.7$ |
| | path len | $2.15 \pm 0.25$ | $3.15 \pm 0.23$ | $3.31 \pm 0.12$ | $3.85 \pm 0.20$ | $5.33 \pm 0.25$ | $5.57 \pm 0.17$ | $6.49 \pm 0.32$ |
| | gini | $.619 \pm .031$ | $.658 \pm .019$ | $.674 \pm .026$ | $.536 \pm .076$ | $.511 \pm .044$ | $.360 \pm .038$ | $.406 \pm .033$ |
| | time | $2.4 \pm 0.3$ | $4.0 \pm 0.1$ | $4.8 \pm 0.3$ | $6.3 \pm 0.5$ | $11.5 \pm 0.6$ | $42.8 \pm 15.5$ | $50.7 \pm 14.2$ |
| wine | acc | $.972 \pm .028$ | $.989 \pm .022$ | $.972 \pm .028$ | $.955 \pm .034$ | $.961 \pm .026$ | $.978 \pm .027$ | $.972 \pm .028$ |
| | nodes | $2.0 \pm 0.0$ | $2.1 \pm 0.3$ | $3.0 \pm 0.4$ | $3.0 \pm 0.4$ | $5.0 \pm 1.5$ | $9.1 \pm 2.7$ | $13.1 \pm 3.0$ |
| | path len | $1.68 \pm 0.07$ | $1.73 \pm 0.21$ | $2.09 \pm 0.18$ | $2.03 \pm 0.17$ | $2.53 \pm 0.37$ | $3.41 \pm 0.49$ | $4.24 \pm 0.58$ |
| | gini | $.795 \pm .015$ | $.792 \pm .033$ | $.755 \pm .042$ | $.749 \pm .062$ | $.752 \pm .063$ | $.737 \pm .034$ | $.733 \pm .025$ |
| | time | $0.5 \pm 0.0$ | $0.5 \pm 0.0$ | $0.7 \pm 0.1$ | $0.7 \pm 0.1$ | $1.0 \pm 0.2$ | $1.6 \pm 0.4$ | $2.2 \pm 0.4$ |
| car | acc | $.700 \pm .044$ | $.890 \pm .033$ | $.945 \pm .038$ | $.976 \pm .013$ | $.992 \pm .006$ | $.996 \pm .006$ | $.992 \pm .006$ |
| | nodes | $0.0 \pm 0.0$ | $2.5 \pm 0.8$ | $5.9 \pm 1.5$ | $14.0 \pm 2.4$ | $29.8 \pm 6.0$ | $32.4 \pm 3.7$ | $39.3 \pm 7.5$ |
| | path len | $0.00 \pm 0.00$ | $1.53 \pm 0.29$ | $2.09 \pm 0.26$ | $2.78 \pm 0.14$ | $3.39 \pm 0.25$ | $3.64 \pm 0.42$ | $3.65 \pm 0.35$ |
| | gini | $.000 \pm .000$ | $.580 \pm .133$ | $.566 \pm .036$ | $.542 \pm .071$ | $.580 \pm .081$ | $.573 \pm .062$ | $.507 \pm .043$ |
| | time | $0.2 \pm 0.0$ | $1.0 \pm 0.2$ | $1.8 \pm 0.4$ | $3.6 \pm 0.6$ | $7.1 \pm 1.5$ | $7.6 \pm 1.0$ | $9.1 \pm 1.5$ |
| wdbc | acc | $.975 \pm .020$ | $.974 \pm .021$ | $.974 \pm .016$ | $.968 \pm .015$ | $.965 \pm .025$ | $.963 \pm .035$ | $.961 \pm .028$ |
| | nodes | $1.0 \pm 0.0$ | $1.0 \pm 0.0$ | $1.0 \pm 0.0$ | $3.0 \pm 0.0$ | $7.2 \pm 2.5$ | $17.1 \pm 4.4$ | $31.5 \pm 4.7$ |
| | path len | $1.00 \pm 0.00$ | $1.00 \pm 0.00$ | $1.00 \pm 0.00$ | $2.00 \pm 0.00$ | $2.80 \pm 0.62$ | $3.54 \pm 0.66$ | $4.71 \pm 0.58$ |
| | gini | $.816 \pm .018$ | $.812 \pm .023$ | $.810 \pm .021$ | $.824 \pm .034$ | $.933 \pm .007$ | $.936 \pm .013$ | $.937 \pm .008$ |
| | time | $0.5 \pm 0.0$ | $0.5 \pm 0.0$ | $0.5 \pm 0.0$ | $1.0 \pm 0.0$ | $1.9 \pm 0.5$ | $2.8 \pm 0.6$ | $7.0 \pm 1.2$ |
| sonar | acc | $.712 \pm .100$ | $.769 \pm .083$ | $.812 \pm .063$ | $.774 \pm .105$ | $.818 \pm .062$ | $.846 \pm .083$ | $.822 \pm .098$ |
| | nodes | $1.0 \pm 0.0$ | $2.5 \pm 0.7$ | $7.3 \pm 1.2$ | $7.2 \pm 1.0$ | $13.3 \pm 3.8$ | $18.7 \pm 4.8$ | $27.5 \pm 6.2$ |
| | path len | $1.00 \pm 0.00$ | $1.77 \pm 0.18$ | $3.04 \pm 0.27$ | $3.16 \pm 0.46$ | $3.87 \pm 0.53$ | $4.84 \pm 0.65$ | $5.64 \pm 0.61$ |
| | gini | $.946 \pm .008$ | $.910 \pm .028$ | $.853 \pm .044$ | $.872 \pm .036$ | $.894 \pm .042$ | $.920 \pm .023$ | $.928 \pm .019$ |
| | time | $0.5 \pm 0.0$ | $0.9 \pm 0.2$ | $1.9 \pm 0.2$ | $1.9 \pm 0.2$ | $3.2 \pm 0.8$ | $4.3 \pm 1.1$ | $5.2 \pm 1.2$ |
| pendigits | acc | $.094 \pm .003$ | $.885 \pm .013$ | $.938 \pm .011$ | $.960 \pm .010$ | $.968 \pm .005$ | $.969 \pm .006$ | $.976 \pm .004$ |
| | nodes | $0.0 \pm 0.0$ | $9.1 \pm 0.3$ | $11.7 \pm 0.9$ | $18.6 \pm 1.5$ | $42.1 \pm 5.0$ | $120.7 \pm 7.1$ | $290.0 \pm 12.9$ |
| | path len | $0.00 \pm 0.00$ | $3.83 \pm 0.27$ | $4.41 \pm 0.39$ | $5.08 \pm 0.20$ | $5.91 \pm 0.29$ | $7.49 \pm 0.40$ | $8.97 \pm 0.34$ |
| | gini | $.000 \pm .000$ | $.803 \pm .013$ | $.795 \pm .027$ | $.751 \pm .014$ | $.689 \pm .042$ | $.503 \pm .029$ | $.329 \pm .032$ |
| | time | $0.3 \pm 0.0$ | $5.2 \pm 0.2$ | $6.3 \pm 0.4$ | $9.1 \pm 0.6$ | $17.4 \pm 1.7$ | $40.2 \pm 2.4$ | $86.1 \pm 5.0$ |
| ionosphere | acc | $.835 \pm .044$ | $.892 \pm .057$ | $.917 \pm .043$ | $.920 \pm .038$ | $.897 \pm .054$ | $.909 \pm .055$ | $.909 \pm .069$ |
| | nodes | $1.0 \pm 0.0$ | $2.2 \pm 0.4$ | $4.4 \pm 0.5$ | $6.2 \pm 1.2$ | $14.9 \pm 3.1$ | $26.4 \pm 4.8$ | $30.4 \pm 5.4$ |
| | path len | $1.00 \pm 0.00$ | $1.92 \pm 0.32$ | $3.51 \pm 0.44$ | $3.94 \pm 0.44$ | $5.54 \pm 0.85$ | $7.67 \pm 0.90$ | $9.23 \pm 1.41$ |
| | gini | $.447 \pm .056$ | $.640 \pm .147$ | $.533 \pm .129$ | $.658 \pm .070$ | $.691 \pm .060$ | $.772 \pm .081$ | $.781 \pm .077$ |
| | time | $0.5 \pm 0.0$ | $0.8 \pm 0.1$ | $1.2 \pm 0.1$ | $1.6 \pm 0.3$ | $3.4 \pm 0.6$ | $5.8 \pm 1.0$ | $6.7 \pm 1.2$ |

Ours: prototype features, L1 regularization, non-random initialization, fuzzy

| data | metric | 1e-1 | 3e-2 | 1e-2 | 3e-3 | 1e-3 | 3e-4 | 1e-4 |
|---|---|---|---|---|---|---|---|---|
| iris | acc | $.940 \pm .055$ | $.940 \pm .055$ | $.933 \pm .067$ | $.933 \pm .052$ | $.927 \pm .063$ | $.900 \pm .086$ | $.920 \pm .088$ |
| | nodes | $2.0 \pm 0.0$ | $2.0 \pm 0.0$ | $3.3 \pm 0.8$ | $5.3 \pm 0.6$ | $5.9 \pm 0.9$ | $6.3 \pm 1.0$ | $6.1 \pm 0.8$ |
| | path len | $1.66 \pm 0.09$ | $1.66 \pm 0.09$ | $2.02 \pm 0.30$ | $2.24 \pm 0.23$ | $2.30 \pm 0.14$ | $2.34 \pm 0.19$ | $2.32 \pm 0.22$ |
| | gini | $.571 \pm .025$ | $.567 \pm .024$ | $.542 \pm .042$ | $.535 \pm .029$ | $.552 \pm .045$ | $.538 \pm .045$ | $.527 \pm .046$ |
| | time | $1.1 \pm 0.0$ | $1.1 \pm 0.0$ | $1.3 \pm 0.3$ | $2.1 \pm 0.2$ | $2.2 \pm 0.3$ | $2.3 \pm 0.3$ | $2.3 \pm 0.2$ |
| heart | acc | $.756 \pm .065$ | $.743 \pm .060$ | $.759 \pm .071$ | $.786 \pm .063$ | $.746 \pm .066$ | $.753 \pm .061$ | $.736 \pm .062$ |
| | nodes | $1.0 \pm 0.0$ | $1.3 \pm 0.6$ | $7.0 \pm 2.2$ | $27.3 \pm 5.2$ | $46.9 \pm 6.7$ | $52.4 \pm 6.2$ | $54.1 \pm 6.2$ |
| | path len | $1.00 \pm 0.00$ | $1.13 \pm 0.29$ | $2.88 \pm 0.57$ | $4.73 \pm 0.68$ | $5.61 \pm 0.69$ | $6.27 \pm 0.53$ | $6.33 \pm 0.62$ |
| | gini | $.912 \pm .009$ | $.908 \pm .015$ | $.872 \pm .022$ | $.852 \pm .019$ | $.844 \pm .017$ | $.850 \pm .014$ | $.853 \pm .015$ |
| | time | $0.9 \pm 0.1$ | $1.0 \pm 0.2$ | $2.7 \pm 0.7$ | $8.7 \pm 1.5$ | $11.0 \pm 1.5$ | $12.4 \pm 1.4$ | $12.6 \pm 1.3$ |
| dry-bean | acc | $.714 \pm .086$ | $.863 \pm .007$ | $.871 \pm .014$ | $.878 \pm .010$ | $.890 \pm .010$ | $.893 \pm .009$ | $.895 \pm .010$ |
| | nodes | $3.6 \pm 0.7$ | $6.6 \pm 0.5$ | $8.5 \pm 0.9$ | $13.9 \pm 1.8$ | $26.0 \pm 2.1$ | $43.0 \pm 7.4$ | $97.5 \pm 12.4$ |
| | path len | $2.36 \pm 0.31$ | $3.17 \pm 0.17$ | $3.50 \pm 0.23$ | $4.23 \pm 0.29$ | $5.07 \pm 0.25$ | $5.44 \pm 0.30$ | $6.73 \pm 0.31$ |
| | gini | $.830 \pm .033$ | $.806 \pm .029$ | $.719 \pm .047$ | $.583 \pm .066$ | $.524 \pm .066$ | $.516 \pm .052$ | $.509 \pm .030$ |
| | time | $3.3 \pm 0.4$ | $4.9 \pm 0.3$ | $6.0 \pm 0.6$ | $8.4 \pm 0.9$ | $13.9 \pm 1.3$ | $70.2 \pm 22.0$ | $120.3 \pm 23.2$ |
| wine | acc | $.893 \pm .030$ | $.927 \pm .051$ | $.905 \pm .050$ | $.911 \pm .051$ | $.938 \pm .047$ | $.922 \pm .044$ | $.949 \pm .060$ |
| | nodes | $2.0 \pm 0.0$ | $2.4 \pm 0.5$ | $5.7 \pm 2.1$ | $7.4 \pm 1.9$ | $8.1 \pm 1.0$ | $9.2 \pm 2.4$ | $10.7 \pm 3.0$ |
| | path len | $1.67 \pm 0.12$ | $1.79 \pm 0.21$ | $2.77 \pm 0.64$ | $3.12 \pm 0.37$ | $3.21 \pm 0.23$ | $3.53 \pm 0.39$ | $3.73 \pm 0.50$ |
| | gini | $.815 \pm .043$ | $.811 \pm .054$ | $.729 \pm .056$ | $.699 \pm .054$ | $.677 \pm .082$ | $.685 \pm .068$ | $.680 \pm .057$ |
| | time | $0.9 \pm 0.1$ | $1.0 \pm 0.1$ | $1.8 \pm 0.5$ | $2.2 \pm 0.4$ | $2.4 \pm 0.2$ | $2.6 \pm 0.5$ | $2.9 \pm 0.7$ |
| car | acc | $.700 \pm .044$ | $.700 \pm .044$ | $.740 \pm .098$ | $.780 \pm .102$ | $.800 \pm .094$ | $.806 \pm .111$ | $.790 \pm .126$ |
| | nodes | $0.0 \pm 0.0$ | $0.3 \pm 0.5$ | $4.9 \pm 3.2$ | $15.4 \pm 5.8$ | $39.5 \pm 23.4$ | $83.5 \pm 57.5$ | $105.4 \pm 59.8$ |
| | path len | $0.00 \pm 0.00$ | $0.30 \pm 0.46$ | $2.43 \pm 1.05$ | $4.16 \pm 0.92$ | $5.10 \pm 0.92$ | $6.22 \pm 1.67$ | $6.62 \pm 1.34$ |
| | gini | $.000 \pm .000$ | $.266 \pm .407$ | $.896 \pm .015$ | $.889 \pm .012$ | $.879 \pm .014$ | $.870 \pm .015$ | $.871 \pm .012$ |
| | time | $0.3 \pm 0.0$ | $0.4 \pm 0.2$ | $1.9 \pm 0.9$ | $4.8 \pm 1.4$ | $10.4 \pm 5.5$ | $20.6 \pm 13.0$ | $25.3 \pm 13.1$ |
| wdbc | acc | $.937 \pm .029$ | $.942 \pm .029$ | $.942 \pm .029$ | $.928 \pm .041$ | $.928 \pm .036$ | $.924 \pm .038$ | $.930 \pm .044$ |
| | nodes | $1.0 \pm 0.0$ | $1.1 \pm 0.3$ | $2.9 \pm 0.8$ | $10.0 \pm 2.9$ | $18.2 \pm 1.6$ | $22.2 \pm 2.8$ | $25.7 \pm 2.5$ |
| | path len | $1.00 \pm 0.00$ | $1.04 \pm 0.13$ | $1.77 \pm 0.37$ | $3.37 \pm 0.62$ | $4.56 \pm 0.60$ | $5.03 \pm 0.85$ | $6.05 \pm 0.94$ |
| | gini | $.917 \pm .022$ | $.915 \pm .022$ | $.925 \pm .012$ | $.878 \pm .025$ | $.842 \pm .023$ | $.816 \pm .027$ | $.786 \pm .054$ |
| | time | $0.7 \pm 0.0$ | $0.8 \pm 0.1$ | $1.2 \pm 0.2$ | $3.0 \pm 0.8$ | $5.0 \pm 0.5$ | $6.0 \pm 0.7$ | $6.9 \pm 0.6$ |
| sonar | acc | $.716 \pm .102$ | $.735 \pm .085$ | $.764 \pm .052$ | $.754 \pm .084$ | $.764 \pm .118$ | $.774 \pm .084$ | $.763 \pm .089$ |
| | nodes | $1.0 \pm 0.0$ | $3.9 \pm 0.7$ | $11.6 \pm 1.4$ | $19.2 \pm 2.7$ | $18.7 \pm 2.1$ | $18.9 \pm 2.2$ | $19.5 \pm 2.2$ |
| | path len | $1.00 \pm 0.00$ | $2.25 \pm 0.20$ | $4.01 \pm 0.46$ | $5.25 \pm 0.55$ | $5.10 \pm 0.63$ | $5.07 \pm 0.65$ | $5.07 \pm 0.73$ |
| | gini | $.888 \pm .069$ | $.921 \pm .035$ | $.890 \pm .037$ | $.875 \pm .034$ | $.891 \pm .020$ | $.888 \pm .044$ | $.886 \pm .025$ |
| | time | $0.7 \pm 0.0$ | $1.5 \pm 0.2$ | $3.2 \pm 0.3$ | $4.9 \pm 0.7$ | $4.8 \pm 0.6$ | $4.8 \pm 0.5$ | $5.0 \pm 0.5$ |
| pendigits | acc | $.094 \pm .003$ | $.718 \pm .041$ | $.791 \pm .024$ | $.869 \pm .018$ | $.902 \pm .011$ | $.922 \pm .010$ | $.931 \pm .006$ |
| | nodes | $0.0 \pm 0.0$ | $8.4 \pm 0.9$ | $15.0 \pm 1.6$ | $34.3 \pm 2.2$ | $78.0 \pm 3.3$ | $180.0 \pm 12.1$ | $408.8 \pm 21.5$ |
| | path len | $0.00 \pm 0.00$ | $3.81 \pm 0.27$ | $4.46 \pm 0.26$ | $5.79 \pm 0.17$ | $6.77 \pm 0.26$ | $8.06 \pm 0.30$ | $9.26 \pm 0.19$ |
| | gini | $.000 \pm .000$ | $.527 \pm .070$ | $.407 \pm .104$ | $.317 \pm .071$ | $.294 \pm .063$ | $.295 \pm .047$ | $.288 \pm .036$ |
| | time | $0.4 \pm 0.0$ | $5.6 \pm 0.6$ | $8.8 \pm 0.8$ | $17.0 \pm 0.9$ | $33.9 \pm 1.2$ | $78.3 \pm 12.4$ | $143.9 \pm 8.4$ |
| ionosphere | acc | $.875 \pm .048$ | $.903 \pm .055$ | $.914 \pm .036$ | $.906 \pm .060$ | $.903 \pm .045$ | $.912 \pm .049$ | $.897 \pm .050$ |
| | nodes | $1.0 \pm 0.0$ | $1.9 \pm 0.3$ | $3.7 \pm 1.4$ | $10.5 \pm 2.4$ | $17.1 \pm 4.1$ | $22.3 \pm 3.3$ | $21.3 \pm 3.0$ |
| | path len | $1.00 \pm 0.00$ | $1.64 \pm 0.22$ | $2.42 \pm 0.49$ | $4.20 \pm 0.86$ | $6.03 \pm 1.07$ | $7.26 \pm 1.46$ | $7.16 \pm 1.34$ |
| | gini | $.750 \pm .009$ | $.767 \pm .040$ | $.769 \pm .031$ | $.768 \pm .038$ | $.783 \pm .042$ | $.774 \pm .045$ | $.787 \pm .045$ |
| | time | $0.7 \pm 0.0$ | $1.0 \pm 0.1$ | $1.3 \pm 0.3$ | $2.9 \pm 0.6$ | $4.5 \pm 1.0$ | $14.4 \pm 2.0$ | $7.4 \pm 1.0$ |

Ours: prototype features, L1 regularization, non-random initialization, crisp

| data | metric | 1e-1 | 3e-2 | 1e-2 | 3e-3 | 1e-3 | 3e-4 | 1e-4 |
|------|--------|------|------|------|------|------|------|------|
| iris | acc | $.940 \pm .055$ | $.940 \pm .055$ | $.933 \pm .067$ | $.927 \pm .055$ | $.927 \pm .063$ | $.900 \pm .086$ | $.933 \pm .067$ |
| | nodes | $2.0 \pm 0.0$ | $2.0 \pm 0.0$ | $3.3 \pm 0.8$ | $5.3 \pm 0.6$ | $5.9 \pm 0.9$ | $6.3 \pm 1.0$ | $6.1 \pm 0.8$ |
| | path len | $1.66 \pm 0.09$ | $1.66 \pm 0.09$ | $2.02 \pm 0.30$ | $2.27 \pm 0.22$ | $2.30 \pm 0.15$ | $2.35 \pm 0.18$ | $2.27 \pm 0.18$ |
| | gini | $.571 \pm .025$ | $.569 \pm .025$ | $.542 \pm .042$ | $.524 \pm .052$ | $.552 \pm .045$ | $.539 \pm .045$ | $.535 \pm .036$ |
| | time | $1.0 \pm 0.0$ | $1.0 \pm 0.0$ | $1.4 \pm 0.2$ | $1.9 \pm 0.2$ | $2.0 \pm 0.2$ | $2.1 \pm 0.3$ | $2.1 \pm 0.2$ |
| heart | acc | $.756 \pm .065$ | $.743 \pm .060$ | $.759 \pm .071$ | $.782 \pm .066$ | $.743 \pm .073$ | $.753 \pm .060$ | $.733 \pm .059$ |
| | nodes | $1.0 \pm 0.0$ | $1.3 \pm 0.6$ | $7.0 \pm 2.2$ | $27.3 \pm 5.2$ | $46.9 \pm 6.7$ | $52.4 \pm 6.2$ | $54.1 \pm 6.2$ |
| | path len | $1.00 \pm 0.00$ | $1.14 \pm 0.29$ | $2.88 \pm 0.57$ | $4.73 \pm 0.68$ | $5.60 \pm 0.69$ | $6.27 \pm 0.53$ | $6.34 \pm 0.62$ |
| | gini | $.912 \pm .009$ | $.908 \pm .015$ | $.872 \pm .022$ | $.852 \pm .019$ | $.844 \pm .016$ | $.850 \pm .014$ | $.853 \pm .015$ |
| | time | $0.7 \pm 0.0$ | $0.8 \pm 0.2$ | $2.0 \pm 0.5$ | $6.1 \pm 1.2$ | $9.5 \pm 1.2$ | $10.6 \pm 1.2$ | $11.0 \pm 1.1$ |
| dry-bean | acc | $.713 \pm .086$ | $.863 \pm .007$ | $.870 \pm .016$ | $.876 \pm .011$ | $.886 \pm .010$ | $.890 \pm .008$ | $.891 \pm .009$ |
| | nodes | $3.6 \pm 0.7$ | $6.6 \pm 0.5$ | $8.5 \pm 0.9$ | $13.9 \pm 1.8$ | $26.0 \pm 2.1$ | $43.0 \pm 7.4$ | $97.5 \pm 12.4$ |
| | path len | $2.37 \pm 0.31$ | $3.17 \pm 0.17$ | $3.50 \pm 0.22$ | $4.23 \pm 0.29$ | $5.07 \pm 0.25$ | $5.45 \pm 0.31$ | $6.74 \pm 0.32$ |
| | gini | $.830 \pm .033$ | $.806 \pm .029$ | $.720 \pm .047$ | $.583 \pm .066$ | $.525 \pm .066$ | $.516 \pm .052$ | $.509 \pm .030$ |
| | time | $3.0 \pm 0.4$ | $4.6 \pm 0.3$ | $5.5 \pm 0.5$ | $7.7 \pm 0.8$ | $12.4 \pm 1.0$ | $65.5 \pm 21.2$ | $75.9 \pm 20.9$ |
| wine | acc | $.899 \pm .033$ | $.927 \pm .051$ | $.905 \pm .050$ | $.905 \pm .050$ | $.938 \pm .047$ | $.911 \pm .051$ | $.949 \pm .060$ |
| | nodes | $2.0 \pm 0.0$ | $2.4 \pm 0.5$ | $5.7 \pm 2.1$ | $7.4 \pm 1.9$ | $8.1 \pm 1.0$ | $9.2 \pm 2.4$ | $10.7 \pm 3.0$ |
| | path len | $1.66 \pm 0.12$ | $1.78 \pm 0.22$ | $2.77 \pm 0.64$ | $3.12 \pm 0.38$ | $3.19 \pm 0.22$ | $3.54 \pm 0.41$ | $3.70 \pm 0.49$ |
| | gini | $.815 \pm .043$ | $.811 \pm .054$ | $.730 \pm .056$ | $.697 \pm .050$ | $.680 \pm .075$ | $.684 \pm .069$ | $.682 \pm .055$ |
| | time | $0.9 \pm 0.1$ | $1.0 \pm 0.1$ | $1.7 \pm 0.4$ | $2.0 \pm 0.4$ | $2.1 \pm 0.2$ | $2.4 \pm 0.5$ | $2.6 \pm 0.6$ |
| car | acc | $.700 \pm .044$ | $.700 \pm .044$ | $.740 \pm .098$ | $.779 \pm .103$ | $.797 \pm .096$ | $.807 \pm .111$ | $.794 \pm .126$ |
| | nodes | $0.0 \pm 0.0$ | $0.3 \pm 0.5$ | $4.9 \pm 3.2$ | $15.4 \pm 5.8$ | $39.5 \pm 23.4$ | $83.5 \pm 57.5$ | $105.4 \pm 59.8$ |
| | path len | $0.00 \pm 0.00$ | $0.30 \pm 0.46$ | $2.43 \pm 1.05$ | $4.16 \pm 0.92$ | $5.10 \pm 0.92$ | $6.22 \pm 1.67$ | $6.62 \pm 1.34$ |
| | gini | $.000 \pm .000$ | $.266 \pm .407$ | $.896 \pm .015$ | $.889 \pm .012$ | $.879 \pm .014$ | $.870 \pm .014$ | $.871 \pm .012$ |
| | time | $0.3 \pm 0.0$ | $0.4 \pm 0.2$ | $1.7 \pm 0.8$ | $4.3 \pm 1.3$ | $9.3 \pm 4.9$ | $18.5 \pm 11.9$ | $22.3 \pm 11.6$ |
| wdbc | acc | $.937 \pm .029$ | $.942 \pm .029$ | $.945 \pm .032$ | $.928 \pm .041$ | $.930 \pm .037$ | $.923 \pm .036$ | $.931 \pm .046$ |
| | nodes | $1.0 \pm 0.0$ | $1.1 \pm 0.3$ | $2.9 \pm 0.8$ | $10.0 \pm 2.9$ | $18.2 \pm 1.6$ | $22.2 \pm 2.8$ | $25.7 \pm 2.5$ |
| | path len | $1.00 \pm 0.00$ | $1.04 \pm 0.13$ | $1.77 \pm 0.36$ | $3.38 \pm 0.61$ | $4.56 \pm 0.60$ | $5.03 \pm 0.85$ | $6.05 \pm 0.95$ |
| | gini | $.917 \pm .022$ | $.915 \pm .022$ | $.925 \pm .012$ | $.878 \pm .025$ | $.842 \pm .023$ | $.815 \pm .027$ | $.786 \pm .054$ |
| | time | $0.7 \pm 0.0$ | $0.7 \pm 0.1$ | $1.1 \pm 0.2$ | $2.7 \pm 0.7$ | $4.5 \pm 0.4$ | $5.4 \pm 0.7$ | $6.1 \pm 0.6$ |
| sonar | acc | $.716 \pm .102$ | $.735 \pm .085$ | $.769 \pm .060$ | $.754 \pm .080$ | $.759 \pm .121$ | $.778 \pm .076$ | $.763 \pm .074$ |
| | nodes | $1.0 \pm 0.0$ | $3.9 \pm 0.7$ | $11.7 \pm 1.5$ | $19.2 \pm 2.7$ | $18.7 \pm 2.1$ | $18.9 \pm 2.2$ | $19.5 \pm 2.2$ |
| | path len | $1.00 \pm 0.00$ | $2.24 \pm 0.20$ | $4.04 \pm 0.46$ | $5.23 \pm 0.56$ | $5.12 \pm 0.63$ | $5.07 \pm 0.66$ | $5.08 \pm 0.75$ |
| | gini | $.888 \pm .069$ | $.921 \pm .035$ | $.890 \pm .036$ | $.875 \pm .034$ | $.891 \pm .020$ | $.888 \pm .045$ | $.886 \pm .025$ |
| | time | $0.7 \pm 0.0$ | $1.3 \pm 0.2$ | $2.9 \pm 0.3$ | $4.4 \pm 0.6$ | $4.3 \pm 0.5$ | $4.3 \pm 0.5$ | $4.5 \pm 0.5$ |
| pendigits | acc | $.094 \pm .003$ | $.718 \pm .041$ | $.790 \pm .024$ | $.868 \pm .018$ | $.901 \pm .012$ | $.921 \pm .011$ | $.930 \pm .006$ |
| | nodes | $0.0 \pm 0.0$ | $8.4 \pm 0.9$ | $15.0 \pm 1.6$ | $34.3 \pm 2.2$ | $78.0 \pm 3.3$ | $180.0 \pm 12.1$ | $408.8 \pm 21.5$ |
| | path len | $0.00 \pm 0.00$ | $3.81 \pm 0.27$ | $4.46 \pm 0.26$ | $5.79 \pm 0.17$ | $6.77 \pm 0.26$ | $8.06 \pm 0.30$ | $9.26 \pm 0.19$ |
| | gini | $.000 \pm .000$ | $.527 \pm .070$ | $.407 \pm .104$ | $.317 \pm .071$ | $.294 \pm .063$ | $.295 \pm .047$ | $.288 \pm .036$ |
| | time | $0.4 \pm 0.0$ | $5.2 \pm 0.5$ | $6.3 \pm 0.5$ | $11.6 \pm 0.6$ | $22.3 \pm 0.8$ | $55.3 \pm 3.1$ | $114.9 \pm 6.4$ |
| ionosphere | acc | $.875 \pm .048$ | $.903 \pm .055$ | $.917 \pm .037$ | $.909 \pm .058$ | $.906 \pm .041$ | $.912 \pm .049$ | $.897 \pm .050$ |
| | nodes | $1.0 \pm 0.0$ | $1.9 \pm 0.3$ | $3.7 \pm 1.4$ | $10.5 \pm 2.4$ | $17.1 \pm 4.1$ | $22.3 \pm 3.3$ | $21.3 \pm 3.0$ |
| | path len | $1.00 \pm 0.00$ | $1.64 \pm 0.22$ | $2.42 \pm 0.50$ | $4.20 \pm 0.87$ | $6.05 \pm 1.09$ | $7.22 \pm 1.46$ | $7.18 \pm 1.35$ |
| | gini | $.750 \pm .009$ | $.767 \pm .040$ | $.769 \pm .031$ | $.768 \pm .038$ | $.783 \pm .042$ | $.774 \pm .045$ | $.787 \pm .045$ |
| | time | $0.7 \pm 0.0$ | $0.9 \pm 0.1$ | $1.2 \pm 0.3$ | $2.6 \pm 0.5$ | $4.1 \pm 0.9$ | $6.8 \pm 0.9$ | $6.6 \pm 0.9$ |

Ours: linear features, L2 regularization, random initialization, fuzzy

| data | metric | 1e-1 | 3e-2 | 1e-2 | 3e-3 | 1e-3 | 3e-4 | 1e-4 |
|------|--------|------|------|------|------|------|------|------|
| iris | acc | .960 ± .044 | .947 ± .065 | .960 ± .053 | .947 ± .050 | .953 ± .043 | .960 ± .033 | .980 ± .031 |
| | nodes | 2.0 ± 0.0 | 2.8 ± 0.4 | 2.4 ± 0.5 | 7.0 ± 1.9 | 10.6 ± 0.9 | 10.9 ± 1.5 | 19.0 ± 2.2 |
| | path len | 1.67 ± 0.09 | 1.96 ± 0.23 | 1.79 ± 0.20 | 2.59 ± 0.31 | 2.91 ± 0.42 | 2.86 ± 0.28 | 3.82 ± 0.46 |
| | gini | .627 ± .005 | .602 ± .048 | .611 ± .018 | .681 ± .036 | .681 ± .031 | .601 ± .045 | .633 ± .036 |
| | time | 0.6 ± 0.0 | 0.8 ± 0.1 | 0.7 ± 0.1 | 1.8 ± 0.4 | 2.8 ± 0.2 | 2.9 ± 0.3 | 4.7 ± 0.5 |
| heart | acc | .819 ± .048 | .802 ± .066 | .772 ± .048 | .753 ± .078 | .756 ± .066 | .763 ± .039 | .743 ± .063 |
| | nodes | 1.0 ± 0.0 | 1.2 ± 0.4 | 6.2 ± 0.7 | 19.7 ± 2.2 | 22.6 ± 2.5 | 25.9 ± 2.5 | 26.1 ± 3.6 |
| | path len | 1.00 ± 0.00 | 1.10 ± 0.21 | 3.12 ± 0.46 | 4.76 ± 0.98 | 5.82 ± 0.53 | 7.00 ± 0.70 | 6.12 ± 0.82 |
| | gini | .924 ± .006 | .925 ± .004 | .920 ± .008 | .921 ± .004 | .919 ± .003 | .919 ± .003 | .920 ± .004 |
| | time | 0.5 ± 0.0 | 0.6 ± 0.1 | 1.9 ± 0.2 | 5.1 ± 0.5 | 4.2 ± 0.4 | 4.7 ± 0.4 | 4.8 ± 0.6 |
| dry-bean | acc | .668 ± .012 | .906 ± .008 | .917 ± .006 | .918 ± .005 | .920 ± .006 | .923 ± .007 | .924 ± .005 |
| | nodes | 3.0 ± 0.0 | 6.0 ± 0.0 | 8.0 ± 0.0 | 10.0 ± 1.3 | 19.9 ± 1.4 | 28.4 ± 5.4 | 66.0 ± 7.9 |
| | path len | 2.00 ± 0.00 | 2.95 ± 0.11 | 3.38 ± 0.04 | 3.71 ± 0.29 | 5.14 ± 0.43 | 4.88 ± 0.46 | 6.05 ± 0.37 |
| | gini | .907 ± .005 | .905 ± .003 | .908 ± .002 | .908 ± .003 | .908 ± .004 | .906 ± .003 | .908 ± .002 |
| | time | 2.6 ± 0.1 | 4.3 ± 0.1 | 5.4 ± 0.1 | 6.4 ± 0.7 | 11.3 ± 0.8 | 23.8 ± 6.3 | 66.0 ± 9.6 |
| wine | acc | .966 ± .027 | .961 ± .026 | .960 ± .026 | .960 ± .026 | .966 ± .028 | .966 ± .027 | .966 ± .028 |
| | nodes | 2.0 ± 0.0 | 2.0 ± 0.0 | 2.1 ± 0.3 | 2.2 ± 0.4 | 2.1 ± 0.3 | 2.1 ± 0.3 | 2.4 ± 0.7 |
| | path len | 1.66 ± 0.10 | 1.66 ± 0.10 | 1.69 ± 0.14 | 1.72 ± 0.17 | 1.69 ± 0.14 | 1.68 ± 0.14 | 1.79 ± 0.25 |
| | gini | .872 ± .009 | .873 ± .009 | .874 ± .007 | .876 ± .010 | .873 ± .010 | .873 ± .011 | .876 ± .013 |
| | time | 0.6 ± 0.0 | 0.6 ± 0.0 | 0.6 ± 0.1 | 0.6 ± 0.1 | 0.6 ± 0.0 | 0.6 ± 0.1 | 0.6 ± 0.1 |
| car | acc | .700 ± .044 | .910 ± .025 | .940 ± .022 | .983 ± .011 | .993 ± .009 | .989 ± .011 | .991 ± .006 |
| | nodes | 0.0 ± 0.0 | 2.0 ± 0.0 | 4.1 ± 0.9 | 10.1 ± 1.8 | 17.6 ± 2.2 | 26.4 ± 4.8 | 25.8 ± 5.6 |
| | path len | 0.00 ± 0.00 | 1.32 ± 0.04 | 1.68 ± 0.18 | 2.47 ± 0.21 | 3.04 ± 0.50 | 4.33 ± 0.86 | 3.99 ± 0.85 |
| | gini | .000 ± .000 | .921 ± .001 | .914 ± .008 | .905 ± .012 | .904 ± .007 | .915 ± .011 | .909 ± .009 |
| | time | 0.2 ± 0.0 | 0.9 ± 0.0 | 1.5 ± 0.3 | 3.0 ± 0.5 | 4.9 ± 0.6 | 7.4 ± 1.3 | 7.2 ± 1.5 |
| wdbc | acc | .963 ± .025 | .965 ± .024 | .967 ± .025 | .961 ± .023 | .963 ± .025 | .961 ± .030 | .968 ± .028 |
| | nodes | 1.0 ± 0.0 | 1.0 ± 0.0 | 1.0 ± 0.0 | 2.5 ± 1.4 | 5.2 ± 1.4 | 13.1 ± 2.8 | 19.6 ± 4.4 |
| | path len | 1.00 ± 0.00 | 1.00 ± 0.00 | 1.00 ± 0.00 | 1.54 ± 0.40 | 2.42 ± 0.38 | 3.53 ± 0.51 | 4.51 ± 0.63 |
| | gini | .955 ± .003 | .955 ± .002 | .955 ± .003 | .953 ± .002 | .951 ± .003 | .951 ± .002 | .952 ± .002 |
| | time | 0.5 ± 0.0 | 0.5 ± 0.0 | 0.5 ± 0.0 | 1.0 ± 0.4 | 1.7 ± 0.4 | 3.7 ± 0.8 | 5.3 ± 1.1 |
| sonar | acc | .736 ± .109 | .788 ± .070 | .807 ± .058 | .807 ± .089 | .793 ± .118 | .860 ± .034 | .822 ± .081 |
| | nodes | 1.0 ± 0.0 | 2.3 ± 0.5 | 4.8 ± 1.2 | 5.5 ± 1.0 | 5.7 ± 1.4 | 6.4 ± 2.2 | 6.5 ± 1.9 |
| | path len | 1.00 ± 0.00 | 1.65 ± 0.21 | 2.65 ± 0.55 | 2.82 ± 0.54 | 2.81 ± 0.50 | 3.14 ± 0.69 | 3.16 ± 0.69 |
| | gini | .976 ± .001 | .975 ± .001 | .975 ± .001 | .974 ± .001 | .974 ± .001 | .974 ± .001 | .974 ± .001 |
| | time | 0.5 ± 0.0 | 0.9 ± 0.1 | 1.5 ± 0.3 | 1.7 ± 0.3 | 1.8 ± 0.4 | 1.9 ± 0.5 | 2.0 ± 0.4 |
| pendigits | acc | .094 ± .003 | .906 ± .025 | .948 ± .012 | .977 ± .007 | .983 ± .004 | .989 ± .003 | .989 ± .003 |
| | nodes | 0.0 ± 0.0 | 9.2 ± 0.6 | 12.4 ± 1.6 | 20.1 ± 2.0 | 31.5 ± 3.2 | 56.1 ± 6.0 | 123.5 ± 10.5 |
| | path len | 0.00 ± 0.00 | 3.99 ± 0.12 | 4.74 ± 0.45 | 5.01 ± 0.42 | 5.80 ± 0.29 | 6.33 ± 0.36 | 8.10 ± 0.53 |
| | gini | .000 ± .000 | .908 ± .001 | .900 ± .006 | .899 ± .005 | .898 ± .004 | .900 ± .005 | .897 ± .004 |
| | time | 0.3 ± 0.0 | 4.6 ± 0.2 | 5.9 ± 0.6 | 8.6 ± 0.7 | 12.2 ± 1.0 | 25.6 ± 2.9 | 47.6 ± 4.0 |
| ionosphere | acc | .843 ± .093 | .909 ± .040 | .912 ± .039 | .923 ± .048 | .926 ± .039 | .935 ± .040 | .926 ± .043 |
| | nodes | 1.0 ± 0.0 | 2.8 ± 0.4 | 3.6 ± 0.7 | 4.5 ± 0.7 | 8.2 ± 1.8 | 12.4 ± 2.9 | 13.5 ± 3.6 |
| | path len | 1.00 ± 0.00 | 2.25 ± 0.26 | 2.75 ± 0.49 | 3.28 ± 0.33 | 3.96 ± 0.61 | 5.22 ± 0.88 | 5.13 ± 0.85 |
| | gini | .952 ± .004 | .934 ± .007 | .927 ± .010 | .925 ± .004 | .933 ± .011 | .935 ± .007 | .938 ± .005 |
| | time | 0.5 ± 0.0 | 1.0 ± 0.1 | 1.2 ± 0.2 | 1.4 ± 0.2 | 2.3 ± 0.5 | 3.3 ± 0.7 | 3.6 ± 1.0 |

Ours: linear features, L2 regularization, random initialization, crisp

| data | metric | 1e-1 | 3e-2 | 1e-2 | 3e-3 | 1e-3 | 3e-4 | 1e-4 |
|------|--------|------|------|------|------|------|------|------|
| iris | acc | $.967 \pm .045$ | $.953 \pm .067$ | $.960 \pm .053$ | $.953 \pm .052$ | $.953 \pm .043$ | $.967 \pm .033$ | $.980 \pm .031$ |
| | nodes | $2.0 \pm 0.0$ | $2.8 \pm 0.4$ | $2.4 \pm 0.5$ | $7.0 \pm 1.9$ | $10.6 \pm 0.9$ | $10.9 \pm 1.5$ | $19.0 \pm 2.2$ |
| | path len | $1.67 \pm 0.09$ | $1.95 \pm 0.23$ | $1.79 \pm 0.20$ | $2.62 \pm 0.34$ | $2.95 \pm 0.50$ | $2.90 \pm 0.26$ | $3.82 \pm 0.48$ |
| | gini | $.627 \pm .005$ | $.602 \pm .048$ | $.611 \pm .018$ | $.680 \pm .037$ | $.681 \pm .030$ | $.599 \pm .044$ | $.633 \pm .036$ |
| | time | $0.7 \pm 0.0$ | $0.9 \pm 0.1$ | $0.8 \pm 0.1$ | $1.8 \pm 0.4$ | $2.5 \pm 0.2$ | $2.5 \pm 0.3$ | $4.1 \pm 0.4$ |
| heart | acc | $.815 \pm .051$ | $.802 \pm .066$ | $.762 \pm .046$ | $.753 \pm .070$ | $.753 \pm .068$ | $.762 \pm .053$ | $.746 \pm .051$ |
| | nodes | $1.0 \pm 0.0$ | $1.2 \pm 0.4$ | $6.2 \pm 0.7$ | $19.7 \pm 2.2$ | $22.6 \pm 2.5$ | $25.9 \pm 2.5$ | $26.1 \pm 3.6$ |
| | path len | $1.00 \pm 0.00$ | $1.11 \pm 0.22$ | $3.13 \pm 0.46$ | $4.78 \pm 0.99$ | $5.86 \pm 0.54$ | $7.10 \pm 0.78$ | $6.16 \pm 0.79$ |
| | gini | $.924 \pm .006$ | $.925 \pm .004$ | $.919 \pm .008$ | $.921 \pm .004$ | $.919 \pm .003$ | $.919 \pm .003$ | $.920 \pm .003$ |
| | time | $0.5 \pm 0.0$ | $0.6 \pm 0.1$ | $1.7 \pm 0.2$ | $4.5 \pm 0.5$ | $3.7 \pm 0.4$ | $4.2 \pm 0.4$ | $4.2 \pm 0.6$ |
| dry-bean | acc | $.673 \pm .011$ | $.903 \pm .009$ | $.914 \pm .006$ | $.915 \pm .007$ | $.913 \pm .008$ | $.920 \pm .008$ | $.920 \pm .006$ |
| | nodes | $3.0 \pm 0.0$ | $6.0 \pm 0.0$ | $8.0 \pm 0.0$ | $10.0 \pm 1.3$ | $19.9 \pm 1.4$ | $28.4 \pm 5.4$ | $66.0 \pm 7.9$ |
| | path len | $2.00 \pm 0.00$ | $2.95 \pm 0.12$ | $3.38 \pm 0.04$ | $3.72 \pm 0.30$ | $5.17 \pm 0.44$ | $4.90 \pm 0.48$ | $6.09 \pm 0.39$ |
| | gini | $.907 \pm .005$ | $.905 \pm .003$ | $.908 \pm .002$ | $.908 \pm .003$ | $.908 \pm .004$ | $.906 \pm .003$ | $.908 \pm .002$ |
| | time | $2.5 \pm 0.1$ | $4.0 \pm 0.1$ | $4.9 \pm 0.1$ | $5.8 \pm 0.6$ | $9.9 \pm 0.7$ | $36.6 \pm 13.6$ | $38.4 \pm 9.7$ |
| wine | acc | $.966 \pm .027$ | $.961 \pm .026$ | $.960 \pm .026$ | $.960 \pm .026$ | $.966 \pm .028$ | $.966 \pm .027$ | $.966 \pm .028$ |
| | nodes | $2.0 \pm 0.0$ | $2.0 \pm 0.0$ | $2.1 \pm 0.3$ | $2.2 \pm 0.4$ | $2.1 \pm 0.3$ | $2.1 \pm 0.3$ | $2.4 \pm 0.7$ |
| | path len | $1.66 \pm 0.10$ | $1.66 \pm 0.10$ | $1.68 \pm 0.14$ | $1.72 \pm 0.17$ | $1.68 \pm 0.14$ | $1.69 \pm 0.14$ | $1.79 \pm 0.25$ |
| | gini | $.872 \pm .009$ | $.873 \pm .009$ | $.874 \pm .007$ | $.876 \pm .010$ | $.873 \pm .010$ | $.873 \pm .011$ | $.876 \pm .013$ |
| | time | $0.5 \pm 0.0$ | $0.5 \pm 0.0$ | $0.5 \pm 0.0$ | $0.5 \pm 0.1$ | $0.5 \pm 0.1$ | $0.5 \pm 0.1$ | $0.6 \pm 0.1$ |
| car | acc | $.700 \pm .044$ | $.897 \pm .026$ | $.937 \pm .025$ | $.983 \pm .011$ | $.993 \pm .009$ | $.989 \pm .011$ | $.991 \pm .006$ |
| | nodes | $0.0 \pm 0.0$ | $2.0 \pm 0.0$ | $4.1 \pm 0.9$ | $10.1 \pm 1.8$ | $17.6 \pm 2.2$ | $26.4 \pm 4.8$ | $25.8 \pm 5.6$ |
| | path len | $0.00 \pm 0.00$ | $1.32 \pm 0.04$ | $1.68 \pm 0.18$ | $2.48 \pm 0.22$ | $3.05 \pm 0.50$ | $4.33 \pm 0.85$ | $3.98 \pm 0.84$ |
| | gini | $.000 \pm .000$ | $.921 \pm .001$ | $.914 \pm .008$ | $.905 \pm .011$ | $.904 \pm .007$ | $.915 \pm .011$ | $.909 \pm .009$ |
| | time | $0.2 \pm 0.0$ | $0.8 \pm 0.0$ | $1.3 \pm 0.2$ | $2.7 \pm 0.4$ | $4.4 \pm 0.5$ | $6.5 \pm 1.1$ | $6.3 \pm 1.3$ |
| wdbc | acc | $.961 \pm .025$ | $.963 \pm .023$ | $.967 \pm .025$ | $.961 \pm .025$ | $.960 \pm .022$ | $.961 \pm .025$ | $.963 \pm .025$ |
| | nodes | $1.0 \pm 0.0$ | $1.0 \pm 0.0$ | $1.0 \pm 0.0$ | $2.5 \pm 1.4$ | $5.2 \pm 1.4$ | $13.1 \pm 2.8$ | $19.6 \pm 4.4$ |
| | path len | $1.00 \pm 0.00$ | $1.00 \pm 0.00$ | $1.00 \pm 0.00$ | $1.54 \pm 0.41$ | $2.42 \pm 0.38$ | $3.53 \pm 0.53$ | $4.53 \pm 0.62$ |
| | gini | $.955 \pm .003$ | $.955 \pm .002$ | $.955 \pm .003$ | $.953 \pm .002$ | $.951 \pm .003$ | $.951 \pm .002$ | $.952 \pm .002$ |
| | time | $0.5 \pm 0.0$ | $0.5 \pm 0.0$ | $0.5 \pm 0.0$ | $0.9 \pm 0.3$ | $1.5 \pm 0.3$ | $3.2 \pm 0.7$ | $4.6 \pm 0.9$ |
| sonar | acc | $.731 \pm .111$ | $.783 \pm .067$ | $.807 \pm .058$ | $.803 \pm .093$ | $.789 \pm .115$ | $.851 \pm .045$ | $.817 \pm .077$ |
| | nodes | $1.0 \pm 0.0$ | $2.3 \pm 0.5$ | $4.8 \pm 1.2$ | $5.5 \pm 1.0$ | $5.7 \pm 1.4$ | $6.4 \pm 2.2$ | $6.5 \pm 1.9$ |
| | path len | $1.00 \pm 0.00$ | $1.65 \pm 0.22$ | $2.65 \pm 0.56$ | $2.79 \pm 0.56$ | $2.81 \pm 0.50$ | $3.15 \pm 0.67$ | $3.16 \pm 0.70$ |
| | gini | $.976 \pm .001$ | $.975 \pm .001$ | $.975 \pm .001$ | $.974 \pm .001$ | $.974 \pm .001$ | $.974 \pm .001$ | $.974 \pm .001$ |
| | time | $0.5 \pm 0.0$ | $0.9 \pm 0.1$ | $1.4 \pm 0.2$ | $1.6 \pm 0.2$ | $1.6 \pm 0.3$ | $1.7 \pm 0.4$ | $1.8 \pm 0.4$ |
| pendigits | acc | $.094 \pm .003$ | $.905 \pm .025$ | $.946 \pm .012$ | $.974 \pm .007$ | $.980 \pm .004$ | $.987 \pm .002$ | $.989 \pm .003$ |
| | nodes | $0.0 \pm 0.0$ | $9.2 \pm 0.6$ | $12.4 \pm 1.6$ | $20.1 \pm 2.0$ | $31.5 \pm 3.2$ | $56.1 \pm 6.0$ | $123.5 \pm 10.5$ |
| | path len | $0.00 \pm 0.00$ | $3.99 \pm 0.13$ | $4.74 \pm 0.45$ | $5.01 \pm 0.42$ | $5.80 \pm 0.28$ | $6.33 \pm 0.36$ | $8.12 \pm 0.53$ |
| | gini | $.000 \pm .000$ | $.908 \pm .001$ | $.900 \pm .006$ | $.899 \pm .005$ | $.898 \pm .004$ | $.900 \pm .005$ | $.898 \pm .004$ |
| | time | $0.3 \pm 0.0$ | $4.2 \pm 0.2$ | $5.3 \pm 0.6$ | $7.7 \pm 0.6$ | $10.8 \pm 0.9$ | $20.8 \pm 2.1$ | $41.1 \pm 3.3$ |
| ionosphere | acc | $.843 \pm .093$ | $.915 \pm .042$ | $.914 \pm .042$ | $.932 \pm .053$ | $.926 \pm .043$ | $.929 \pm .041$ | $.923 \pm .051$ |
| | nodes | $1.0 \pm 0.0$ | $2.8 \pm 0.4$ | $3.6 \pm 0.7$ | $4.5 \pm 0.7$ | $8.2 \pm 1.8$ | $12.4 \pm 2.9$ | $13.5 \pm 3.6$ |
| | path len | $1.00 \pm 0.00$ | $2.26 \pm 0.26$ | $2.79 \pm 0.53$ | $3.35 \pm 0.37$ | $4.02 \pm 0.63$ | $5.30 \pm 0.91$ | $5.22 \pm 0.87$ |
| | gini | $.952 \pm .004$ | $.934 \pm .007$ | $.928 \pm .010$ | $.925 \pm .004$ | $.934 \pm .011$ | $.935 \pm .007$ | $.938 \pm .005$ |
| | time | $0.5 \pm 0.0$ | $0.9 \pm 0.1$ | $1.1 \pm 0.1$ | $1.3 \pm 0.1$ | $2.0 \pm 0.4$ | $3.0 \pm 0.6$ | $3.3 \pm 1.0$ |

Ours: prototype features, L2 regularization, random initialization, fuzzy

| data | metric | 1e-1 | 3e-2 | 1e-2 | 3e-3 | 1e-3 | 3e-4 | 1e-4 |
|------|--------|------|------|------|------|------|------|------|
| iris | acc | $.953 \pm .043$ | $.947 \pm .065$ | $.947 \pm .050$ | $.953 \pm .043$ | $.953 \pm .043$ | $.953 \pm .052$ | $.947 \pm .040$ |
| | nodes | $2.0 \pm 0.0$ | $2.1 \pm 0.3$ | $3.2 \pm 1.1$ | $5.6 \pm 1.9$ | $10.5 \pm 2.9$ | $17.4 \pm 3.1$ | $22.1 \pm 3.6$ |
| | path len | $1.67 \pm 0.09$ | $1.71 \pm 0.17$ | $2.01 \pm 0.30$ | $2.52 \pm 0.24$ | $2.96 \pm 0.49$ | $3.49 \pm 0.34$ | $3.97 \pm 0.57$ |
| | gini | $.674 \pm .041$ | $.666 \pm .018$ | $.670 \pm .041$ | $.664 \pm .027$ | $.670 \pm .032$ | $.670 \pm .033$ | $.679 \pm .032$ |
| | time | $1.1 \pm 0.0$ | $1.2 \pm 0.2$ | $1.5 \pm 0.3$ | $2.2 \pm 0.6$ | $5.3 \pm 1.3$ | $8.4 \pm 1.4$ | $7.0 \pm 1.1$ |
| heart | acc | $.809 \pm .038$ | $.812 \pm .035$ | $.795 \pm .041$ | $.769 \pm .034$ | $.762 \pm .059$ | $.749 \pm .061$ | $.743 \pm .035$ |
| | nodes | $1.0 \pm 0.0$ | $1.0 \pm 0.0$ | $4.5 \pm 0.8$ | $13.9 \pm 2.0$ | $37.8 \pm 3.7$ | $60.1 \pm 5.3$ | $85.7 \pm 5.1$ |
| | path len | $1.00 \pm 0.00$ | $1.00 \pm 0.00$ | $2.40 \pm 0.32$ | $4.00 \pm 0.43$ | $5.30 \pm 0.58$ | $6.49 \pm 0.53$ | $7.67 \pm 0.53$ |
| | gini | $.935 \pm .003$ | $.937 \pm .003$ | $.931 \pm .004$ | $.916 \pm .005$ | $.904 \pm .006$ | $.894 \pm .008$ | $.894 \pm .008$ |
| | time | $0.7 \pm 0.1$ | $0.7 \pm 0.1$ | $1.6 \pm 0.2$ | $3.8 \pm 0.4$ | $9.3 \pm 0.7$ | $14.2 \pm 1.2$ | $20.1 \pm 1.2$ |
| dry-bean | acc | $.665 \pm .045$ | $.875 \pm .037$ | $.905 \pm .011$ | $.911 \pm .008$ | $.913 \pm .006$ | $.913 \pm .009$ | $.915 \pm .007$ |
| | nodes | $2.9 \pm 0.3$ | $5.6 \pm 0.5$ | $7.3 \pm 0.8$ | $14.1 \pm 1.5$ | $31.1 \pm 2.5$ | $56.0 \pm 6.0$ | $127.0 \pm 9.0$ |
| | path len | $2.04 \pm 0.26$ | $2.77 \pm 0.11$ | $3.25 \pm 0.26$ | $4.17 \pm 0.30$ | $5.23 \pm 0.27$ | $5.94 \pm 0.33$ | $7.56 \pm 0.53$ |
| | gini | $.881 \pm .004$ | $.871 \pm .005$ | $.855 \pm .017$ | $.784 \pm .027$ | $.708 \pm .040$ | $.647 \pm .041$ | $.559 \pm .056$ |
| | time | $2.9 \pm 0.2$ | $4.4 \pm 0.3$ | $5.4 \pm 0.5$ | $9.1 \pm 0.8$ | $17.3 \pm 1.2$ | $49.4 \pm 7.9$ | $67.0 \pm 9.7$ |
| wine | acc | $.955 \pm .033$ | $.978 \pm .027$ | $.967 \pm .037$ | $.960 \pm .036$ | $.961 \pm .050$ | $.956 \pm .042$ | $.949 \pm .039$ |
| | nodes | $2.0 \pm 0.0$ | $2.0 \pm 0.0$ | $2.6 \pm 0.5$ | $4.9 \pm 1.6$ | $6.9 \pm 2.4$ | $12.2 \pm 4.4$ | $18.6 \pm 6.7$ |
| | path len | $1.66 \pm 0.10$ | $1.66 \pm 0.09$ | $1.90 \pm 0.25$ | $2.51 \pm 0.41$ | $2.91 \pm 0.47$ | $3.75 \pm 0.72$ | $4.83 \pm 0.80$ |
| | gini | $.892 \pm .004$ | $.886 \pm .005$ | $.875 \pm .011$ | $.842 \pm .033$ | $.825 \pm .032$ | $.802 \pm .050$ | $.758 \pm .031$ |
| | time | $0.9 \pm 0.0$ | $0.9 \pm 0.0$ | $1.1 \pm 0.2$ | $1.6 \pm 0.4$ | $2.1 \pm 0.5$ | $3.3 \pm 1.0$ | $4.7 \pm 1.5$ |
| car | acc | $.700 \pm .044$ | $.801 \pm .060$ | $.910 \pm .028$ | $.956 \pm .018$ | $.975 \pm .016$ | $.985 \pm .015$ | $.992 \pm .009$ |
| | nodes | $0.0 \pm 0.0$ | $0.7 \pm 0.5$ | $4.5 \pm 1.7$ | $13.1 \pm 4.5$ | $24.4 \pm 5.2$ | $39.8 \pm 5.9$ | $55.1 \pm 8.3$ |
| | path len | $0.00 \pm 0.00$ | $0.70 \pm 0.46$ | $2.00 \pm 0.30$ | $2.88 \pm 0.50$ | $3.36 \pm 0.40$ | $3.80 \pm 0.63$ | $3.65 \pm 0.49$ |
| | gini | $.000 \pm .000$ | $.644 \pm .422$ | $.896 \pm .011$ | $.875 \pm .009$ | $.839 \pm .020$ | $.803 \pm .034$ | $.775 \pm .036$ |
| | time | $0.3 \pm 0.0$ | $0.7 \pm 0.2$ | $1.8 \pm 0.6$ | $3.9 \pm 1.1$ | $6.7 \pm 1.3$ | $10.2 \pm 1.3$ | $13.6 \pm 2.0$ |
| wdbc | acc | $.967 \pm .012$ | $.979 \pm .017$ | $.968 \pm .013$ | $.968 \pm .022$ | $.965 \pm .026$ | $.960 \pm .029$ | $.958 \pm .033$ |
| | nodes | $1.0 \pm 0.0$ | $1.0 \pm 0.0$ | $1.8 \pm 0.6$ | $4.3 \pm 1.0$ | $11.6 \pm 2.5$ | $30.4 \pm 4.2$ | $52.9 \pm 6.4$ |
| | path len | $1.00 \pm 0.00$ | $1.00 \pm 0.00$ | $1.42 \pm 0.33$ | $2.44 \pm 0.36$ | $3.80 \pm 0.47$ | $5.09 \pm 0.77$ | $6.08 \pm 0.48$ |
| | gini | $.942 \pm .002$ | $.942 \pm .001$ | $.931 \pm .010$ | $.930 \pm .011$ | $.902 \pm .015$ | $.845 \pm .019$ | $.841 \pm .031$ |
| | time | $0.7 \pm 0.0$ | $0.7 \pm 0.0$ | $1.0 \pm 0.2$ | $1.5 \pm 0.2$ | $3.3 \pm 0.7$ | $8.0 \pm 1.1$ | $13.3 \pm 1.6$ |
| sonar | acc | $.534 \pm .144$ | $.821 \pm .074$ | $.889 \pm .058$ | $.826 \pm .052$ | $.822 \pm .060$ | $.860 \pm .094$ | $.812 \pm .057$ |
| | nodes | $0.0 \pm 0.0$ | $2.6 \pm 0.5$ | $7.8 \pm 1.2$ | $12.3 \pm 1.7$ | $14.7 \pm 2.5$ | $20.4 \pm 7.4$ | $30.2 \pm 7.4$ |
| | path len | $0.00 \pm 0.00$ | $1.99 \pm 0.33$ | $3.69 \pm 0.45$ | $4.54 \pm 0.41$ | $4.90 \pm 0.44$ | $5.69 \pm 0.64$ | $6.40 \pm 1.05$ |
| | gini | $.000 \pm .000$ | $.972 \pm .001$ | $.970 \pm .001$ | $.964 \pm .003$ | $.952 \pm .009$ | $.953 \pm .007$ | $.934 \pm .012$ |
| | time | $0.3 \pm 0.0$ | $1.2 \pm 0.1$ | $2.3 \pm 0.3$ | $3.5 \pm 0.6$ | $4.0 \pm 0.7$ | $5.3 \pm 1.7$ | $8.1 \pm 2.3$ |
| pendigits | acc | $.094 \pm .003$ | $.855 \pm .026$ | $.898 \pm .011$ | $.913 \pm .014$ | $.935 \pm .009$ | $.949 \pm .008$ | $.962 \pm .005$ |
| | nodes | $0.0 \pm 0.0$ | $8.7 \pm 0.5$ | $11.7 \pm 1.0$ | $21.2 \pm 2.3$ | $55.0 \pm 6.0$ | $135.4 \pm 8.4$ | $347.3 \pm 25.0$ |
| | path len | $0.00 \pm 0.00$ | $4.81 \pm 0.38$ | $4.83 \pm 0.28$ | $5.05 \pm 0.42$ | $6.27 \pm 0.29$ | $7.58 \pm 0.26$ | $9.45 \pm 0.48$ |
| | gini | $.000 \pm .000$ | $.869 \pm .011$ | $.846 \pm .022$ | $.736 \pm .027$ | $.612 \pm .046$ | $.533 \pm .042$ | $.478 \pm .030$ |
| | time | $0.4 \pm 0.0$ | $5.0 \pm 0.2$ | $6.2 \pm 0.4$ | $9.6 \pm 0.9$ | $20.2 \pm 1.9$ | $57.6 \pm 9.5$ | $135.9 \pm 13.0$ |
| ionosphere | acc | $.914 \pm .057$ | $.915 \pm .059$ | $.920 \pm .046$ | $.926 \pm .062$ | $.909 \pm .049$ | $.920 \pm .033$ | $.917 \pm .047$ |
| | nodes | $1.5 \pm 0.5$ | $2.5 \pm 0.8$ | $4.2 \pm 1.2$ | $8.2 \pm 2.6$ | $16.4 \pm 1.8$ | $22.2 \pm 4.0$ | $22.9 \pm 3.4$ |
| | path len | $1.24 \pm 0.25$ | $1.65 \pm 0.40$ | $2.25 \pm 0.45$ | $3.20 \pm 0.54$ | $4.90 \pm 0.67$ | $5.58 \pm 1.09$ | $5.53 \pm 0.88$ |
| | gini | $.953 \pm .003$ | $.950 \pm .010$ | $.952 \pm .004$ | $.935 \pm .015$ | $.872 \pm .037$ | $.842 \pm .031$ | $.820 \pm .018$ |
| | time | $0.8 \pm 0.1$ | $1.0 \pm 0.2$ | $1.4 \pm 0.3$ | $2.4 \pm 0.7$ | $4.4 \pm 0.5$ | $9.9 \pm 7.4$ | $8.1 \pm 1.1$ |

Ours: prototype features, L2 regularization, random initialization, crisp

| data | metric | 1e-1 | 3e-2 | 1e-2 | 3e-3 | 1e-3 | 3e-4 | 1e-4 |
|------|--------|------|------|------|------|------|------|------|
| iris | acc | $.953 \pm .043$ | $.947 \pm .065$ | $.947 \pm .050$ | $.947 \pm .050$ | $.967 \pm .045$ | $.933 \pm .052$ | $.953 \pm .052$ |
| | nodes | $2.0 \pm 0.0$ | $2.1 \pm 0.3$ | $3.2 \pm 1.1$ | $5.6 \pm 1.9$ | $10.5 \pm 2.9$ | $17.4 \pm 3.1$ | $22.1 \pm 3.6$ |
| | path len | $1.67 \pm 0.09$ | $1.71 \pm 0.17$ | $2.00 \pm 0.29$ | $2.51 \pm 0.25$ | $2.90 \pm 0.46$ | $3.49 \pm 0.35$ | $3.95 \pm 0.57$ |
| | gini | $.674 \pm .041$ | $.666 \pm .017$ | $.670 \pm .041$ | $.664 \pm .027$ | $.670 \pm .032$ | $.669 \pm .032$ | $.678 \pm .032$ |
| | time | $1.0 \pm 0.0$ | $1.0 \pm 0.1$ | $1.3 \pm 0.3$ | $2.0 \pm 0.5$ | $3.2 \pm 0.7$ | $5.0 \pm 0.8$ | $6.1 \pm 1.0$ |
| heart | acc | $.805 \pm .044$ | $.809 \pm .040$ | $.782 \pm .043$ | $.753 \pm .044$ | $.749 \pm .067$ | $.733 \pm .042$ | $.716 \pm .040$ |
| | nodes | $1.0 \pm 0.0$ | $1.0 \pm 0.0$ | $4.5 \pm 0.8$ | $13.9 \pm 2.0$ | $37.8 \pm 3.7$ | $60.1 \pm 5.3$ | $85.7 \pm 5.1$ |
| | path len | $1.00 \pm 0.00$ | $1.00 \pm 0.00$ | $2.41 \pm 0.32$ | $4.03 \pm 0.48$ | $5.23 \pm 0.51$ | $6.49 \pm 0.62$ | $7.79 \pm 0.50$ |
| | gini | $.935 \pm .003$ | $.937 \pm .003$ | $.931 \pm .004$ | $.916 \pm .006$ | $.904 \pm .006$ | $.894 \pm .008$ | $.894 \pm .008$ |
| | time | $0.7 \pm 0.0$ | $0.7 \pm 0.0$ | $1.5 \pm 0.2$ | $3.4 \pm 0.4$ | $8.0 \pm 0.7$ | $12.2 \pm 1.0$ | $17.1 \pm 0.8$ |
| dry-bean | acc | $.665 \pm .039$ | $.872 \pm .037$ | $.904 \pm .012$ | $.903 \pm .008$ | $.907 \pm .006$ | $.909 \pm .006$ | $.906 \pm .010$ |
| | nodes | $2.9 \pm 0.3$ | $5.6 \pm 0.5$ | $7.3 \pm 0.8$ | $14.1 \pm 1.5$ | $31.1 \pm 2.5$ | $56.0 \pm 6.0$ | $127.0 \pm 9.0$ |
| | path len | $2.04 \pm 0.26$ | $2.77 \pm 0.12$ | $3.25 \pm 0.27$ | $4.18 \pm 0.31$ | $5.26 \pm 0.29$ | $5.96 \pm 0.38$ | $7.70 \pm 0.65$ |
| | gini | $.881 \pm .004$ | $.871 \pm .005$ | $.855 \pm .017$ | $.784 \pm .027$ | $.708 \pm .040$ | $.647 \pm .041$ | $.560 \pm .054$ |
| | time | $2.7 \pm 0.2$ | $4.0 \pm 0.2$ | $4.9 \pm 0.4$ | $7.8 \pm 0.6$ | $13.9 \pm 0.9$ | $24.5 \pm 7.9$ | $43.8 \pm 4.6$ |
| wine | acc | $.955 \pm .033$ | $.978 \pm .027$ | $.972 \pm .028$ | $.960 \pm .036$ | $.949 \pm .052$ | $.939 \pm .058$ | $.939 \pm .039$ |
| | nodes | $2.0 \pm 0.0$ | $2.0 \pm 0.0$ | $2.6 \pm 0.5$ | $4.9 \pm 1.6$ | $6.9 \pm 2.4$ | $12.2 \pm 4.4$ | $18.6 \pm 6.7$ |
| | path len | $1.66 \pm 0.11$ | $1.66 \pm 0.10$ | $1.90 \pm 0.25$ | $2.53 \pm 0.41$ | $2.93 \pm 0.49$ | $3.74 \pm 0.75$ | $4.89 \pm 0.88$ |
| | gini | $.892 \pm .004$ | $.886 \pm .005$ | $.875 \pm .011$ | $.842 \pm .033$ | $.825 \pm .032$ | $.802 \pm .051$ | $.757 \pm .031$ |
| | time | $0.9 \pm 0.1$ | $0.8 \pm 0.1$ | $1.0 \pm 0.1$ | $1.5 \pm 0.3$ | $1.9 \pm 0.5$ | $2.9 \pm 0.8$ | $4.1 \pm 1.2$ |
| car | acc | $.700 \pm .044$ | $.765 \pm .036$ | $.906 \pm .024$ | $.955 \pm .018$ | $.975 \pm .016$ | $.984 \pm .014$ | $.991 \pm .009$ |
| | nodes | $0.0 \pm 0.0$ | $0.7 \pm 0.5$ | $4.5 \pm 1.7$ | $13.1 \pm 4.5$ | $24.4 \pm 5.2$ | $39.8 \pm 5.9$ | $55.1 \pm 8.3$ |
| | path len | $0.00 \pm 0.00$ | $0.70 \pm 0.46$ | $2.00 \pm 0.30$ | $2.89 \pm 0.51$ | $3.35 \pm 0.42$ | $3.80 \pm 0.63$ | $3.65 \pm 0.48$ |
| | gini | $.000 \pm .000$ | $.644 \pm .422$ | $.896 \pm .011$ | $.875 \pm .009$ | $.839 \pm .020$ | $.803 \pm .034$ | $.775 \pm .036$ |
| | time | $0.3 \pm 0.0$ | $0.6 \pm 0.2$ | $1.6 \pm 0.4$ | $3.4 \pm 1.0$ | $6.0 \pm 1.0$ | $9.0 \pm 1.1$ | $11.8 \pm 1.7$ |
| wdbc | acc | $.967 \pm .012$ | $.979 \pm .017$ | $.968 \pm .013$ | $.968 \pm .022$ | $.951 \pm .022$ | $.954 \pm .030$ | $.947 \pm .029$ |
| | nodes | $1.0 \pm 0.0$ | $1.0 \pm 0.0$ | $1.8 \pm 0.6$ | $4.3 \pm 1.0$ | $11.6 \pm 2.5$ | $30.4 \pm 4.2$ | $52.9 \pm 6.4$ |
| | path len | $1.00 \pm 0.00$ | $1.00 \pm 0.00$ | $1.42 \pm 0.33$ | $2.44 \pm 0.36$ | $3.82 \pm 0.49$ | $5.12 \pm 0.79$ | $6.15 \pm 0.52$ |
| | gini | $.942 \pm .002$ | $.942 \pm .001$ | $.931 \pm .010$ | $.930 \pm .011$ | $.902 \pm .015$ | $.845 \pm .019$ | $.841 \pm .031$ |
| | time | $0.7 \pm 0.0$ | $0.7 \pm 0.0$ | $0.9 \pm 0.2$ | $1.4 \pm 0.2$ | $2.9 \pm 0.5$ | $6.8 \pm 0.9$ | $11.1 \pm 1.2$ |
| sonar | acc | $.534 \pm .144$ | $.793 \pm .073$ | $.865 \pm .061$ | $.851 \pm .063$ | $.827 \pm .069$ | $.860 \pm .066$ | $.808 \pm .053$ |
| | nodes | $0.0 \pm 0.0$ | $2.6 \pm 0.5$ | $7.8 \pm 1.2$ | $12.3 \pm 1.7$ | $14.7 \pm 2.5$ | $20.4 \pm 7.4$ | $30.2 \pm 7.4$ |
| | path len | $0.00 \pm 0.00$ | $2.01 \pm 0.33$ | $3.73 \pm 0.49$ | $4.65 \pm 0.48$ | $4.93 \pm 0.45$ | $5.91 \pm 0.84$ | $6.55 \pm 1.10$ |
| | gini | $.000 \pm .000$ | $.972 \pm .001$ | $.970 \pm .001$ | $.964 \pm .003$ | $.952 \pm .008$ | $.953 \pm .007$ | $.934 \pm .012$ |
| | time | $0.3 \pm 0.0$ | $1.1 \pm 0.1$ | $2.1 \pm 0.2$ | $3.1 \pm 0.4$ | $3.6 \pm 0.5$ | $4.8 \pm 1.4$ | $6.6 \pm 1.5$ |
| pendigits | acc | $.094 \pm .003$ | $.852 \pm .026$ | $.897 \pm .012$ | $.907 \pm .015$ | $.923 \pm .012$ | $.938 \pm .008$ | $.951 \pm .007$ |
| | nodes | $0.0 \pm 0.0$ | $8.7 \pm 0.5$ | $11.7 \pm 1.0$ | $21.2 \pm 2.3$ | $55.0 \pm 6.0$ | $135.4 \pm 8.4$ | $347.3 \pm 25.0$ |
| | path len | $0.00 \pm 0.00$ | $4.82 \pm 0.38$ | $4.84 \pm 0.28$ | $5.06 \pm 0.43$ | $6.28 \pm 0.31$ | $7.57 \pm 0.27$ | $9.46 \pm 0.50$ |
| | gini | $.000 \pm .000$ | $.869 \pm .011$ | $.846 \pm .022$ | $.736 \pm .027$ | $.612 \pm .046$ | $.533 \pm .042$ | $.478 \pm .030$ |
| | time | $0.4 \pm 0.0$ | $4.5 \pm 0.2$ | $5.4 \pm 0.3$ | $8.3 \pm 0.7$ | $16.9 \pm 1.5$ | $43.9 \pm 2.6$ | $99.0 \pm 6.9$ |
| ionosphere | acc | $.917 \pm .052$ | $.917 \pm .059$ | $.917 \pm .049$ | $.923 \pm .060$ | $.900 \pm .059$ | $.906 \pm .043$ | $.900 \pm .050$ |
| | nodes | $1.5 \pm 0.5$ | $2.5 \pm 0.8$ | $4.2 \pm 1.2$ | $8.2 \pm 2.6$ | $16.4 \pm 1.8$ | $22.2 \pm 4.0$ | $22.9 \pm 3.4$ |
| | path len | $1.24 \pm 0.25$ | $1.65 \pm 0.39$ | $2.23 \pm 0.47$ | $3.22 \pm 0.58$ | $4.88 \pm 0.69$ | $5.66 \pm 1.17$ | $5.56 \pm 0.91$ |
| | gini | $.953 \pm .003$ | $.950 \pm .010$ | $.952 \pm .004$ | $.935 \pm .016$ | $.872 \pm .037$ | $.842 \pm .030$ | $.819 \pm .019$ |
| | time | $0.8 \pm 0.1$ | $1.0 \pm 0.1$ | $1.3 \pm 0.3$ | $2.2 \pm 0.5$ | $3.9 \pm 0.4$ | $7.4 \pm 2.0$ | $6.8 \pm 0.9$ |

Ours: linear features, L1 regularization, random initialization, fuzzy

| data | metric | 1e-1 | 3e-2 | 1e-2 | 3e-3 | 1e-3 | 3e-4 | 1e-4 |
|------|--------|------|------|------|------|------|------|------|
| iris | acc | $.953 \pm .043$ | $.960 \pm .044$ | $.967 \pm .033$ | $.960 \pm .053$ | $.960 \pm .033$ | $.967 \pm .045$ | $.960 \pm .044$ |
| | nodes | $2.0 \pm 0.0$ | $2.0 \pm 0.0$ | $3.4 \pm 1.0$ | $5.9 \pm 1.2$ | $10.6 \pm 1.7$ | $17.1 \pm 1.8$ | $27.1 \pm 1.8$ |
| | path len | $1.67 \pm 0.09$ | $1.67 \pm 0.09$ | $2.12 \pm 0.35$ | $2.55 \pm 0.26$ | $3.14 \pm 0.40$ | $3.28 \pm 0.34$ | $4.32 \pm 0.48$ |
| | gini | $.573 \pm .032$ | $.566 \pm .018$ | $.547 \pm .078$ | $.480 \pm .074$ | $.562 \pm .049$ | $.481 \pm .016$ | $.097 \pm .035$ |
| | time | $0.7 \pm 0.1$ | $0.9 \pm 0.0$ | $2.4 \pm 2.9$ | $22.1 \pm 40.2$ | $14.7 \pm 35.8$ | $6.5 \pm 1.0$ | $36.9 \pm 19.5$ |
| heart | acc | $.819 \pm .038$ | $.828 \pm .036$ | $.802 \pm .046$ | $.789 \pm .070$ | $.812 \pm .038$ | $.766 \pm .064$ | $.786 \pm .061$ |
| | nodes | $1.0 \pm 0.0$ | $1.2 \pm 0.6$ | $5.3 \pm 1.7$ | $19.3 \pm 3.8$ | $26.8 \pm 5.0$ | $39.7 \pm 5.0$ | $57.2 \pm 6.8$ |
| | path len | $1.00 \pm 0.00$ | $1.10 \pm 0.30$ | $2.72 \pm 0.44$ | $4.74 \pm 0.55$ | $5.48 \pm 0.53$ | $5.57 \pm 0.40$ | $6.42 \pm 0.58$ |
| | gini | $.886 \pm .017$ | $.864 \pm .056$ | $.759 \pm .062$ | $.721 \pm .056$ | $.710 \pm .045$ | $.620 \pm .079$ | $.557 \pm .064$ |
| | time | $0.6 \pm 0.0$ | $0.6 \pm 0.2$ | $1.7 \pm 0.4$ | $4.0 \pm 0.7$ | $4.1 \pm 0.7$ | $5.9 \pm 0.7$ | $8.4 \pm 1.0$ |
| dry-bean | acc | $.528 \pm .008$ | $.901 \pm .009$ | $.913 \pm .006$ | $.913 \pm .005$ | $.913 \pm .008$ | $.919 \pm .006$ | $.920 \pm .007$ |
| | nodes | $2.0 \pm 0.0$ | $6.0 \pm 0.0$ | $7.8 \pm 0.4$ | $11.5 \pm 1.1$ | $28.0 \pm 0.9$ | $58.2 \pm 2.9$ | $139.2 \pm 15.3$ |
| | path len | $1.59 \pm 0.01$ | $3.04 \pm 0.20$ | $3.37 \pm 0.09$ | $3.76 \pm 0.19$ | $5.41 \pm 0.14$ | $6.24 \pm 0.26$ | $7.93 \pm 0.37$ |
| | gini | $.753 \pm .013$ | $.676 \pm .033$ | $.677 \pm .040$ | $.540 \pm .084$ | $.470 \pm .054$ | $.387 \pm .070$ | $.398 \pm .040$ |
| | time | $2.4 \pm 0.1$ | $5.9 \pm 4.5$ | $5.2 \pm 0.2$ | $6.8 \pm 0.4$ | $13.1 \pm 0.4$ | $23.6 \pm 0.8$ | $52.3 \pm 6.2$ |
| wine | acc | $.983 \pm .025$ | $.977 \pm .028$ | $.966 \pm .037$ | $.960 \pm .036$ | $.966 \pm .027$ | $.966 \pm .027$ | $.972 \pm .028$ |
| | nodes | $2.0 \pm 0.0$ | $2.0 \pm 0.0$ | $2.5 \pm 0.5$ | $3.1 \pm 0.7$ | $4.7 \pm 1.1$ | $7.5 \pm 1.8$ | $13.2 \pm 2.6$ |
| | path len | $1.67 \pm 0.10$ | $1.66 \pm 0.10$ | $1.86 \pm 0.23$ | $2.10 \pm 0.32$ | $2.46 \pm 0.19$ | $3.20 \pm 0.42$ | $4.11 \pm 0.46$ |
| | gini | $.818 \pm .011$ | $.789 \pm .011$ | $.814 \pm .019$ | $.775 \pm .045$ | $.779 \pm .032$ | $.756 \pm .035$ | $.740 \pm .045$ |
| | time | $0.5 \pm 0.0$ | $0.5 \pm 0.0$ | $0.5 \pm 0.1$ | $0.6 \pm 0.1$ | $0.9 \pm 0.2$ | $1.3 \pm 0.3$ | $2.1 \pm 0.4$ |
| car | acc | $.700 \pm .044$ | $.700 \pm .044$ | $.949 \pm .013$ | $.977 \pm .013$ | $.990 \pm .011$ | $.991 \pm .009$ | $.990 \pm .010$ |
| | nodes | $0.0 \pm 0.0$ | $0.0 \pm 0.0$ | $5.5 \pm 1.0$ | $15.6 \pm 4.1$ | $31.3 \pm 3.1$ | $36.1 \pm 7.3$ | $41.1 \pm 9.8$ |
| | path len | $0.00 \pm 0.00$ | $0.00 \pm 0.00$ | $2.13 \pm 0.25$ | $2.98 \pm 0.42$ | $3.41 \pm 0.46$ | $3.50 \pm 0.50$ | $3.61 \pm 0.69$ |
| | gini | $.000 \pm .000$ | $.000 \pm .000$ | $.720 \pm .019$ | $.703 \pm .045$ | $.703 \pm .032$ | $.655 \pm .033$ | $.663 \pm .031$ |
| | time | $0.3 \pm 0.0$ | $0.3 \pm 0.0$ | $2.0 \pm 0.3$ | $4.7 \pm 1.1$ | $8.8 \pm 0.9$ | $9.9 \pm 2.0$ | $11.3 \pm 2.8$ |
| wdbc | acc | $.970 \pm .019$ | $.970 \pm .019$ | $.974 \pm .018$ | $.968 \pm .017$ | $.979 \pm .020$ | $.968 \pm .022$ | $.979 \pm .015$ |
| | nodes | $1.0 \pm 0.0$ | $1.0 \pm 0.0$ | $1.0 \pm 0.0$ | $3.2 \pm 0.6$ | $7.0 \pm 3.0$ | $17.4 \pm 4.1$ | $34.9 \pm 8.9$ |
| | path len | $1.00 \pm 0.00$ | $1.00 \pm 0.00$ | $1.00 \pm 0.00$ | $2.01 \pm 0.02$ | $2.57 \pm 0.49$ | $3.93 \pm 0.41$ | $4.73 \pm 0.60$ |
| | gini | $.817 \pm .019$ | $.819 \pm .022$ | $.816 \pm .019$ | $.835 \pm .045$ | $.936 \pm .008$ | $.936 \pm .006$ | $.936 \pm .007$ |
| | time | $0.6 \pm 0.0$ | $0.6 \pm 0.0$ | $0.6 \pm 0.0$ | $1.2 \pm 0.2$ | $2.3 \pm 0.8$ | $4.9 \pm 1.1$ | $9.2 \pm 2.3$ |
| sonar | acc | $.655 \pm .169$ | $.841 \pm .074$ | $.812 \pm .070$ | $.807 \pm .078$ | $.817 \pm .074$ | $.812 \pm .084$ | $.846 \pm .076$ |
| | nodes | $0.9 \pm 0.3$ | $2.6 \pm 0.7$ | $5.6 \pm 0.9$ | $7.0 \pm 1.6$ | $12.6 \pm 4.8$ | $21.7 \pm 3.7$ | $28.9 \pm 7.4$ |
| | path len | $0.90 \pm 0.30$ | $1.86 \pm 0.28$ | $2.73 \pm 0.35$ | $2.98 \pm 0.29$ | $3.95 \pm 0.65$ | $5.00 \pm 0.84$ | $5.67 \pm 0.82$ |
| | gini | $.849 \pm .283$ | $.939 \pm .014$ | $.899 \pm .029$ | $.899 \pm .019$ | $.915 \pm .020$ | $.933 \pm .022$ | $.923 \pm .033$ |
| | time | $0.8 \pm 0.1$ | $1.4 \pm 0.2$ | $2.6 \pm 0.3$ | $3.1 \pm 0.6$ | $5.2 \pm 1.8$ | $8.4 \pm 1.4$ | $8.6 \pm 3.1$ |
| pendigits | acc | $.094 \pm .003$ | $.901 \pm .008$ | $.940 \pm .011$ | $.968 \pm .002$ | $.976 \pm .004$ | $.977 \pm .005$ | $.981 \pm .003$ |
| | nodes | $0.0 \pm 0.0$ | $9.0 \pm 0.0$ | $11.9 \pm 1.6$ | $17.9 \pm 1.6$ | $42.2 \pm 4.0$ | $120.1 \pm 7.1$ | $298.2 \pm 9.1$ |
| | path len | $0.00 \pm 0.00$ | $3.96 \pm 0.04$ | $4.55 \pm 0.45$ | $4.84 \pm 0.26$ | $5.91 \pm 0.30$ | $7.40 \pm 0.18$ | $8.89 \pm 0.32$ |
| | gini | $.000 \pm .000$ | $.809 \pm .010$ | $.813 \pm .024$ | $.764 \pm .022$ | $.696 \pm .041$ | $.492 \pm .046$ | $.320 \pm .027$ |
| | time | $9.5 \pm 8.9$ | $7.6 \pm 3.3$ | $139.3 \pm 83.5$ | $109.5 \pm 27.3$ | $202.4 \pm 82.2$ | $392.0 \pm 106.4$ | $1032.4 \pm 182.0$ |
| ionosphere | acc | $.855 \pm .079$ | $.909 \pm .062$ | $.926 \pm .038$ | $.923 \pm .054$ | $.926 \pm .050$ | $.920 \pm .064$ | $.940 \pm .035$ |
| | nodes | $1.0 \pm 0.0$ | $2.4 \pm 0.5$ | $3.4 \pm 0.5$ | $6.3 \pm 1.6$ | $15.7 \pm 4.0$ | $32.2 \pm 4.6$ | $52.3 \pm 5.4$ |
| | path len | $1.00 \pm 0.00$ | $2.02 \pm 0.33$ | $2.73 \pm 0.41$ | $3.83 \pm 0.55$ | $5.60 \pm 1.09$ | $8.38 \pm 0.77$ | $11.72 \pm 0.88$ |
| | gini | $.910 \pm .010$ | $.828 \pm .094$ | $.699 \pm .090$ | $.663 \pm .067$ | $.712 \pm .063$ | $.785 \pm .048$ | $.853 \pm .028$ |
| | time | $0.6 \pm 0.0$ | $1.0 \pm 0.1$ | $1.3 \pm 0.1$ | $2.0 \pm 0.4$ | $2.8 \pm 0.8$ | $5.1 \pm 0.7$ | $10.4 \pm 2.9$ |

Ours: linear features, L1 regularization, random initialization, crisp

| data | metric | 1e-1 | 3e-2 | 1e-2 | 3e-3 | 1e-3 | 3e-4 | 1e-4 |
|---|---|---|---|---|---|---|---|---|
| iris | acc | $.953 \pm .043$ | $.960 \pm .044$ | $.960 \pm .033$ | $.953 \pm .052$ | $.960 \pm .044$ | $.960 \pm .044$ | $.967 \pm .045$ |
| | nodes | $2.0 \pm 0.0$ | $2.0 \pm 0.0$ | $3.4 \pm 1.0$ | $5.9 \pm 1.2$ | $10.6 \pm 1.7$ | $17.1 \pm 1.8$ | $27.1 \pm 1.8$ |
| | path len | $1.67 \pm 0.09$ | $1.67 \pm 0.09$ | $2.12 \pm 0.35$ | $2.53 \pm 0.26$ | $3.14 \pm 0.44$ | $3.25 \pm 0.35$ | $4.45 \pm 0.58$ |
| | gini | $.573 \pm .032$ | $.566 \pm .018$ | $.547 \pm .078$ | $.469 \pm .082$ | $.563 \pm .049$ | $.480 \pm .015$ | $.099 \pm .038$ |
| | time | $0.4 \pm 0.0$ | $0.4 \pm 0.0$ | $0.6 \pm 0.1$ | $0.9 \pm 0.1$ | $1.5 \pm 0.2$ | $2.2 \pm 0.2$ | $3.4 \pm 0.2$ |
| heart | acc | $.819 \pm .039$ | $.828 \pm .036$ | $.786 \pm .051$ | $.780 \pm .066$ | $.812 \pm .037$ | $.756 \pm .071$ | $.786 \pm .064$ |
| | nodes | $1.0 \pm 0.0$ | $1.2 \pm 0.6$ | $5.3 \pm 1.7$ | $19.3 \pm 3.8$ | $26.8 \pm 5.0$ | $39.7 \pm 5.0$ | $57.2 \pm 6.8$ |
| | path len | $1.00 \pm 0.00$ | $1.10 \pm 0.30$ | $2.72 \pm 0.44$ | $4.72 \pm 0.53$ | $5.49 \pm 0.53$ | $5.56 \pm 0.42$ | $6.42 \pm 0.66$ |
| | gini | $.886 \pm .017$ | $.864 \pm .056$ | $.760 \pm .063$ | $.720 \pm .056$ | $.709 \pm .046$ | $.618 \pm .079$ | $.559 \pm .065$ |
| | time | $0.4 \pm 0.0$ | $0.5 \pm 0.1$ | $1.2 \pm 0.3$ | $3.6 \pm 0.7$ | $3.6 \pm 0.6$ | $5.1 \pm 0.6$ | $7.1 \pm 0.8$ |
| dry-bean | acc | $.530 \pm .007$ | $.892 \pm .012$ | $.909 \pm .007$ | $.905 \pm .008$ | $.907 \pm .007$ | $.913 \pm .007$ | $.910 \pm .005$ |
| | nodes | $2.0 \pm 0.0$ | $6.0 \pm 0.0$ | $7.8 \pm 0.4$ | $11.5 \pm 1.1$ | $28.0 \pm 0.9$ | $58.2 \pm 2.9$ | $139.2 \pm 15.3$ |
| | path len | $1.60 \pm 0.01$ | $3.05 \pm 0.23$ | $3.36 \pm 0.08$ | $3.77 \pm 0.19$ | $5.46 \pm 0.15$ | $6.27 \pm 0.26$ | $8.13 \pm 0.46$ |
| | gini | $.753 \pm .013$ | $.676 \pm .033$ | $.677 \pm .040$ | $.540 \pm .083$ | $.471 \pm .054$ | $.387 \pm .070$ | $.398 \pm .040$ |
| | time | $2.3 \pm 0.0$ | $4.1 \pm 0.1$ | $4.8 \pm 0.2$ | $5.9 \pm 0.3$ | $10.5 \pm 0.3$ | $18.0 \pm 0.7$ | $37.1 \pm 3.7$ |
| wine | acc | $.983 \pm .025$ | $.977 \pm .028$ | $.966 \pm .037$ | $.960 \pm .036$ | $.967 \pm .027$ | $.967 \pm .027$ | $.966 \pm .027$ |
| | nodes | $2.0 \pm 0.0$ | $2.0 \pm 0.0$ | $2.5 \pm 0.5$ | $3.1 \pm 0.7$ | $4.7 \pm 1.1$ | $7.5 \pm 1.8$ | $13.2 \pm 2.6$ |
| | path len | $1.67 \pm 0.09$ | $1.66 \pm 0.09$ | $1.85 \pm 0.23$ | $2.10 \pm 0.31$ | $2.45 \pm 0.20$ | $3.19 \pm 0.44$ | $4.12 \pm 0.48$ |
| | gini | $.817 \pm .011$ | $.789 \pm .011$ | $.814 \pm .019$ | $.775 \pm .045$ | $.779 \pm .032$ | $.756 \pm .035$ | $.740 \pm .045$ |
| | time | $0.4 \pm 0.0$ | $0.4 \pm 0.0$ | $0.5 \pm 0.1$ | $0.6 \pm 0.1$ | $0.8 \pm 0.1$ | $1.1 \pm 0.2$ | $1.8 \pm 0.3$ |
| car | acc | $.700 \pm .044$ | $.700 \pm .044$ | $.949 \pm .013$ | $.978 \pm .013$ | $.990 \pm .011$ | $.992 \pm .009$ | $.990 \pm .010$ |
| | nodes | $0.0 \pm 0.0$ | $0.0 \pm 0.0$ | $5.5 \pm 1.0$ | $15.6 \pm 4.1$ | $31.3 \pm 3.1$ | $36.1 \pm 7.3$ | $41.1 \pm 9.8$ |
| | path len | $0.00 \pm 0.00$ | $0.00 \pm 0.00$ | $2.13 \pm 0.25$ | $2.98 \pm 0.42$ | $3.41 \pm 0.46$ | $3.50 \pm 0.50$ | $3.62 \pm 0.69$ |
| | gini | $.000 \pm .000$ | $.000 \pm .000$ | $.720 \pm .020$ | $.703 \pm .045$ | $.703 \pm .032$ | $.655 \pm .033$ | $.663 \pm .031$ |
| | time | $0.3 \pm 0.0$ | $0.3 \pm 0.0$ | $1.9 \pm 0.3$ | $4.2 \pm 1.0$ | $7.8 \pm 0.8$ | $8.8 \pm 1.7$ | $9.9 \pm 2.2$ |
| wdbc | acc | $.970 \pm .019$ | $.970 \pm .019$ | $.974 \pm .018$ | $.968 \pm .017$ | $.977 \pm .016$ | $.965 \pm .022$ | $.970 \pm .025$ |
| | nodes | $1.0 \pm 0.0$ | $1.0 \pm 0.0$ | $1.0 \pm 0.0$ | $3.2 \pm 0.6$ | $7.0 \pm 3.0$ | $17.4 \pm 4.1$ | $34.9 \pm 8.9$ |
| | path len | $1.00 \pm 0.00$ | $1.00 \pm 0.00$ | $1.00 \pm 0.00$ | $2.00 \pm 0.01$ | $2.57 \pm 0.49$ | $3.94 \pm 0.41$ | $4.72 \pm 0.61$ |
| | gini | $.817 \pm .019$ | $.819 \pm .022$ | $.816 \pm .019$ | $.835 \pm .045$ | $.936 \pm .008$ | $.936 \pm .006$ | $.936 \pm .007$ |
| | time | $0.6 \pm 0.0$ | $0.6 \pm 0.0$ | $0.6 \pm 0.0$ | $1.1 \pm 0.1$ | $2.0 \pm 0.7$ | $4.3 \pm 0.9$ | $7.8 \pm 1.8$ |
| sonar | acc | $.665 \pm .174$ | $.841 \pm .071$ | $.807 \pm .081$ | $.797 \pm .079$ | $.827 \pm .065$ | $.803 \pm .084$ | $.832 \pm .071$ |
| | nodes | $0.9 \pm 0.3$ | $2.6 \pm 0.7$ | $5.6 \pm 0.9$ | $7.0 \pm 1.6$ | $12.6 \pm 4.8$ | $21.7 \pm 3.7$ | $28.9 \pm 7.4$ |
| | path len | $0.90 \pm 0.30$ | $1.86 \pm 0.28$ | $2.70 \pm 0.36$ | $3.00 \pm 0.26$ | $3.97 \pm 0.66$ | $4.96 \pm 0.83$ | $5.68 \pm 0.81$ |
| | gini | $.849 \pm .283$ | $.939 \pm .013$ | $.899 \pm .029$ | $.900 \pm .018$ | $.915 \pm .020$ | $.933 \pm .022$ | $.924 \pm .032$ |
| | time | $0.5 \pm 0.1$ | $1.0 \pm 0.2$ | $1.6 \pm 0.2$ | $2.0 \pm 0.3$ | $3.2 \pm 1.0$ | $5.2 \pm 0.8$ | $3.9 \pm 0.9$ |
| pendigits | acc | $.094 \pm .003$ | $.899 \pm .009$ | $.936 \pm .009$ | $.963 \pm .004$ | $.970 \pm .006$ | $.970 \pm .004$ | $.975 \pm .005$ |
| | nodes | $0.0 \pm 0.0$ | $9.0 \pm 0.0$ | $11.9 \pm 1.6$ | $17.9 \pm 1.6$ | $42.2 \pm 4.0$ | $120.1 \pm 7.1$ | $292.1 \pm 10.3$ |
| | path len | $0.00 \pm 0.00$ | $3.97 \pm 0.03$ | $4.56 \pm 0.45$ | $4.84 \pm 0.26$ | $5.90 \pm 0.31$ | $7.38 \pm 0.19$ | $8.78 \pm 0.33$ |
| | gini | $.000 \pm .000$ | $.809 \pm .010$ | $.813 \pm .024$ | $.764 \pm .022$ | $.695 \pm .041$ | $.492 \pm .045$ | $.321 \pm .027$ |
| | time | $11.3 \pm 5.0$ | $72.8 \pm 33.8$ | $83.9 \pm 14.0$ | $112.8 \pm 51.7$ | $220.7 \pm 82.2$ | $31.2 \pm 1.6$ | $84.9 \pm 8.8$ |
| ionosphere | acc | $.857 \pm .077$ | $.909 \pm .060$ | $.923 \pm .038$ | $.917 \pm .055$ | $.926 \pm .045$ | $.900 \pm .058$ | $.915 \pm .044$ |
| | nodes | $1.0 \pm 0.0$ | $2.4 \pm 0.5$ | $3.4 \pm 0.5$ | $6.3 \pm 1.6$ | $15.7 \pm 4.0$ | $32.2 \pm 4.6$ | $52.3 \pm 5.4$ |
| | path len | $1.00 \pm 0.00$ | $2.02 \pm 0.34$ | $2.77 \pm 0.39$ | $3.89 \pm 0.55$ | $5.63 \pm 1.16$ | $8.56 \pm 0.85$ | $12.33 \pm 0.88$ |
| | gini | $.910 \pm .010$ | $.828 \pm .093$ | $.700 \pm .088$ | $.665 \pm .067$ | $.711 \pm .062$ | $.786 \pm .047$ | $.855 \pm .027$ |
| | time | $0.5 \pm 0.0$ | $0.9 \pm 0.1$ | $1.2 \pm 0.1$ | $1.8 \pm 0.4$ | $2.4 \pm 0.7$ | $4.3 \pm 0.6$ | $8.7 \pm 2.8$ |

Ours: prototype features, L1 regularization, random initialization, fuzzy

| data | metric | 1e-1 | 3e-2 | 1e-2 | 3e-3 | 1e-3 | 3e-4 | 1e-4 |
|------|--------|------|------|------|------|------|------|------|
| iris | acc | $.947 \pm .065$ | $.940 \pm .047$ | $.947 \pm .050$ | $.967 \pm .045$ | $.947 \pm .072$ | $.933 \pm .060$ | $.947 \pm .050$ |
| | nodes | $2.0 \pm 0.0$ | $2.0 \pm 0.0$ | $3.1 \pm 0.7$ | $6.2 \pm 1.1$ | $9.5 \pm 2.2$ | $18.3 \pm 4.4$ | $26.3 \pm 7.3$ |
| | path len | $1.67 \pm 0.09$ | $1.67 \pm 0.09$ | $2.02 \pm 0.22$ | $2.58 \pm 0.32$ | $2.92 \pm 0.54$ | $3.95 \pm 0.63$ | $4.40 \pm 0.69$ |
| | gini | $.616 \pm .043$ | $.646 \pm .040$ | $.604 \pm .059$ | $.575 \pm .061$ | $.577 \pm .094$ | $.538 \pm .054$ | $.580 \pm .045$ |
| | time | $1.1 \pm 0.0$ | $1.1 \pm 0.0$ | $1.5 \pm 0.2$ | $2.4 \pm 0.3$ | $2.7 \pm 0.5$ | $6.0 \pm 1.3$ | $8.4 \pm 2.2$ |
| heart | acc | $.812 \pm .057$ | $.805 \pm .048$ | $.775 \pm .049$ | $.756 \pm .054$ | $.766 \pm .081$ | $.756 \pm .052$ | $.798 \pm .036$ |
| | nodes | $1.0 \pm 0.0$ | $1.2 \pm 0.4$ | $3.9 \pm 1.1$ | $13.5 \pm 1.7$ | $45.9 \pm 5.0$ | $71.1 \pm 6.8$ | $97.1 \pm 9.0$ |
| | path len | $1.00 \pm 0.00$ | $1.08 \pm 0.16$ | $2.19 \pm 0.34$ | $3.85 \pm 0.45$ | $5.48 \pm 0.48$ | $6.99 \pm 0.75$ | $7.68 \pm 0.84$ |
| | gini | $.936 \pm .003$ | $.935 \pm .002$ | $.929 \pm .005$ | $.900 \pm .012$ | $.852 \pm .022$ | $.839 \pm .018$ | $.828 \pm .031$ |
| | time | $0.9 \pm 0.0$ | $1.0 \pm 0.1$ | $1.8 \pm 0.3$ | $4.8 \pm 0.6$ | $14.7 \pm 1.5$ | $22.3 \pm 2.1$ | $22.9 \pm 2.3$ |
| dry-bean | acc | $.617 \pm .068$ | $.869 \pm .035$ | $.894 \pm .011$ | $.902 \pm .006$ | $.905 \pm .006$ | $.909 \pm .005$ | $.915 \pm .006$ |
| | nodes | $2.6 \pm 0.5$ | $5.6 \pm 0.5$ | $7.5 \pm 1.0$ | $16.1 \pm 1.9$ | $37.1 \pm 2.8$ | $79.7 \pm 7.0$ | $190.1 \pm 15.1$ |
| | path len | $1.92 \pm 0.31$ | $2.83 \pm 0.17$ | $3.15 \pm 0.29$ | $4.38 \pm 0.19$ | $5.59 \pm 0.31$ | $6.95 \pm 0.34$ | $8.86 \pm 0.71$ |
| | gini | $.724 \pm .052$ | $.718 \pm .027$ | $.596 \pm .078$ | $.385 \pm .058$ | $.356 \pm .046$ | $.310 \pm .046$ | $.303 \pm .042$ |
| | time | $2.7 \pm 0.4$ | $4.5 \pm 0.2$ | $5.8 \pm 0.8$ | $10.2 \pm 0.9$ | $20.6 \pm 1.3$ | $38.1 \pm 3.7$ | $86.0 \pm 9.2$ |
| wine | acc | $.955 \pm .043$ | $.960 \pm .036$ | $.955 \pm .042$ | $.950 \pm .058$ | $.955 \pm .042$ | $.939 \pm .046$ | $.939 \pm .052$ |
| | nodes | $2.0 \pm 0.0$ | $2.2 \pm 0.4$ | $3.1 \pm 0.5$ | $8.3 \pm 2.2$ | $14.1 \pm 3.2$ | $25.2 \pm 6.4$ | $35.1 \pm 9.5$ |
| | path len | $1.66 \pm 0.10$ | $1.74 \pm 0.20$ | $2.01 \pm 0.25$ | $3.26 \pm 0.49$ | $4.17 \pm 0.49$ | $5.18 \pm 0.69$ | $5.70 \pm 0.72$ |
| | gini | $.845 \pm .012$ | $.847 \pm .020$ | $.815 \pm .026$ | $.614 \pm .111$ | $.529 \pm .085$ | $.418 \pm .094$ | $.403 \pm .070$ |
| | time | $0.9 \pm 0.0$ | $1.0 \pm 0.1$ | $1.2 \pm 0.1$ | $2.4 \pm 0.5$ | $3.7 \pm 0.7$ | $6.2 \pm 1.4$ | $8.4 \pm 2.1$ |
| car | acc | $.700 \pm .044$ | $.770 \pm .052$ | $.833 \pm .046$ | $.953 \pm .017$ | $.969 \pm .014$ | $.974 \pm .015$ | $.983 \pm .009$ |
| | nodes | $0.0 \pm 0.0$ | $1.1 \pm 0.7$ | $2.8 \pm 1.2$ | $12.9 \pm 2.9$ | $26.5 \pm 6.8$ | $49.8 \pm 12.4$ | $70.4 \pm 13.1$ |
| | path len | $0.00 \pm 0.00$ | $0.97 \pm 0.39$ | $1.61 \pm 0.38$ | $2.86 \pm 0.40$ | $3.38 \pm 0.54$ | $3.70 \pm 0.59$ | $4.00 \pm 0.49$ |
| | gini | $.000 \pm .000$ | $.699 \pm .236$ | $.816 \pm .040$ | $.769 \pm .035$ | $.707 \pm .056$ | $.658 \pm .035$ | $.596 \pm .038$ |
| | time | $0.3 \pm 0.0$ | $0.8 \pm 0.2$ | $1.3 \pm 0.3$ | $3.9 \pm 0.7$ | $7.1 \pm 1.5$ | $12.2 \pm 2.8$ | $16.7 \pm 2.9$ |
| wdbc | acc | $.961 \pm .025$ | $.956 \pm .024$ | $.961 \pm .027$ | $.970 \pm .019$ | $.963 \pm .024$ | $.961 \pm .022$ | $.953 \pm .024$ |
| | nodes | $1.0 \pm 0.0$ | $1.1 \pm 0.3$ | $1.8 \pm 0.7$ | $6.3 \pm 1.3$ | $19.1 \pm 1.8$ | $37.4 \pm 3.1$ | $68.8 \pm 4.4$ |
| | path len | $1.00 \pm 0.00$ | $1.07 \pm 0.22$ | $1.36 \pm 0.37$ | $2.69 \pm 0.28$ | $3.90 \pm 0.41$ | $4.75 \pm 0.32$ | $6.08 \pm 0.48$ |
| | gini | $.864 \pm .015$ | $.851 \pm .039$ | $.821 \pm .038$ | $.779 \pm .049$ | $.646 \pm .062$ | $.569 \pm .064$ | $.633 \pm .045$ |
| | time | $0.7 \pm 0.0$ | $0.8 \pm 0.1$ | $0.9 \pm 0.2$ | $2.0 \pm 0.3$ | $5.0 \pm 0.4$ | $9.3 \pm 0.8$ | $16.7 \pm 1.0$ |
| sonar | acc | $.534 \pm .144$ | $.759 \pm .091$ | $.798 \pm .072$ | $.789 \pm .052$ | $.774 \pm .046$ | $.779 \pm .081$ | $.812 \pm .058$ |
| | nodes | $0.0 \pm 0.0$ | $2.0 \pm 0.4$ | $7.6 \pm 1.4$ | $18.7 \pm 2.6$ | $29.5 \pm 4.3$ | $39.3 \pm 3.3$ | $50.4 \pm 6.3$ |
| | path len | $0.00 \pm 0.00$ | $1.64 \pm 0.28$ | $3.38 \pm 0.29$ | $4.70 \pm 0.36$ | $5.30 \pm 0.50$ | $5.77 \pm 0.44$ | $6.24 \pm 0.67$ |
| | gini | $.000 \pm .000$ | $.931 \pm .014$ | $.905 \pm .020$ | $.844 \pm .024$ | $.770 \pm .029$ | $.743 \pm .031$ | $.731 \pm .037$ |
| | time | $0.3 \pm 0.0$ | $1.0 \pm 0.1$ | $2.4 \pm 0.5$ | $4.9 \pm 0.6$ | $7.3 \pm 1.0$ | $9.6 \pm 0.8$ | $12.0 \pm 1.5$ |
| pendigits | acc | $.094 \pm .003$ | $.783 \pm .057$ | $.854 \pm .024$ | $.881 \pm .019$ | $.912 \pm .012$ | $.939 \pm .009$ | $.949 \pm .011$ |
| | nodes | $0.0 \pm 0.0$ | $8.2 \pm 0.7$ | $12.8 \pm 1.2$ | $30.2 \pm 3.7$ | $69.7 \pm 10.1$ | $166.7 \pm 10.7$ | $372.4 \pm 23.3$ |
| | path len | $0.00 \pm 0.00$ | $4.10 \pm 0.47$ | $4.56 \pm 0.56$ | $5.28 \pm 0.34$ | $6.58 \pm 0.27$ | $7.93 \pm 0.34$ | $9.49 \pm 0.18$ |
| | gini | $.000 \pm .000$ | $.770 \pm .029$ | $.684 \pm .043$ | $.475 \pm .090$ | $.317 \pm .030$ | $.353 \pm .045$ | $.354 \pm .031$ |
| | time | $0.4 \pm 0.0$ | $4.7 \pm 0.3$ | $6.6 \pm 0.5$ | $12.2 \pm 1.2$ | $24.2 \pm 2.8$ | $62.9 \pm 4.2$ | $179.6 \pm 18.3$ |
| ionosphere | acc | $.903 \pm .045$ | $.892 \pm .057$ | $.920 \pm .045$ | $.909 \pm .035$ | $.897 \pm .092$ | $.917 \pm .052$ | $.906 \pm .054$ |
| | nodes | $1.1 \pm 0.3$ | $1.8 \pm 0.6$ | $3.8 \pm 1.1$ | $10.5 \pm 2.3$ | $23.3 \pm 4.2$ | $40.5 \pm 7.4$ | $58.8 \pm 6.4$ |
| | path len | $1.03 \pm 0.10$ | $1.42 \pm 0.34$ | $2.29 \pm 0.46$ | $4.35 \pm 0.66$ | $5.96 \pm 0.88$ | $7.48 \pm 0.88$ | $9.28 \pm 1.87$ |
| | gini | $.894 \pm .012$ | $.887 \pm .029$ | $.839 \pm .033$ | $.706 \pm .022$ | $.616 \pm .050$ | $.596 \pm .051$ | $.594 \pm .058$ |
| | time | $0.7 \pm 0.1$ | $0.9 \pm 0.2$ | $1.4 \pm 0.3$ | $2.9 \pm 0.6$ | $6.1 \pm 0.9$ | $13.2 \pm 2.3$ | $18.9 \pm 1.9$ |

Ours: prototype features, L1 regularization, random initialization, crisp

| data | metric | 1e-1 | 3e-2 | 1e-2 | 3e-3 | 1e-3 | 3e-4 | 1e-4 |
|---|---|---|---|---|---|---|---|---|
| iris | acc | $.947 \pm .065$ | $.940 \pm .047$ | $.947 \pm .050$ | $.947 \pm .050$ | $.947 \pm .058$ | $.927 \pm .063$ | $.947 \pm .050$ |
| | nodes | $2.0 \pm 0.0$ | $2.0 \pm 0.0$ | $3.1 \pm 0.7$ | $6.2 \pm 1.1$ | $9.5 \pm 2.2$ | $18.3 \pm 4.4$ | $26.3 \pm 7.3$ |
| | path len | $1.67 \pm 0.09$ | $1.67 \pm 0.09$ | $2.03 \pm 0.21$ | $2.56 \pm 0.33$ | $2.89 \pm 0.53$ | $4.00 \pm 0.61$ | $4.43 \pm 0.71$ |
| | gini | $.616 \pm .043$ | $.646 \pm .040$ | $.604 \pm .058$ | $.574 \pm .061$ | $.575 \pm .095$ | $.538 \pm .056$ | $.578 \pm .048$ |
| | time | $1.0 \pm 0.0$ | $1.0 \pm 0.0$ | $1.4 \pm 0.2$ | $1.7 \pm 0.2$ | $2.4 \pm 0.5$ | $5.3 \pm 1.1$ | $7.4 \pm 1.9$ |
| heart | acc | $.809 \pm .055$ | $.802 \pm .036$ | $.772 \pm .061$ | $.743 \pm .066$ | $.737 \pm .082$ | $.750 \pm .061$ | $.792 \pm .041$ |
| | nodes | $1.0 \pm 0.0$ | $1.2 \pm 0.4$ | $3.9 \pm 1.1$ | $13.5 \pm 1.7$ | $45.9 \pm 5.0$ | $71.1 \pm 6.8$ | $97.1 \pm 9.0$ |
| | path len | $1.00 \pm 0.00$ | $1.09 \pm 0.17$ | $2.19 \pm 0.37$ | $3.84 \pm 0.48$ | $5.47 \pm 0.53$ | $7.09 \pm 0.80$ | $7.81 \pm 0.98$ |
| | gini | $.936 \pm .003$ | $.935 \pm .002$ | $.929 \pm .005$ | $.900 \pm .012$ | $.852 \pm .022$ | $.840 \pm .020$ | $.828 \pm .031$ |
| | time | $0.7 \pm 0.0$ | $0.7 \pm 0.1$ | $1.3 \pm 0.2$ | $3.3 \pm 0.4$ | $9.4 \pm 0.9$ | $14.1 \pm 1.3$ | $19.1 \pm 1.8$ |
| dry-bean | acc | $.612 \pm .065$ | $.861 \pm .035$ | $.887 \pm .011$ | $.890 \pm .007$ | $.892 \pm .007$ | $.894 \pm .006$ | $.893 \pm .017$ |
| | nodes | $2.6 \pm 0.5$ | $5.6 \pm 0.5$ | $7.5 \pm 1.0$ | $16.1 \pm 1.9$ | $37.1 \pm 2.8$ | $79.7 \pm 7.0$ | $190.1 \pm 15.1$ |
| | path len | $1.93 \pm 0.31$ | $2.84 \pm 0.18$ | $3.14 \pm 0.29$ | $4.39 \pm 0.20$ | $5.65 \pm 0.33$ | $7.07 \pm 0.40$ | $9.45 \pm 1.11$ |
| | gini | $.724 \pm .052$ | $.718 \pm .026$ | $.596 \pm .078$ | $.383 \pm .059$ | $.357 \pm .046$ | $.313 \pm .045$ | $.308 \pm .045$ |
| | time | $2.5 \pm 0.3$ | $4.1 \pm 0.2$ | $5.0 \pm 0.5$ | $8.6 \pm 0.7$ | $15.8 \pm 1.0$ | $27.2 \pm 2.6$ | $56.9 \pm 4.7$ |
| wine | acc | $.949 \pm .040$ | $.960 \pm .036$ | $.939 \pm .058$ | $.944 \pm .066$ | $.933 \pm .054$ | $.922 \pm .067$ | $.916 \pm .037$ |
| | nodes | $2.0 \pm 0.0$ | $2.2 \pm 0.4$ | $3.1 \pm 0.5$ | $8.3 \pm 2.2$ | $14.1 \pm 3.2$ | $25.2 \pm 6.4$ | $35.1 \pm 9.5$ |
| | path len | $1.67 \pm 0.11$ | $1.74 \pm 0.20$ | $2.02 \pm 0.26$ | $3.26 \pm 0.52$ | $4.22 \pm 0.55$ | $5.25 \pm 0.76$ | $5.76 \pm 0.75$ |
| | gini | $.845 \pm .012$ | $.847 \pm .020$ | $.815 \pm .026$ | $.612 \pm .110$ | $.529 \pm .085$ | $.420 \pm .094$ | $.403 \pm .071$ |
| | time | $0.8 \pm 0.1$ | $0.9 \pm 0.1$ | $1.1 \pm 0.1$ | $2.2 \pm 0.4$ | $3.3 \pm 0.6$ | $5.4 \pm 1.2$ | $7.2 \pm 1.8$ |
| car | acc | $.700 \pm .044$ | $.771 \pm .053$ | $.832 \pm .044$ | $.953 \pm .017$ | $.969 \pm .013$ | $.974 \pm .015$ | $.982 \pm .009$ |
| | nodes | $0.0 \pm 0.0$ | $1.1 \pm 0.7$ | $2.8 \pm 1.2$ | $12.9 \pm 2.9$ | $26.5 \pm 6.8$ | $49.8 \pm 12.4$ | $70.4 \pm 13.1$ |
| | path len | $0.00 \pm 0.00$ | $0.97 \pm 0.39$ | $1.61 \pm 0.39$ | $2.85 \pm 0.39$ | $3.38 \pm 0.54$ | $3.70 \pm 0.59$ | $4.01 \pm 0.49$ |
| | gini | $.000 \pm .000$ | $.700 \pm .236$ | $.816 \pm .040$ | $.769 \pm .035$ | $.707 \pm .056$ | $.658 \pm .035$ | $.596 \pm .038$ |
| | time | $0.3 \pm 0.0$ | $0.8 \pm 0.2$ | $1.2 \pm 0.3$ | $3.4 \pm 0.6$ | $6.3 \pm 1.3$ | $10.9 \pm 2.2$ | $14.8 \pm 2.5$ |
| wdbc | acc | $.961 \pm .025$ | $.956 \pm .024$ | $.961 \pm .027$ | $.958 \pm .026$ | $.947 \pm .025$ | $.951 \pm .020$ | $.937 \pm .037$ |
| | nodes | $1.0 \pm 0.0$ | $1.1 \pm 0.3$ | $1.8 \pm 0.7$ | $6.3 \pm 1.3$ | $19.1 \pm 1.8$ | $37.4 \pm 3.1$ | $68.8 \pm 4.4$ |
| | path len | $1.00 \pm 0.00$ | $1.07 \pm 0.22$ | $1.36 \pm 0.37$ | $2.69 \pm 0.30$ | $3.88 \pm 0.40$ | $4.71 \pm 0.35$ | $6.13 \pm 0.61$ |
| | gini | $.864 \pm .015$ | $.851 \pm .039$ | $.821 \pm .038$ | $.780 \pm .050$ | $.645 \pm .062$ | $.568 \pm .063$ | $.632 \pm .044$ |
| | time | $0.7 \pm 0.0$ | $0.7 \pm 0.1$ | $0.9 \pm 0.2$ | $1.8 \pm 0.3$ | $4.4 \pm 0.4$ | $7.9 \pm 0.7$ | $13.8 \pm 0.8$ |
| sonar | acc | $.534 \pm .144$ | $.769 \pm .115$ | $.813 \pm .062$ | $.798 \pm .046$ | $.778 \pm .057$ | $.788 \pm .066$ | $.803 \pm .072$ |
| | nodes | $0.0 \pm 0.0$ | $2.0 \pm 0.4$ | $7.6 \pm 1.4$ | $18.7 \pm 2.6$ | $29.5 \pm 4.3$ | $39.3 \pm 3.3$ | $50.4 \pm 6.3$ |
| | path len | $0.00 \pm 0.00$ | $1.65 \pm 0.27$ | $3.44 \pm 0.30$ | $4.73 \pm 0.41$ | $5.34 \pm 0.54$ | $5.85 \pm 0.48$ | $6.31 \pm 0.71$ |
| | gini | $.000 \pm .000$ | $.931 \pm .014$ | $.906 \pm .020$ | $.844 \pm .023$ | $.770 \pm .028$ | $.744 \pm .032$ | $.730 \pm .035$ |
| | time | $0.3 \pm 0.0$ | $0.9 \pm 0.1$ | $2.1 \pm 0.3$ | $4.3 \pm 0.5$ | $6.4 \pm 0.8$ | $8.5 \pm 1.1$ | $10.5 \pm 1.6$ |
| pendigits | acc | $.094 \pm .003$ | $.773 \pm .047$ | $.844 \pm .024$ | $.865 \pm .015$ | $.892 \pm .013$ | $.919 \pm .010$ | $.930 \pm .014$ |
| | nodes | $0.0 \pm 0.0$ | $8.3 \pm 0.6$ | $12.8 \pm 1.2$ | $30.2 \pm 3.7$ | $69.7 \pm 10.1$ | $166.7 \pm 10.7$ | $372.4 \pm 23.3$ |
| | path len | $0.00 \pm 0.00$ | $3.93 \pm 0.45$ | $4.57 \pm 0.58$ | $5.28 \pm 0.34$ | $6.58 \pm 0.27$ | $7.93 \pm 0.37$ | $9.51 \pm 0.18$ |
| | gini | $.000 \pm .000$ | $.743 \pm .034$ | $.684 \pm .043$ | $.475 \pm .090$ | $.317 \pm .030$ | $.352 \pm .045$ | $.354 \pm .031$ |
| | time | $0.4 \pm 0.0$ | $4.2 \pm 0.3$ | $5.7 \pm 0.5$ | $10.4 \pm 1.0$ | $20.2 \pm 2.4$ | $51.1 \pm 3.4$ | $152.9 \pm 8.7$ |
| ionosphere | acc | $.903 \pm .045$ | $.892 \pm .057$ | $.923 \pm .048$ | $.906 \pm .043$ | $.909 \pm .079$ | $.900 \pm .057$ | $.883 \pm .079$ |
| | nodes | $1.1 \pm 0.3$ | $1.8 \pm 0.6$ | $3.8 \pm 1.1$ | $10.5 \pm 2.3$ | $23.3 \pm 4.2$ | $40.5 \pm 7.4$ | $58.8 \pm 6.4$ |
| | path len | $1.03 \pm 0.10$ | $1.42 \pm 0.35$ | $2.29 \pm 0.48$ | $4.38 \pm 0.68$ | $6.01 \pm 0.88$ | $7.67 \pm 0.88$ | $10.00 \pm 2.49$ |
| | gini | $.894 \pm .012$ | $.887 \pm .029$ | $.839 \pm .032$ | $.705 \pm .022$ | $.615 \pm .051$ | $.596 \pm .051$ | $.594 \pm .060$ |
| | time | $0.7 \pm 0.1$ | $0.8 \pm 0.1$ | $1.2 \pm 0.2$ | $2.6 \pm 0.5$ | $5.4 \pm 0.9$ | $11.4 \pm 1.9$ | $16.2 \pm 1.6$ |

# 3 Visualization of MNIST Trees

In this section, we show the full decision trees trained on MNIST and Fashion-MNIST that were reported in the main paper, along with one randomly selected prediction from from the training set for each class. The trees with $\alpha = 10^{-5}$ are not shown because of their large size. While the larger trees may be too small to be legible in print, the image resolutions are high, so zooming in to a digital view of this document will make the details visible.

## 3.1 MNIST $\alpha = 10^{-2}$

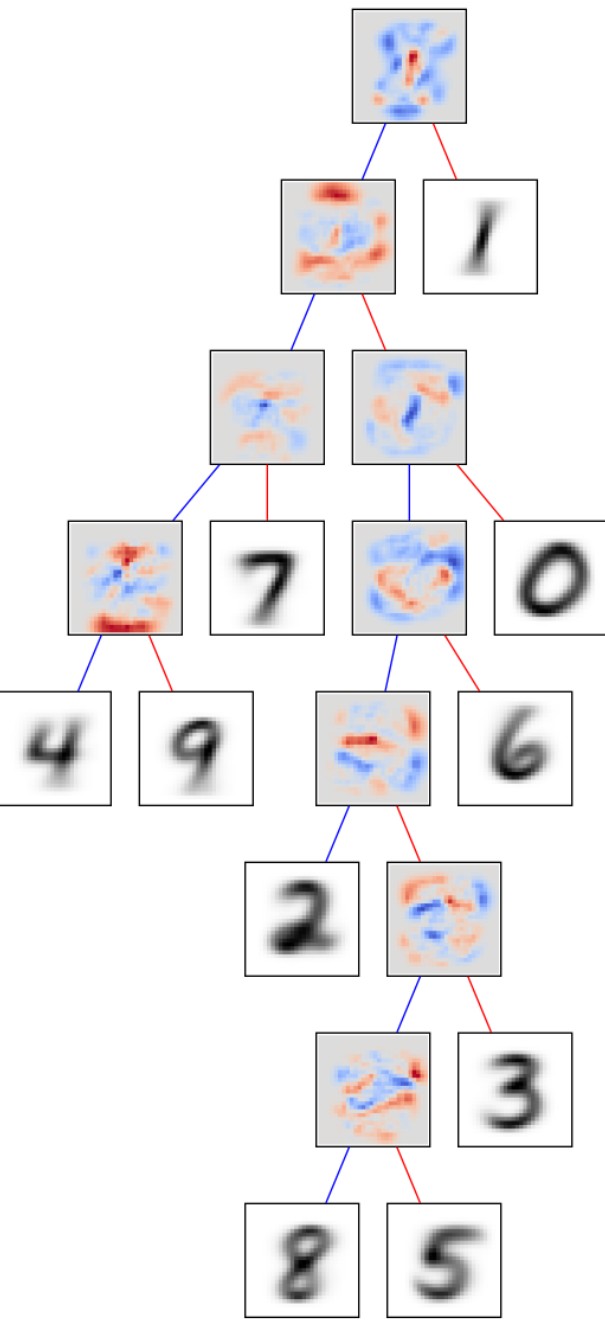

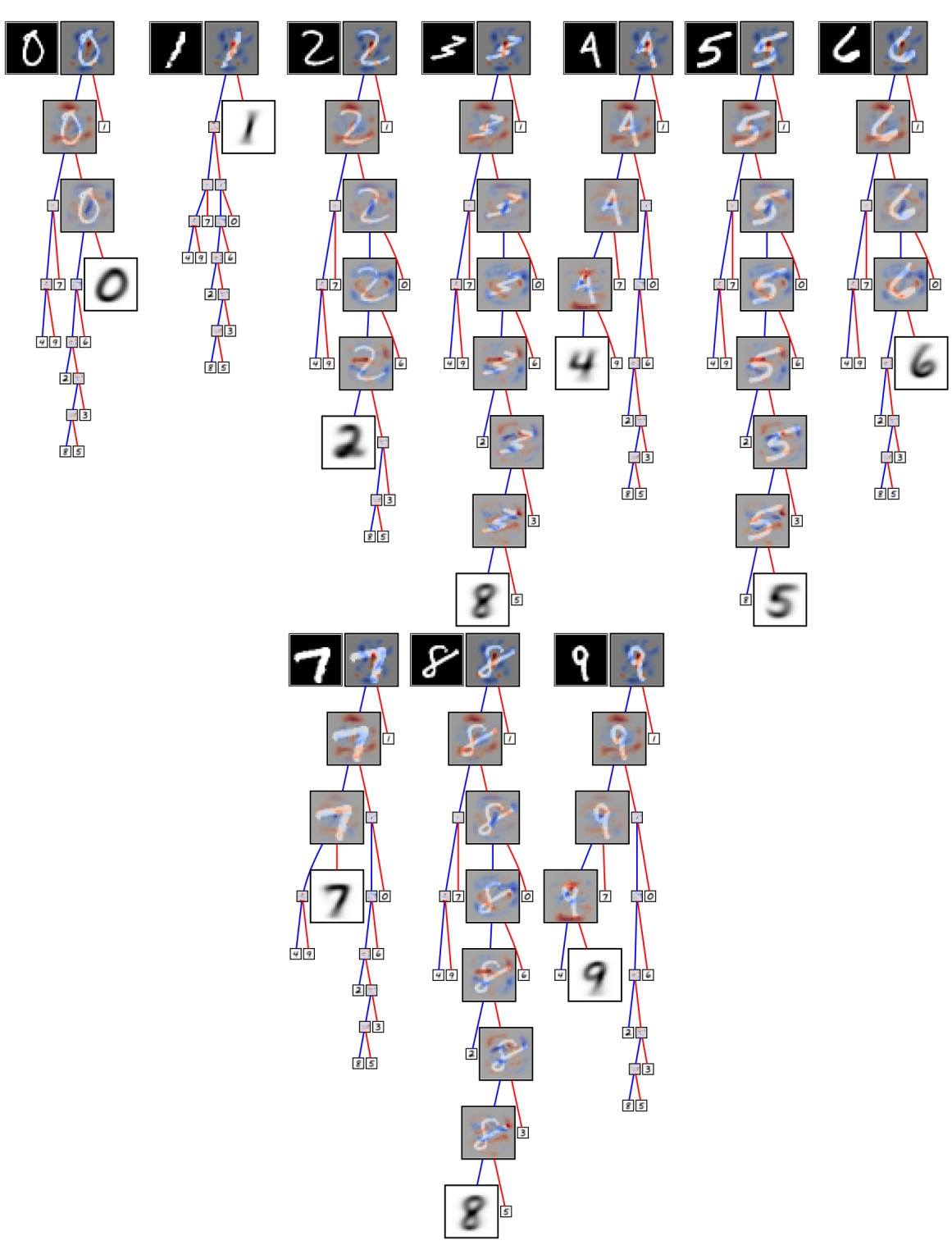

## 3.2 MNIST $\alpha = 10^{-3}$

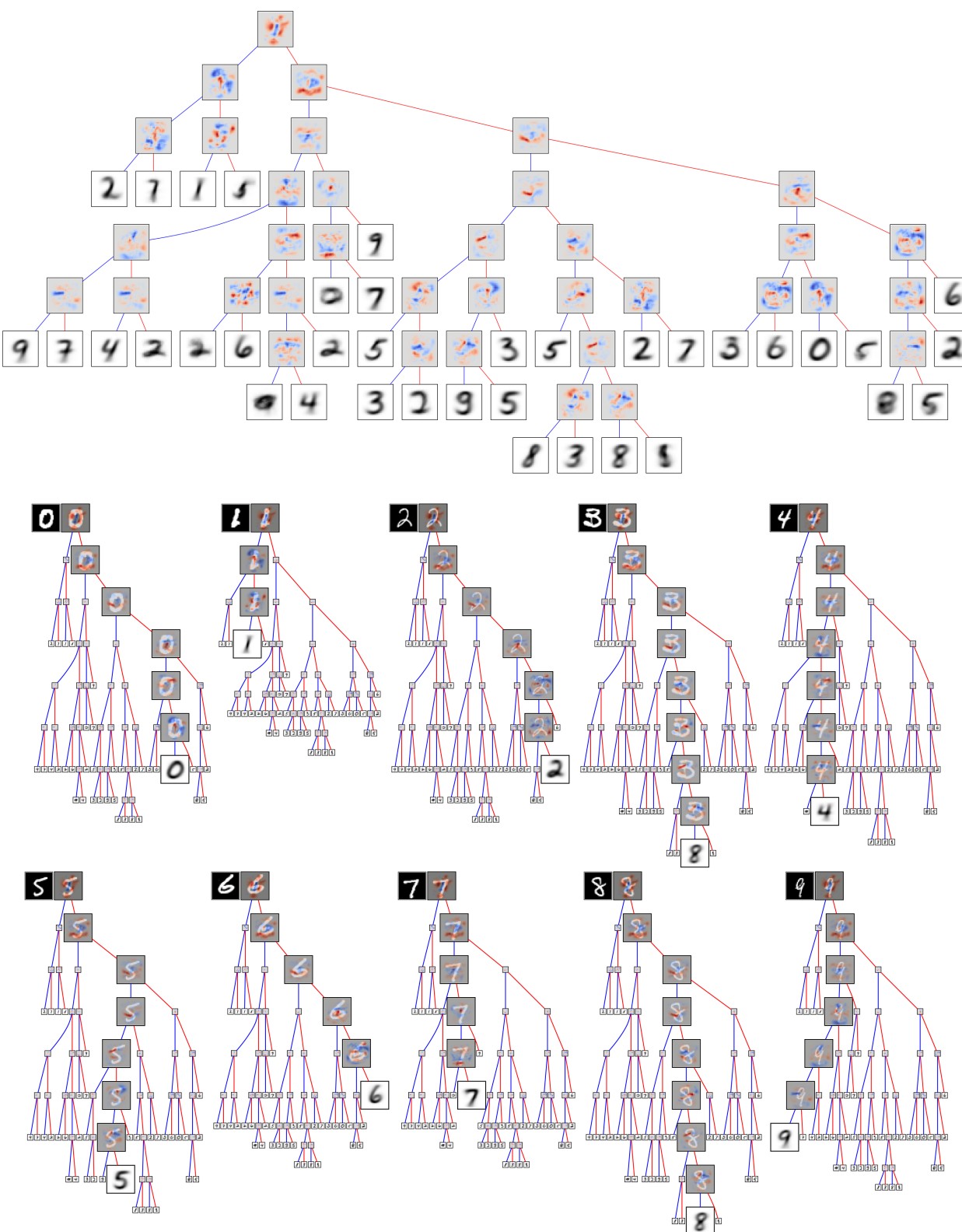

## 3.3  MNIST $\alpha = 10^{-4}$

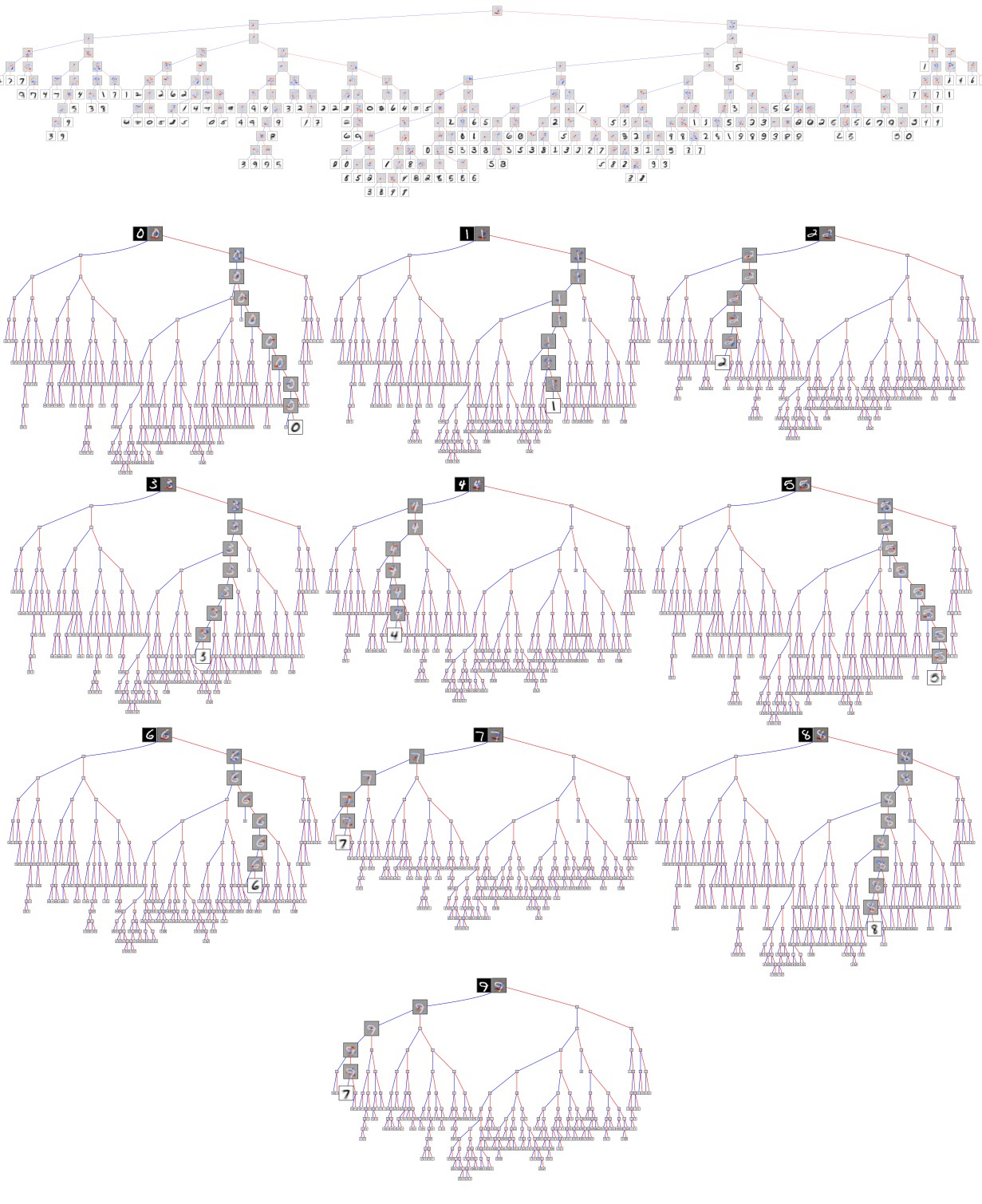

## 3.4    Fashion-MNIST $\alpha = 10^{-2}$

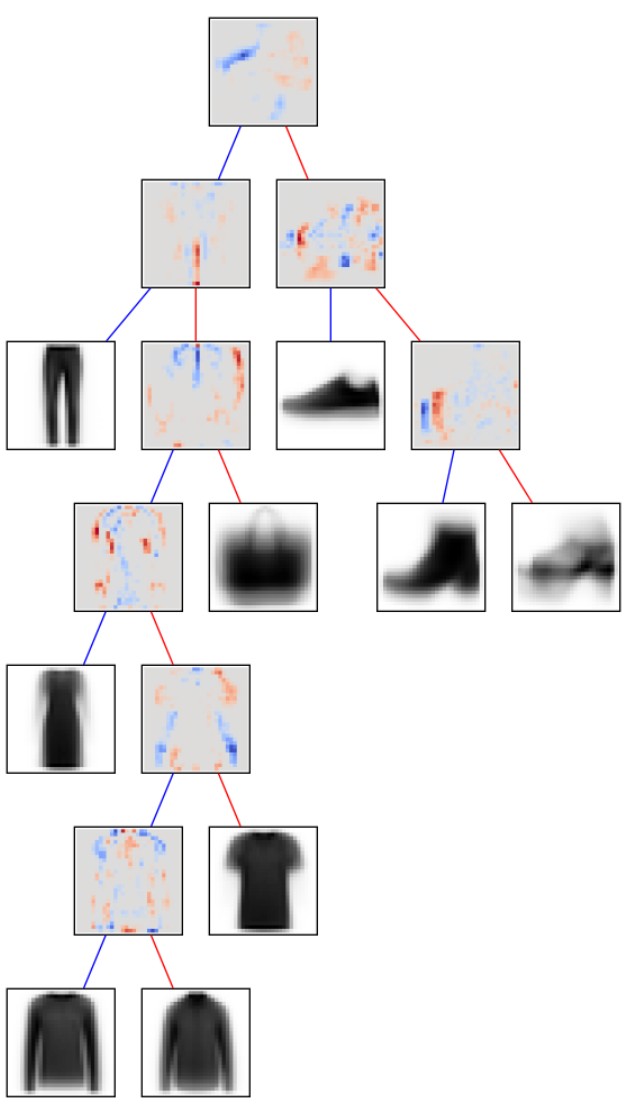

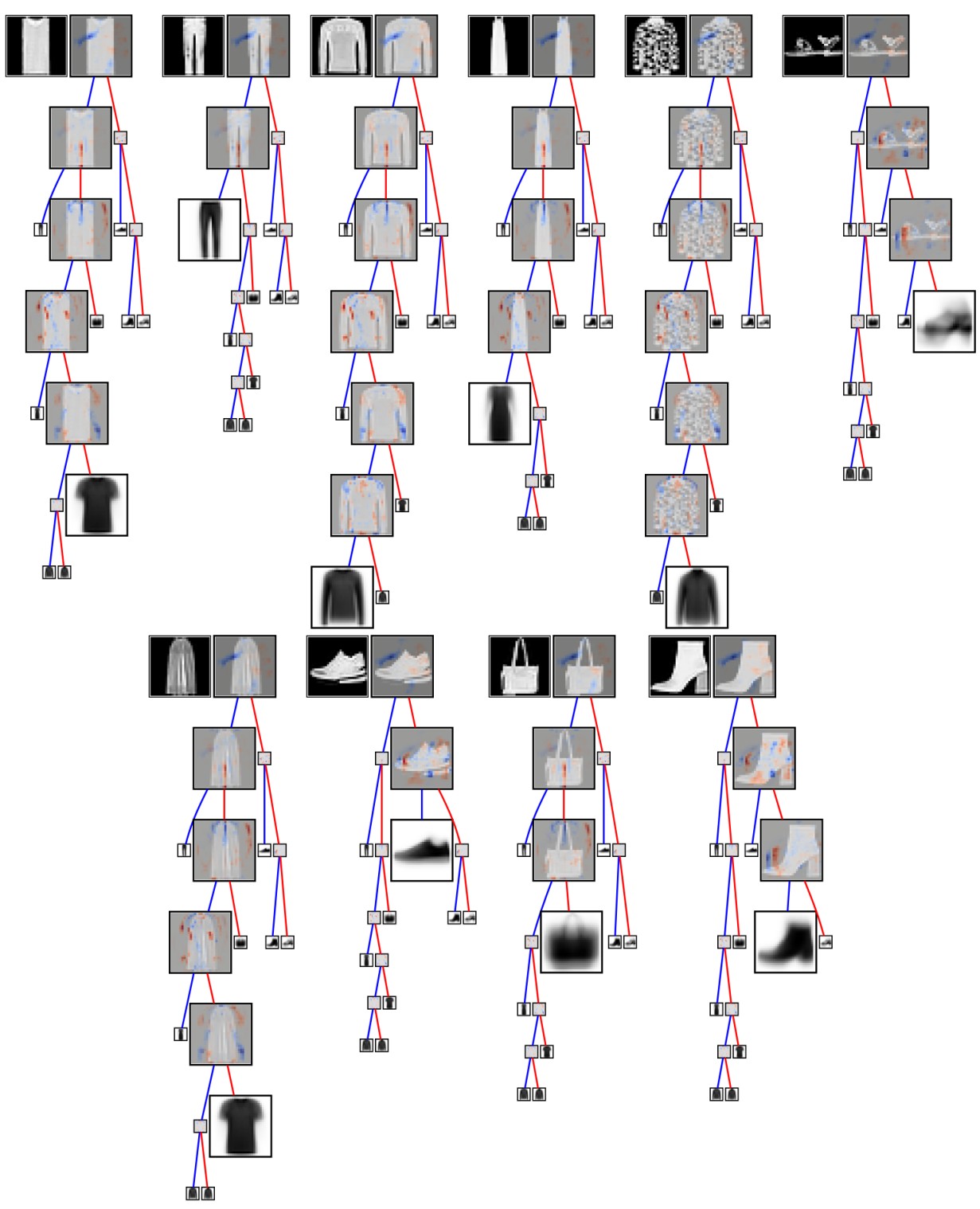

## 3.5   Fashion-MNIST $\alpha = 10^{-3}$

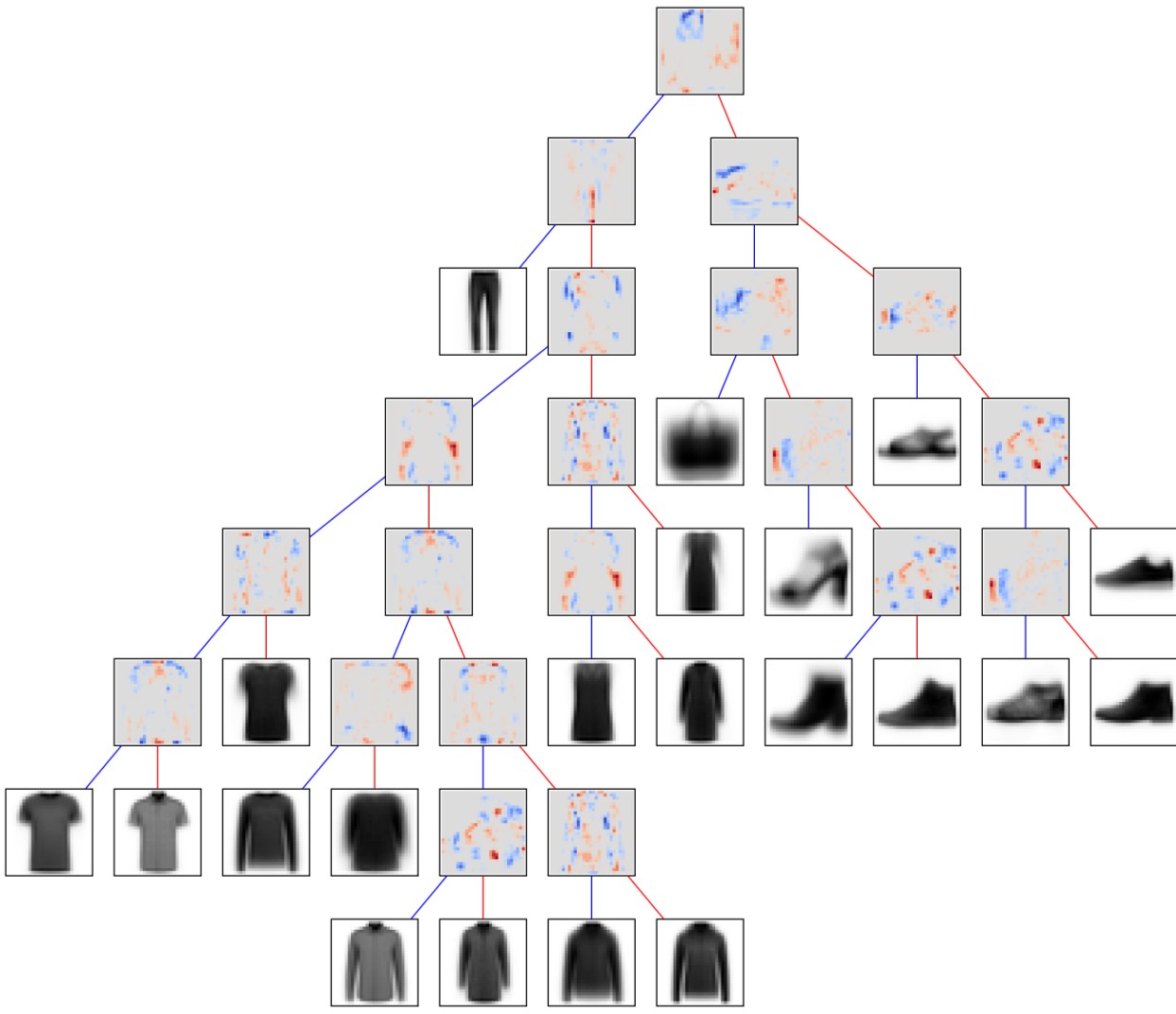

## 3.6 Fashion-MNIST $\alpha = 10^{-4}$

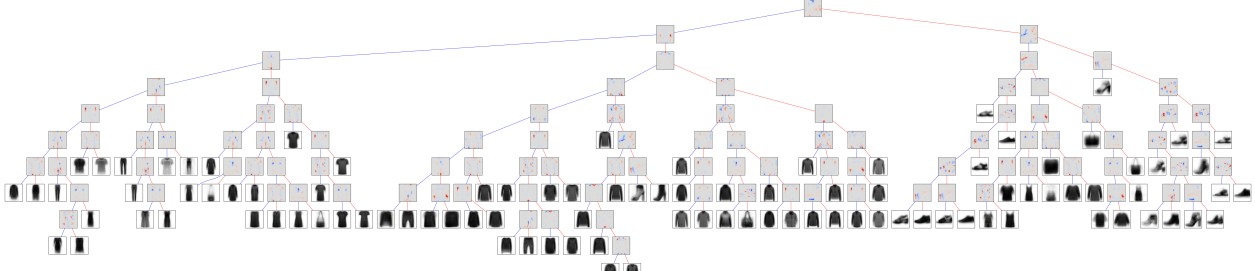

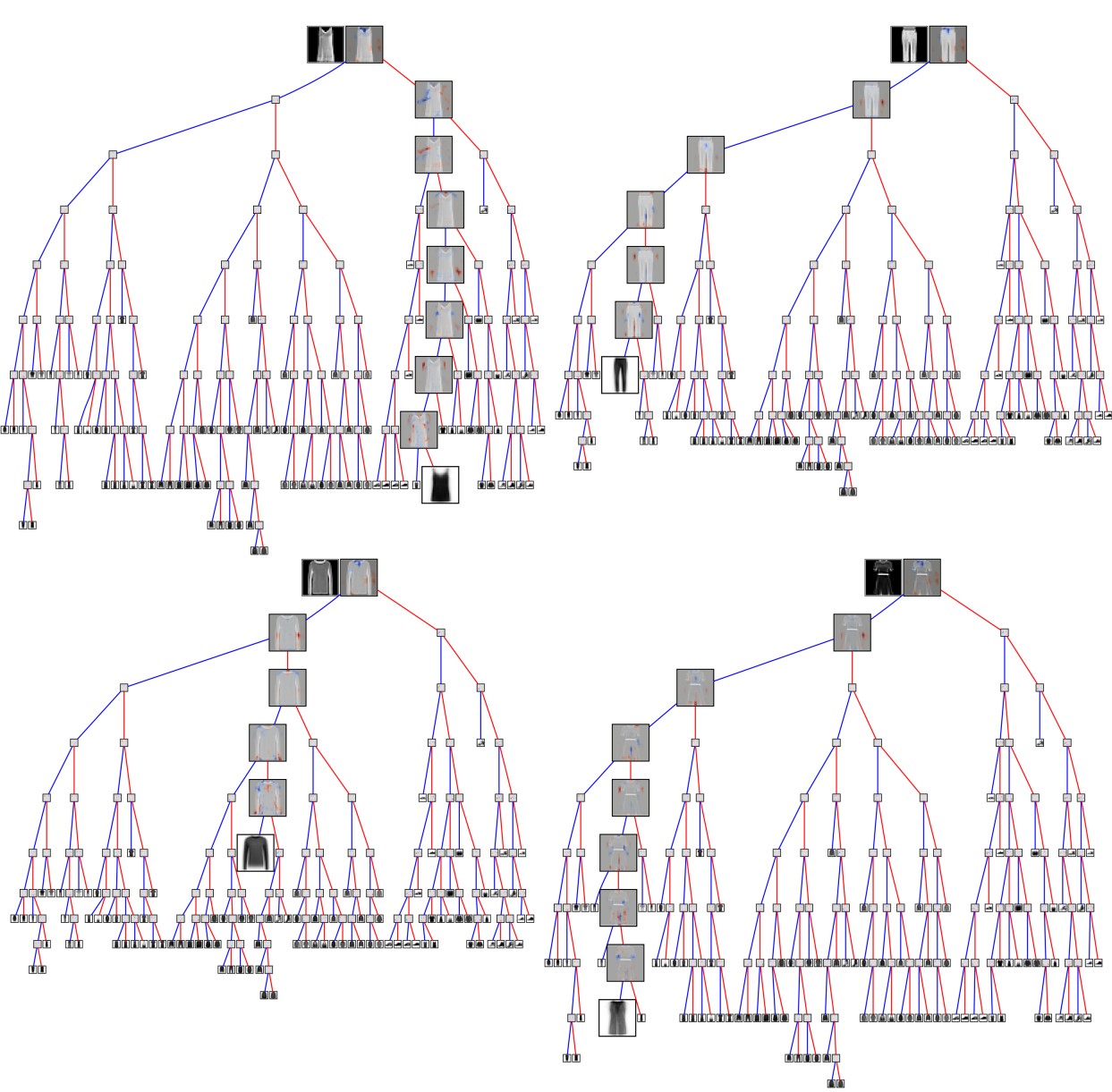

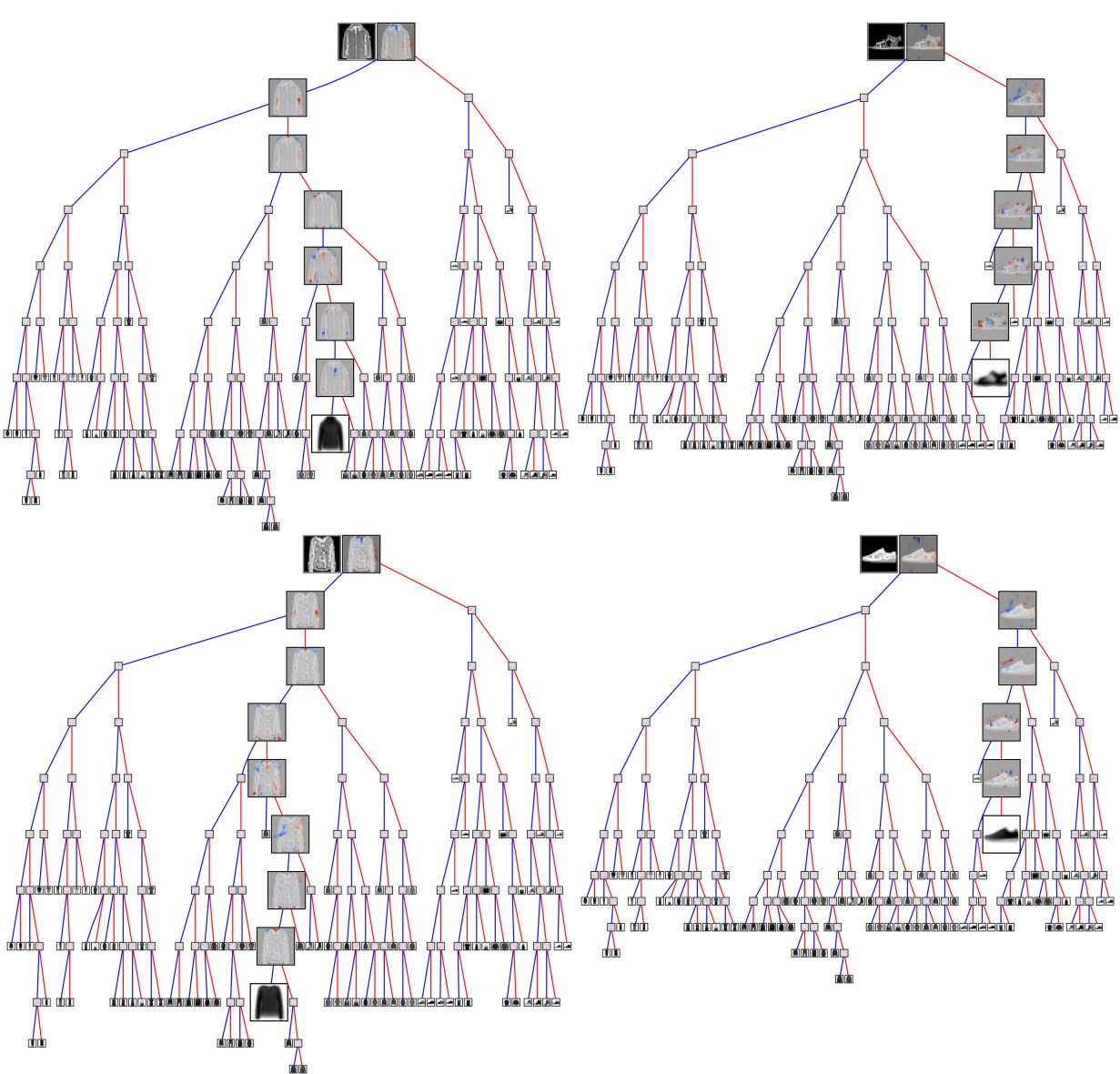

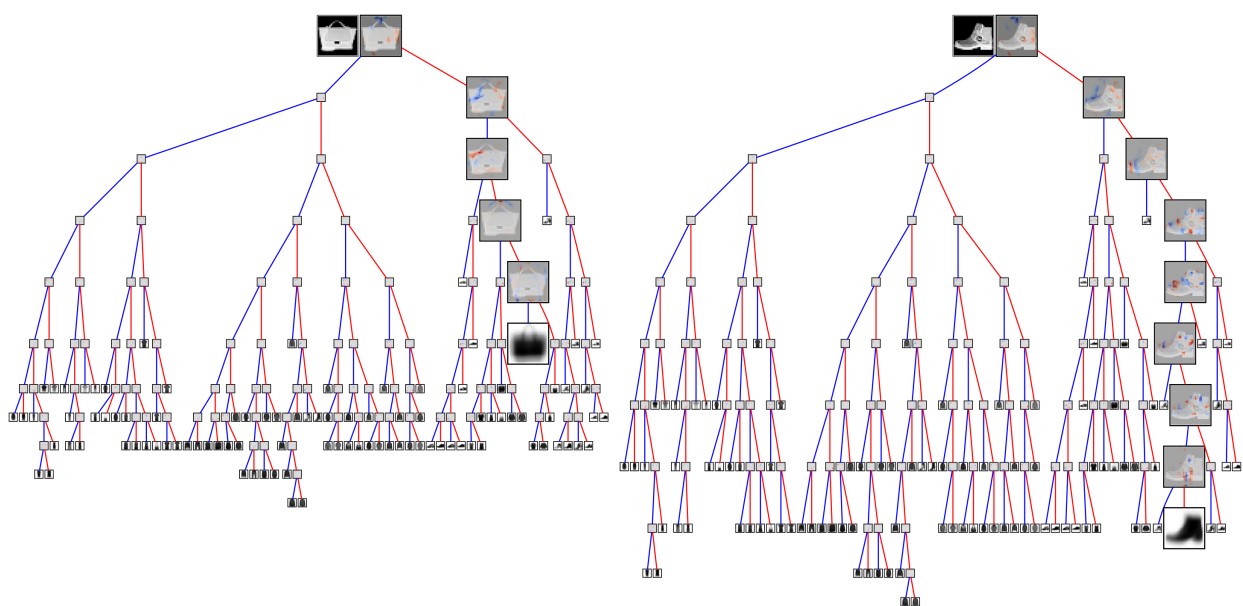