# OpenReview forum: "Feature Learning for Interpretable, Performant Decision Trees"
_NeurIPS.cc/2023/Conference — NeurIPS 2023 poster_

### Official Review · Reviewer_js5S · 2023-06-25

**Soundness:** 3 good
**Presentation:** 2 fair
**Contribution:** 2 fair
**Rating:** 5
**Confidence:** 4

**Summary:**

The paper proposes a method to learn interpretable decision trees by keeping the tree depth limited without sacrificing the performance. Keeping the tree small is achieved through multiple iterations of feature transformation followed by regrowing the tree. Authors have utilized linear and distance to prototype feature transformation. They apply regularization to improve interpretability through increasing feature sparsity and shrinking the output range of the features which results in purer leaves.
Authors report empirical results evaluating their method against decision tree, random forests, and EstraTrees on a set of UCI repositories. Experiments report accuracy , average number of nodes in model, average decision path length, feature sparsity, and inference time (all metrics except accuracy are reported in supplementary material). Results show that in 4 out of 6 data sets, a version of the proposed method outperforms the baselines.

**Strengths:**

- Originality: The work builds on top of existing literature on feature transformation, oblique decision trees, CART, differentiable decision trees, etc. The proposed way of alternating through feature transformation and tree re-growth is fairly novel and the introduced regularizations make sense.
- Quality: The quality of the paper is overall fair. Motivation is clear, related literature is well covered and experimental results are clearly reported with more details in supplementary materials. Refer to weaknesses for rationale on why overall quality is fair.
- Clarity: The paper is written clearly for the most part. I was not able to completely follow the discussions on MNIST data sets and it is not clear to me how the proposed method reforms compared to the baseline approaches in terms of accuracy and explainability for MNIST.
- Significance: The work is generally interesting with potentially significant applications. However, the shortcomings that I've listed in the weaknesses sections cast some doubt on the significance of the proposed method as presented.

**Weaknesses:**

- Interpretation of interpretability: Authors measure interpretability in terms of the tree size and use feature transformation and regularization to keep the tree size under control. However, one can argue that the feature transformation (e.g. as in application of ODT with non axis-aligned partitioning, or distance to prototype) can adversely impact interpretability. For example, consider figure 2 where a node contains a linear combination of a set of features as the decision criteria which is not very interpretable compared with an axis-aligned simple DT method. To get a better idea, consider a domain where raw features are more tangible and interpretable compared to the wine data set such as income prediction based on age where applying a coefficient to a well understood feature such as age and combining it with other scaled features make the whole tree less interpretable.
- Limited and inconclusive competitive analysis: Authors have compared their method against DT, RF, and ET. Experimental results are limited because authors have not considered linear methods such as linear regression which are arguably comparable with the proposed methods in terms of interpretability. Reason is that linear methods provide one linear formula consisting of the input features whereas the proposed method (with linear transformation) consists of multiple nodes each with a linear combination of a limited set of features. They are inconclusive since, the performance of the proposed method varies drastically based on the alpha value which is not picked programmatically (based on cross validation) like what authors have done for DT. Lastly, the baseline methods do not help getting a clear idea about the delta in accuracy between non-interpretable high performing methods (e.g. xgboost) and more interpretable alternatives.

**Questions:**

- It would be useful to understand author(s)' perspective on linear transformation of input features on interpretability.
- Can authors clearly explain the main rationale behind reporting MNIST data sets?
- How would authors justify that the intermediate value of alpha for the most part has the highest performance, although the alpha values that represent highest complexity are expected to outperform the intermediate alpha value.
- Can author shed more light into the big delta between their method's performance for various alphas and how this could be addressed practically so that a fixed programmatically selected alpha can be used with high confidence?

**Limitations:**

- Time complexity: As stated by the authors, proposed approach requires longer training time due to repeatedly fitting a tree to the data. It would have been more useful if experiments instead of inference time would have reported the training time.

---

> ### Author Rebuttal · Authors · 2023-08-10
>
> In response to strengths - clarity: we did not include baseline results for MNIST and fashion-MNIST because they are widely studied and have a set train-test split. It is easy to find baseline results, but we can include them in the supplementary material if it is desired. We use MNIST for some comparison against TAO, a recent approach for training ODTs, but we cannot reproduce their methods, and so we reference their paper for the results. If you have specific concerns about the clarity of this discussion, we would be happy to answer questions and revise the text to improve understanding.
>
> In response to the weaknesses:
>
> We agree that sparse linear or prototype features on their own are less interpretable than single-feature splits used in conventional decision trees. However, we argue that the substantial reduction in tree size and, more importantly, the ability to perform well with a single tree instead of an ensemble more than make up for it. For example, you reference figure 2; a decision tree fit to the same data is on the order of 10x larger and performs very poorly, and an ensemble that performs well is hundreds of times larger and involves aggregating the decision from many models. This argument is generally accepted and is the basis for several of the referenced related works, one prominent of which ([9] in our submission) was published in NeurIPS.
>
> We have added baselines to the experiments. Linear models are very interpretable, but do not perform well on data that is not linearly separable. Moreover, our method can be simpler for multi-class problems because linear models have one linear formula per class along with a softmax function. Because our trees are used sparsely during prediction, a balanced tree can separate n classes with as few as log(n) linear rules on average per prediction. MNIST is a good example; there are 10 classes, but our smallest tree uses far fewer than 10 linear rules per prediction on average. We also remind that the models in the benchmarks in the main paper are using *sparse* linear features.
>
> As for variance of performance with alpha, this is expected. It is used to trade off size for expressive power via early stopping of tree growth. We showed different alpha values to demonstrate this. To make the comparison fairer, in the revision, we report benchmark results with each model having alpha chosen by cross-validation of the tree on the training data transformed with the final learned feature parameters.
>
> We also added highly expressive models such as MLP and XGBoost to the experiments.
>
> In response to the questions:
> - See the first paragraph of our response to the listed weaknesses.
> - The MNIST models are presented to (1) demonstrate performance on a large, many-featured data set; (2) establish the use case of our methods on simple image problems and motivate future work on more challenging image problems; (3) provide compelling visual examples of very small, interpretable trees for complex classification problems; (4) demonstrate the potential usefulness of the hierarchical structure resulting from greedy tree growth, as in the fashion-MNIST figure; and (5) provide some comparison against TAO, which is also benchmarked on MNIST in [9], since MNIST has a pre-determined train/test split and so such a comparison is valid.
> - It is not true that models with higher complexity always outperform models with lower complexity. While our trees are less prone to overfitting than standard decision trees, they can still overfit if allowed to grow too large, which is why we see reduced performance for large alpha on some data sets.
> - To expand on the prior point, alpha being too small results in underfitting, and alpha being too large results in overfitting. This is no different from conventional decision trees, though since our models with transformed features are more expressive, the ideal tree size is smaller and has better performance. Alpha can be efficiently selected programmatically by doing cross-validation on the training data with the final learned features. We have added benchmark results with alpha selected in this way in the revision.

---

> > ### Comment · Reviewer_js5S · 2023-08-19
> >
> > Thanks for your responses to my comments.
> >
> > Reporting results where alpha is chosen by cross-validation addresses my concerns about fairness of the evaluations.
> > Also, authors responses to my 4 questions are satisfactory for the most part. Regarding my question on intermediate alpha values having the best performance authors have mentioned overfitting which can be addressed by regularization (which is utilized in the proposed method). No further action/comment from authors needed here though.
> >
> > Given the authors' responses and the revisions made to the submission. I am willing to raise my evaluation to borderline accept.

---

### Official Review · Reviewer_ejNS · 2023-06-30

**Soundness:** 3 good
**Presentation:** 3 good
**Contribution:** 2 fair
**Rating:** 5
**Confidence:** 4

**Summary:**

In this paper the authors study the problem of learning meaningful feature representation for decision trees. They propose to interleave the learning of parametrized feature representation by gradient descent, and the learning of tree structures by standard greedy method, e.g. CART. A kernel density decision tree is used for the former step for differentiability.

The authors propose a few practices to achieve better performance, which includes a couple of classes of feature transformation, regularization and the choice of KDDT kernel. They empirically show that the proposed method easily outperform similar existing methods, and the learned feature representation are visually meaning when the method is applied to image datasets.

**Strengths:**

1. The paper studies a legit and important research question, that it provides a convincing potential solution to the question of effectively applying decision trees on higher dimensional datasets with proper feature learning.
2. The proposed method has good uses of modern decision tree techniques. I think the authors have exhausted the headroom in their scope, and there is no obvious theoretical improvement left to make.
3. The writing quality is good. The paper is clear, concise and easy to follow.

**Weaknesses:**

One of my major concerns is the divergence of the proposed method from a traditional decision tree and its implied impact on model performance and interpretability. Kernel based fuzzy decision trees are a compromise to differentiability, which already results in a prediction pattern that is not the interpretable yes-no at each node, adding extra computation and more mental overhead. This plus the learned linear feature combination, about whose stability I have doubt, might push the proposed method more like a NN in the shell of a DT.  These observations might have undermined the significance of the submission.

Another concern is about novelty, that as mentioned by the authors, the interleaved learning of tree structure and feature representation has been studies. If so the novelty of the paper is the proposed of several practices of better feature representation learning, which might fail to justify as a NeurIPS publication.

**Questions:**

1. Regarding novelty, I wonder what is the key difference between the proposed method and prior work, as the phrasings in Line 128-131 and Line 132-135 sound almost the same.
2. Both the study on the tabular data and on the image datasets suggest that the linear projection is the most important feature transformation, which is also the most compatible with the sparsity regularization. What exactly do other feature transformations, e.g. distance to prototype and fuzzy cluster membership, bring here? Any theoretical / empirical supporting evidence?
3. Given the linear projection transformation is done in a fashion very similar to the learning of a NN dense layer, I wonder how stable the proposed method is, in particular
a. How's convergence in practice with respect to the learned linear projections? Is the learning robust / can we always reach the same linear features / interpretation of the resulting tree?
b. How much does the tree T structure changes during model fitting? The introduction of growing / pruning methods (L131) actually prevent the tree structure from changing too much, and why do we not consider such strategies?
4. The benchmarking against tree methods is less meaningful given the feature learning perspective of the proposed method. A closer comparison is probably between it and a shallow MLP.
5. Line 126. T \circ f_{\theta}.

I am open to raise my rating should the aforementioned questions be addressed.

**Limitations:**

Yes.

---

> ### Author Rebuttal · Authors · 2023-08-10
>
> In response to the weaknesses:
>
> The fuzzy decision component is really only important to achieve differentiability in this context. It can increase performance slightly, but the feature transformation is more important. While most KDDT decisions are the same as the corresponding crisp tree, we agree that those decisions that are fuzzy may be viewed as less interpretable. If this is a concern, one can simply not use fuzzy decisions for prediction. To reflect this, in the revision, we report results using crisp trees as the final predictor so that these models can be interpreted unambiguously as using a single decision path for each prediction. Most of the scores do not change much as a result. A notable exception is that the largest MNIST tree drops from 97.08% to 95.75% test accuracy.
>
> With or without fuzzy decisions, we disagree that these models are comparable to a neural network. They are sparsely activated for a given prediction, and decisions are a logical series of rules rather than a composition of functions. The overall size is substantially less than a neural network, and the partitioning structure itself aids interpretation by grouping like data; a strongly illustrative example is the visual of the fashion-MNIST tree. The consensus of models like this being valuable for interpretability is backed up by a history of published work learning ODTs for the purpose of interpretability, including [9], which was published at NeurIPS.
>
> As for novelty, we want to clarify that the interleaving of tree fitting with feature learning, as you put it, has not been studied before this work. Existing approaches fit a (crisp) tree at the start, then make it fuzzy and learn features without regrowing the tree.
>
> In response to the questions:
> 1. The difference described in the referenced lines is that existing approaches only grow the tree up-front, and they use a crisp tree growing algorithm. We fit continually throughout the feature learning process, which is a simple and principled way to revise the tree structure based on the changes in input; however, it requires an efficient and stable *fuzzy* decision tree growth algorithm, which we are the first to apply in this way.
> 2. To keep the scope of the experiments reasonable, we only tested linear and prototype features, which we expect to be the most generally applicable in terms of performance and interpretability. The linear features do typically outperform prototype features, but there are cases where the opposite is true. We propose a variety of features in order to best meet modeling and interpretability needs for possible applications, and plan to study specialized domains in future work.
> 3. This is, of course, nonconvex optimization. Convergence is not perfectly smooth, especially for small tabular data sets, but generally we consistently arrive at a good model. We did not look at the tree for every experiment, but for the ones we created visuals of (wine, MNIST, fashion-MNIST), the resulting tree structure and features were consistent over multiple runs with different initialization. For MNIST and fashion-MNIST, the convergence of the loss function was especially smooth and stable, maybe because of the large size of the data sets. Substantial changes to tree structure are frequent early during the learning process, but once the feature transformation parameters start to converge, tree structure changes are infrequent. The complete regrowth approach is desirable because it is not limited to trees that are similar to the previous iteration. Approaches that just trim and grow are not able to change the fundamental layout of the tree. We believe this is why we see smaller, better-performing trees on MNIST compared to TAO, a recent alternative. Also, while DT growth is a heuristic, it is a proven and widely accepted strategy for producing approximately minimal decision trees, and as demonstrated in the fashion-MNIST example, the property that splits are locally most informative can have interpretability benefits.
> 4. We did not think it was important to compare against black-box models, but we have added MLP to the benchmarks in the revision. They are the best-performing model on about half the data sets.
> 5. You are correct - thanks for catching this mistake. We have fixed it in the revision.

---

> > ### Comment · Reviewer_ejNS · 2023-08-19
> >
> > I'd like to thank the authors for their explanation. Bumped up my rating to 5.

---

### Official Review · Reviewer_b4Qh · 2023-07-04

**Soundness:** 3 good
**Presentation:** 2 fair
**Contribution:** 2 fair
**Rating:** 6
**Confidence:** 3

**Summary:**

The paper proposes a novel approach that learns sparse features and constructs decision trees in a differentiable manner. This approach strikes a balance between the complexity of decision rules and the overall model, resulting in small, interpretable trees. Specifically, the authors apply (1)alternating optimization method for trees and features (2)kernel density decision trees (3)different parameterized feature transforms (4)regularization method.
They compare their method against popular tree-based baselines and demonstrate its comparable performance in terms of accuracy while producing smaller models.

**Strengths:**

1. The approach proposed by the paper is novel that combines sparse feature learning with differentiable decision tree construction to produce small, interpretable trees. This approach is different from traditional decision tree models and provides a new perspective on feature learning for interpretable models.

2. The paper provides a comprehensive empirical evaluation of the proposed approach on various datasets. The authors compare their method against popular tree-based baselines, including decision trees, random forests, and ExtraTrees. The results show that the proposed approach outperforms or comes very close to the best baseline's accuracy while being much smaller.

3. The paper is well-written and gives detailed explanations for each technique used.

**Weaknesses:**

1. Lack of comparison with other models: The paper compares the proposed approach against only a few traditional tree-based models like decision trees, random forests, and ExtraTrees. Comparing with more classic baselines can better illustrate the effectiveness of the approach


2. Clarity: The paper covers a variety of techniques (such as KDDT), many of which are improvements built upon previous work. It would be beneficial to clearly highlight the unique contributions made by the authors.

**Questions:**

1.I noticed that in Table 1, when α = 0.1, the accuracy on the dry-bean dataset significantly drops to 0.476/0.354. What could be the reason behind this sharp decline? I am somewhat concerned about the robustness of the method.

2.The models using prototype features consistently fall behind the models employing linear features across all six datasets. In practical terms, what strategies can be employed to complement linear features with nonlinear features?

**Limitations:**

Yes，the authors have adequately addressed the limitations.

---

> ### Author Rebuttal · Authors · 2023-08-10
>
> In response to the weaknesses:
> 1. The intention was to compare against structurally similar models with some degree of interpretability in the standard ML repertoire. After all, the methods are proposed as an alternative to traditional tree-based models. However, in the revision, we have added additional baselines.
> 2. The final paragraphs of the introduction and related works specify the unique contributions. We have modified this text in the revision to make it clearer.
>
> In response to the questions:
> 1. This decline is expected. When alpha = 0.1, we are telling the model not to split unless the overall impurity (between 0 and 1) is improved by at least 0.1. When there are 7 different classes, it is not unexpected that none of the initially available splits results in this much gain, so the final tree is just a single node, or maybe has one or two splits. For data with several classes, alpha = 0.1 is just too large. This is why we don't report alpha = 0.1 for the MNIST results.
> 2. These feature types can be used in conjunction in the same tree if desired. One would simply concatenate their respective outputs to get the input to the tree training algorithm. We do see that linear features usually outperform prototype features, but this is not true for every data set. In addition, one might choose one or the other based on which is more interpretable in a given application context.

---

> > ### Comment · Reviewer_b4Qh · 2023-08-17
> >
> > Thanks a lot for your clarification, I have read your response.

---

### Official Review · Reviewer_vnHK · 2023-07-07

**Soundness:** 3 good
**Presentation:** 3 good
**Contribution:** 2 fair
**Rating:** 5
**Confidence:** 4

**Summary:**

The authors aim to learn small and interpretable decision trees with good predictive performance. To this end, the authors propose efficient differentiable trees based on the kernel density decision trees. To ensure the small size of the tree and increase the model's representability, the authors propose some parameterized feature transformations. Due to its differential structure, the model is capable of learning the parameters in both the tree and feature transformation simultaneously. The experimental results demonstrate the predictive performance and interpretability of the proposed model.

**Strengths:**

The paper is well written and easy to follow. The idea of learning a parametrized feature transformation is interesting as it enhances the model's expressiveness while still maintaining its interpretability. The authors provided extensive experimental results which demonstrate the interpretability of the model.

**Weaknesses:**

1. The main contribution of the paper is the proposed differentiable tree and parameterized transformation. Nevertheless, the differentiable tree is not a new concept. Although the idea of learning a transformation during training is interesting. Similar ideas, such as neural prototypes, have been previously proposed in the computer vision domain. See [1] for an example.
2. No guideline is provided to recommend how to choose transformations in different tasks for users.
3. The datasets used to show the prediction performance are relatively small.
4. Some technique details are missing. For example, how these regularizations are used when learn the tree is unclear to me.

Ref:
[1] Nauta, Meike, Ron Van Bree, and Christin Seifert. "Neural prototype trees for interpretable fine-grained image recognition." Proceedings of the IEEE/CVF Conference on Computer Vision and Pattern Recognition. 2021.

**Questions:**

1. In line 147, “we assume a single marginal F_i=F…”. Where is the assumption used in the following part? In the same line, what is F’?
2. Is the feature transformation mentioned in line 172 the same as these described in Section 3.4? If so, how to guarantee the output range are limit for identiy and element-wise transformation?
3.What is $\alpha$ in Section 4? Is it the same as that in line 207? If so, please provide some discussion why larger $\alpha$ leads to more sparse tree. If not, please provide details how to control the complexity of the model by tuning $\alpha$?
4. Is “ours:linear” in Table 1 the same with ODT? If not, please provide experiments results using ODT.

**Limitations:**

Yes

---

> ### Author Rebuttal · Authors · 2023-08-10
>
> In response to the weaknesses:
> 1. What most clearly separates this work from existing work, methodologically speaking, is that we are the first to re-fit the tree continually throughout training. This is only possible because of recent advances in fuzzy decision tree techniques and results in the structure changing throughout training, resulting in desirable properties of the final model, as described in the paper. We are also the first to weight regularization by feature use in the tree, which results in more uniform sparseness.
> 2. Transformations can be chosen by a combination of the interpretability needs of the application (some applications may have different standards for interpretability than others) as well as validation results that suggest which features result in the best performance. For this paper, we focus on tabular and simple image data, but plan to work on other modalities in the future.
> 3. The tabular results contain some moderately sized data sets, such as dry-bean (13611 samples) and car (1728 samples). The results on MNIST and fashion-MNIST demonstrate use with relatively large data sets.
> 4. If there are specific questions about the methods, we will be happy to answer them in the discussion and clarify in the revision.
>
> In response to the questions:
> 1. That assumption was just meant to simplify notation, but we ended up not using it. We removed the assumption in the revision. Thanks for catching this oversight.
> 2. Yes, the strategies on line 172 are intended to be used with the feature transformations in section 3.4. Generally we use the other approach, but if we want to limit the output range of feature transformation, there are many possibilities. For identity, the output range is already limited to the range of the actual training data; in practice, we normalize the data. For element-wise, we can choose a transformation that maps a bounded domain to a bounded range, or apply an "activation" such as sigmoid.
> 3. Alpha in section 4 is the cost-complexity pruning parameter. This is different from the alpha in section 3.4. We have changed notation to avoid confusion in the revision. Controlling size and overfitting using alpha is standard practice for decision trees, and it is used here as usual. In brief, larger alpha produces smaller trees by stopping growth sooner.
> 4. ODT refers to any decision tree using linear splits. In that sense, "ours: linear" is indeed an ODT.

---

> > ### Comment · Reviewer_vnHK · 2023-08-18
> >
> > Thanks for performing and providing more experimental results, including more experimental baselines and performance of the proposed method with programmatically selected hyperparameters. Also thanks for providing more explainations on the methodology novelty.
> >
> >
> > In terms of the size of dataset tested in the experiments, as argued by the authors,  "(experimental results) contain some moderately sized data sets" and MNIST with relatively large data sets. From the practical and industry perspective, I have to say the selection of these datasets is OK, but not impressive. I would recommend the authors to try on more practical and large-scale datasets with at least millions of samples and hundreds of dimensions.

---

### Author Rebuttal · Authors · 2023-08-10

Thanks to the reviewers for your feedback and insights.

The reviewers' main concerns relate to misunderstanding of the methods and their novelty, misgivings about the interpretability of the proposed models, and the limitations of the scope and presentation of the experiments. We have addressed these through our replies to the individual reviewers and a revision that we plan to submit once revisions are unlocked for the camera-ready version. Given our overall borderline scores, we strongly believe that this qualifies the paper for acceptance.

In particular, we highlight that (1) we are the first to alternate decision tree fitting with feature parameter learning, which is only made possible by recent advancements in fuzzy decision tree methods, and we are also the first to weight regularization of the feature parameters by the usage of each feature in the tree; and (2) models like these, especially ODTs, are generally accepted as being beneficial to interpretability compared to conventional tree-based models, as demonstrated by the related works. One of these, [9], was published in NeurIPS.

A summary of changes made in the revision:
- Add experimental baselines.
- Modify some text in an effort to preclude misunderstandings and answer questions present in the reviews.
- Report performance for our models with programmatically selected hyperparameters.
- Report performance without fuzzy predictions to address doubts about interpretability with fuzzy trees.

---

### Decision · Program_Chairs · 2023-09-21

**Decision:**

Accept (poster)

**Comment:**

The paper presents a new method for learning small decision trees, leveraging sparsity in differentiable tree learning. The reviewers praised the clear writing, the originality of the idea of a parametrized feature transformation and the comprehensive experiments showing near-sota performance with much smaller models. The reviewers also raised some concerns about baselines, differentiation compared to existing techniques and the metrics for interpretability. During the discussion period, the authors have added additional baselines, justified their metrics and convinced the reviewers of the distinction between their method and past methods using differentiable trees. Following the discussion section, all reviewers' scores are in the accept range. Given that the paper has a reasonably original idea, and that the experiments now have proper baselines, I recommend acceptance.